# Temporal Difference Learning with Compressed Updates: Error-Feedback meets Reinforcement Learning

**Aritra Mitra**                                                       *amitra2@ncsu.edu*
*Department of Electrical and Computer Engineering*
*North Carolina State University*

**George J. Pappas**                                                   *pappasg@seas.upenn.edu*
*Department of Electrical and Systems Engineering*
*University of Pennsylvania*

**Hamed Hassani**                                                     *hassani@seas.upenn.edu*
*Department of Electrical and Systems Engineering*
*University of Pennsylvania* *

**Reviewed on OpenReview:** *https://openreview.net/forum?id=dltUedmUVT*

## Abstract

In large-scale distributed machine learning, recent works have studied the effects of compressing gradients in stochastic optimization to alleviate the communication bottleneck. These works have collectively revealed that stochastic gradient descent (SGD) is robust to structured perturbations such as quantization, sparsification, and delays. Perhaps surprisingly, despite the surge of interest in multi-agent reinforcement learning, almost nothing is known about the analogous question: *Are common reinforcement learning (RL) algorithms also robust to similar perturbations?* We investigate this question by studying a variant of the classical temporal difference (TD) learning algorithm with a perturbed update direction, where a general compression operator is used to model the perturbation. Our work makes three important technical contributions. First, we prove that compressed TD algorithms, coupled with an error-feedback mechanism used widely in optimization, exhibit the same non-asymptotic theoretical guarantees as their SGD counterparts. Second, we show that our analysis framework extends seamlessly to nonlinear stochastic approximation schemes that subsume Q-learning. Third, we prove that for multi-agent TD learning, one can achieve linear convergence speedups with respect to the number of agents while communicating just $\tilde{O}(1)$ bits per iteration. Notably, these are the first finite-time results in RL that account for general compression operators and error-feedback in tandem with linear function approximation and Markovian sampling. Our proofs hinge on the construction of novel Lyapunov functions that capture the dynamics of a memory variable introduced by error-feedback.

## 1 Introduction

Stochastic gradient descent (SGD) is at the heart of large-scale distributed machine learning paradigms such as federated learning (FL) (Konečnỳ et al., 2016). In these applications, the task of training high-dimensional weight vectors is distributed among several workers that exchange information over networks of limited bandwidth. While parallelization at such an immense scale helps to reduce the computational burden, it creates several other challenges: delays, asynchrony, and most importantly, a significant communication bottleneck. The popularity and success of SGD can be attributed in no small part to the fact that it is extremely *robust* to such deviations from ideal operating conditions. In fact, by now, there is a rich

---

*This work was supported by NSF Award 1837253, NSF CAREER award CIF 1943064, and The Institute for Learning-enabled Optimization at Scale (TILOS), under award number NSF-CCF-2112665.

literature that analyzes the robustness of SGD to a host of *structured perturbations* that include lossy gradient-quantization (Seide et al., 2014; Alistarh et al., 2017) and sparsification (Wen et al., 2017; Stich et al., 2018). For instance, `SignSGD` - a variant of SGD where each coordinate of the gradient is replaced by its sign - is extensively used to train deep neural networks (Aji & Heafield, 2017; Bernstein et al., 2018). Inspired by these findings, in this paper, we ask a different question: *Are common reinforcement learning (RL) algorithms also robust to similar structured perturbations?*

**Motivation.** Perhaps surprisingly, despite the recent surge of interest in multi-agent/federated RL, almost nothing is known about the above question. To fill this void, we initiate the study of a *robustness theory for iterative RL algorithms with compressed update directions*. We primarily focus on the problem of evaluating the value function associated with a fixed policy $\mu$ in a Markov decision process (MDP). Just as SGD is the workhorse of stochastic optimization, the classical temporal difference (TD) learning algorithm (Sutton, 1988) for policy evaluation forms the core subroutine in a variety of decision-making algorithms in RL (e.g., Watkin's Q-learning algorithm). In fact, in their book (Sutton & Barto, 2018), Sutton and Barto mention: *"If one had to identify one idea as central and novel to reinforcement learning, it would undoubtedly be temporal-difference (TD) learning."* Thus, it stands to reason that we center our investigation around a variant of the `TD(0)` learning algorithm with linear function approximation, where the `TD(0)` update direction is replaced by a compressed version of it. Moreover, to account for general *biased* compression operators (e.g., sign and Top-$k$), we couple this scheme with an error-feedback mechanism that retains some memory of `TD(0)` update directions from the past.

Other than the robustness angle, a key motivation of our work is to design *communication-efficient* algorithms for the emerging paradigms of multi-agent RL (MARL) and federated RL (FRL). In existing works on these topics (Doan et al., 2019; Qi et al., 2021; Jin et al., 2022; Khodadadian et al., 2022), agents typically exchange *high-dimensional* models (parameters) or model-differentials (i.e., gradient-like update directions) over *low-bandwidth channels*, keeping their personal data (i.e., rewards, states, and actions) private. In the recent survey paper on FRL by (Qi et al., 2021), the authors explain how the above issues contribute to a major communication bottleneck, just as in the standard FL setting. *However, no work on MARL or FRL provides any theory whatsoever when it comes to compression/quantization in RL.* Our work takes a significant step towards filling this gap via a suite of comprehensive theoretical results that we discuss below.

## 1.1 Our Contributions

The main contributions of this work are as follows.

1. **Algorithmic Framework for Compressed Stochastic Approximation.** We develop a general framework for analyzing iterative compressed stochastic approximation (SA) algorithms with error-feedback. The generality of this framework stems from two salient features: (i) it applies to nonlinear SA with Markovian noise; and (ii) the compression operator covers several common quantization and (biased) sparsification schemes studied in optimization. As an instance of our framework, we propose a new algorithm called `EF-TD` (Algorithm 1) to study extreme distortions to the `TD(0)` update direction. Examples of such distortion include, but are not limited to, replacing each coordinate of the `TD(0)` update direction with just its sign, or just retaining the coordinate with the largest magnitude. *While such distortions have been extensively studied for SGD, we are unaware of any analogous algorithms or analysis in the context of RL.*

2. **Analysis under Markovian Sampling.** In Theorem 1, we show that with a constant step-size, `EF-TD` guarantees linear convergence to a ball centered around the optimal parameter. Up to a compression factor, our result mirrors existing rates in RL without compression (Srikant & Ying, 2019). Moreover, the effect of the compression factor exactly mirrors that for compressed SGD (Beznosikov et al., 2020). The **significance** of this result is twofold: (i) It is the first result in RL that accounts for general compression operators and error-feedback in tandem with Markovian sampling and linear function approximation; and (ii) It is the first theoretical result on the *robustness* of TD learning algorithms to structured perturbations. One interesting takeaway from Theorem 1 is that *"slowly-mixing" Markov chains have more inherent robustness to distortions/compression.* This appears to be a new observation that we elaborate on in Section 4.

3. **Analysis for General Nonlinear Stochastic Approximation.** In Section 5, we study Algorithm 1 in its full generality by considering a compressed nonlinear SA scheme with error-feedback, and establishing an analog of Theorem 1 in Theorem 2. The **significance** of this result lies in revealing that the power and scope of the popular error-feedback mechanism (Seide et al., 2014) in large-scale ML is not limited to the optimization problems it has been used for thus far. *In particular, since the nonlinear SA scheme we study captures certain instances of Q-learning (Chen et al., 2019), Theorem 2 conveys that our results extend well beyond policy evaluation to decision-making (control) problems as well.*

4. **Communication-Efficient MARL.** Since one of our main goals is to facilitate communication-efficient MARL, we consider a collaborative MARL setting that has shown up in several recent works (Doan et al., 2019; Khodadadian et al., 2022; Liu & Olshevsky, 2021a). Here, $M$ agents interact with the *same* MDP, but observe potentially different realizations of rewards and state transitions. These agents exchange compressed TD update directions via a central server to speed up the process of evaluating a specific policy. In this context, we ask the following fundamental question: *How much information needs to be transmitted to achieve the optimal linear-speedup (w.r.t. the number of agents) for policy-evaluation?* To answer this question, we propose a multi-agent version of `EF-TD`. In Theorem 3, we prove that by collaborating, each agent can achieve an $M$-fold speedup in the dominant term of the convergence rate. Importantly, the effect of compression only shows up in higher-order terms, leaving the dominant term unaffected. Thus, **we prove that by transmitting just $\tilde{O}(1)$ bits per-agent per-round, one can preserve the same asymptotic rates as vanilla `TD(0)`, while achieving an optimal linear convergence speedup w.r.t. the number of agents.** We envision this result will have important implications for MARL. Our analysis also leads to a *tighter* linear dependence on the mixing time compared to the quadratic dependence in the only other paper that establishes linear speedups under Markovian sampling (Khodadadian et al., 2022).

5. **Technical Challenges and Novel Proof Techniques.** One might wonder whether our results above follow as simple extensions of known analysis techniques in RL. In what follows, we explain why this isn't the case by outlining the key technical challenges *unique* to our setup, and our novel proof ideas to overcome them. First, a non-asymptotic analysis of even vanilla `TD(0)` under Markovian sampling is known to be extremely challenging due to complex temporal correlations. Our setting is further complicated by the fact that the dynamics of the parameter *and* the memory variable are intricately coupled with the temporally-correlated Markov data tuples. *This leads to a complex stochastic dynamical system that has not been analyzed before in RL.* Second, we cannot directly appeal to existing proofs of compression in optimization since the problems we study do not involve minimizing a static loss function. Moreover, the aspect of temporally correlated observations is completely absent in compressed optimization since one deals with i.i.d. data. The above discussion motivates the need for new analysis tools beyond what are known for both RL and optimization.

   ● **New Proof Ingredients for Theorems 1 and 2.** To prove Theorem 1, our first innovation is to construct a novel Lyapunov function that accounts for the *joint dynamics* of the parameter and the memory variable introduced by error-feedback. The next natural step is to then analyze the drift of this Lyapunov function by appealing to mixing time arguments - as is typically done in finite-time RL proofs (Bhandari et al., 2018; Srikant & Ying, 2019). This is where we again run into difficulties since the presence of the memory variable due to error-feedback introduces non-standard delay terms in the drift bound. *Notably, this difficulty does not show up when one analyzes error-feedback in optimization since there is no need for mixing time arguments of Markov chains.* To overcome this challenge, we make a connection to what might at first appear unrelated: the *Incremental Aggregated Gradient* (`IAG`) method for finite-sum optimization. Our main observation here is that the elegant analysis of the `IAG` method by Gurbuzbalaban et al. (2017) shows how one can handle the effect of "shocks" (delayed terms), as long as the amplitude and duration of the shocks is not too large. This ends up being precisely what we need for our purpose: we carefully relate the amplitude of the shocks to our Lyapunov function, and their duration to the mixing time of the underlying Markov chain. We spell out these details explicitly in Section 7.

● **New Proof Ingredients for Theorem 3.** Despite the long list of papers on MARL, the **only** one that establishes a linear speedup (w.r.t. the number of agents) in sample-complexity under Markovian sampling is the very recent paper by Khodadadian et al. (2022); all other papers end up making a restrictive i.i.d. sampling assumption (Doan et al., 2019; Liu & Olshevsky, 2021a; Shen et al., 2023) to show a speedup. The proof in (Khodadadian et al., 2022) is quite involved, and relies on Generalized Moreau Envelopes. This makes it particularly challenging to extend their proof framework to our setting where we need to additionally contend with the effects of compression and error-feedback. As such, we provide an alternate proof technique for establishing the linear speedup effect that relies crucially on a new *variance reduction lemma* under Markovian data, namely Lemma 2. This lemma is not just limited to TD learning, and may be of independent interest to the MARL literature. Lemma 2, coupled with a finer Lyapunov function (relative to that for proving Theorem 1) enable us to establish the desired linear speedup result under Markovian sampling for our setting. We elaborate on these points in Section 7.

**Scope of our Work.** We note that our work is primarily theoretical in nature, in the same spirit as several recent finite-time RL papers (Dalal et al., 2018; Bhandari et al., 2018; Srikant & Ying, 2019; Doan et al., 2019; Khodadadian et al., 2022), none of which come with any simulations. We do, however, report simulations on moderately sized (100 states) MDPs that reveal: (i) *without error-feedback (EF), compressed TD(0) can end up making little to no progress*; and (ii) with EF, the performance of compressed TD(0) complies with our theory. These simulations are in line with those known in optimization for deep learning (Stich et al., 2018; Aji & Heafield, 2017; Lin et al., 2017; Stich & Karimireddy, 2019), where one empirically observes significant benefits of performing EF. Succinctly, we convey: *compressed TD learning algorithms with EF are just as robust as their SGD counterparts, and exhibit similar convergence guarantees.*

## 1.2 Related Work

In what follows, we discuss the most relevant threads of related work.

● **Analysis of TD Learning Algorithms**. An analysis of the classical temporal difference learning algorithm (Sutton, 1988) with value function approximation was first provided by Tsitsiklis & Van Roy (1997). They employed the ODE method (Borkar & Meyn, 2000) - commonly used to study stochastic approximation algorithms - to provide an asymptotic convergence rate analysis of TD learning algorithms. Finite-time rates for such algorithms remained elusive for several years, till the work by Korda & La (2015). Soon after, Narayanan & Szepesvári (2017) noted some issues with the proofs in (Korda & La, 2015). Under an i.i.d. sampling assumption, Lakshminarayanan & Szepesvári (2017) and Dalal et al. (2018) went on to provide finite-time rates for TD learning algorithms. For the more challenging Markovian setting, finite-time rates have been recently derived using various perspectives: (i) by making explicit connections to optimization (Bhandari et al., 2018); (ii) by taking a control-theoretic approach and studying the drift of a suitable Lyapunov function (Srikant & Ying, 2019); and (iii) by arguing that the mean-path temporal difference direction acts as a "gradient-splitting" of an appropriately chosen function (Liu & Olshevsky, 2021b). Each of these interpretations provides interesting new insights into the dynamics of TD algorithms.

The above works focus on the vanilla TD learning rule. Our work adds to this literature by providing an understanding of the *robustness* of TD learning algorithms subject to structured distortions.

● **Communication-Efficient Algorithms for (Distributed) Optimization and Learning.** In the last decade or so, a variety of both scalar (Seide et al., 2014; Wen et al., 2017; Bernstein et al., 2018; Alistarh et al., 2017), and vector (Mayekar & Tyagi, 2020; Gandikota et al., 2021) quantization schemes have been explored for optimization/empirical risk minimization. In particular, an aggressive compression scheme employed popularly in deep learning is gradient sparsification, where one only transmits a few components of the gradient vector that have the largest magnitudes. While empirical studies (Aji & Heafield, 2017; Lin et al., 2017) have revealed the benefits of extreme sparsification, theoretical guarantees for such methods are much harder to establish due to their *biased* nature: the output of the compression operator is not an unbiased version of its argument. In fact, for biased schemes, naively compressing gradients can lead to diverging iterates (Karimireddy et al., 2019; Beznosikov et al., 2020). In (Stich et al., 2018; Alistarh et al., 2018), and later in (Karimireddy et al., 2019; Lin et al., 2022), it was shown that the above issue can be

"fixed" by using a mechanism known as error-feedback that exploits memory. This idea is also related to modulation techniques in coding theory (Gray, 2012). Recently, Beznosikov et al. (2020) and Gorbunov et al. (2020) provided theoretical results on biased sparsification for a master-worker type distributed architecture, and Mitra et al. (2021) studied sparsification in a federated learning context. For more recent work on the error-feedback idea, we refer the reader to (Richtárik et al., 2021; Gruntkowska et al., 2022).

Our work can be seen as the first analog of the above results in general, and the error-feedback idea in particular, in the context of iterative algorithms for RL.

Finally, we note that while several compression algorithms have been proposed and analyzed over the years, fundamental performance bounds were only recently identified by Mayekar & Tyagi (2020), Gandikota et al. (2021), and Lin et al. (2022). Computationally efficient algorithms that match such performance bounds were developed by Saha et al. (2022). While all the above results pertain to static optimization, some recent works have also explored quantization schemes in the context of sequential-decision making problems, focusing on multi-armed bandits (Hanna et al., 2022; Mitra et al., 2022; Pase et al., 2022).

## 2 Model and Problem Setup

We consider a Markov Decision Process (MDP) denoted by $\mathcal{M} = (\mathcal{S}, \mathcal{A}, \mathcal{P}, \mathcal{R}, \gamma)$, where $\mathcal{S}$ is a finite state space of size $n$, $\mathcal{A}$ is a finite action space, $\mathcal{P}$ is a set of action-dependent Markov transition kernels, $\mathcal{R}$ is a reward function, and $\gamma \in (0, 1)$ is the discount factor. For the majority of the paper, we will be interested in the *policy evaluation* problem where the goal is to evaluate the value function $V_\mu$ of a given policy $\mu$; here, $\mu : \mathcal{S} \to \mathcal{A}$. The policy $\mu$ induces a Markov reward process (MRP) characterized by a transition matrix $P$, and a reward function $R$.[1] In particular, $P(s, s')$ denotes the probability of transitioning from state $s$ to state $s'$ under the action $\mu(s)$; $R(s)$ denotes the expected instantaneous reward from an initial state $s$ under the action of the policy $\mu$. The discounted expected cumulative reward obtained by playing policy $\mu$ starting from initial state $s$ is given by:

$$V_\mu(s) = \mathbb{E}\left[\sum_{t=0}^{\infty} \gamma^t R(s_t) | s_0 = s\right], \tag{1}$$

where $s_t$ represents the state of the Markov chain (induced by $\mu$) at time $t$, when initiated from $s_0 = s$. It is well-known (Tsitsiklis & Van Roy, 1997) that $V_\mu$ is the fixed point of the policy-specific Bellman operator $\mathcal{T}_\mu : \mathbb{R}^n \to \mathbb{R}^n$, i.e., $\mathcal{T}_\mu V_\mu = V_\mu$, where for any $V \in \mathbb{R}^n$,

$$(\mathcal{T}_\mu V)(s) = R(s) + \gamma \sum_{s' \in \mathcal{S}} P(s, s') V(s'), \ \forall s \in \mathcal{S}.$$

**Linear Function Approximation.** In practice, the size of the state space $\mathcal{S}$ can be extremely large. This renders the task of estimating $V_\mu$ *exactly* (based on observations of rewards and state transitions) intractable. One common approach to tackle this difficulty is to consider a parametric approximation $\hat{V}_\theta$ of $V_\mu$ in the linear subspace spanned by a set $\{\phi_k\}_{k \in [K]}$ of $K \ll n$ basis vectors, where $\phi_k = [\phi_k(1), \ldots, \phi_k(n)]^\top \in \mathbb{R}^n$. Specifically, we have $\hat{V}_\theta(s) = \sum_{k=1}^{K} \theta(k) \phi_k(s)$, where $\theta = [\theta(1), \ldots, \theta(K)]^\top \in \mathbb{R}^K$ is a weight vector. Let $\Phi \in \mathbb{R}^{n \times K}$ be a matrix with $\phi_k$ as its $k$-th column; we will denote the $s$-th row of $\Phi$ by $\phi(s) \in \mathbb{R}^K$, and refer to it as the feature vector corresponding to state $s$. Compactly, $\hat{V}_\theta = \Phi\theta$, and for each $s \in \mathcal{S}$, we have that $\hat{V}_\theta(s) = \langle \phi(s), \theta \rangle$. As is standard, we assume that the columns of $\Phi$ are linearly independent, and that the feature vectors are normalized, i.e., for each $s \in \mathcal{S}$, $\|\phi(s)\|_2^2 \leq 1$.

**The TD(0) Algorithm.** The goal now is to find the best parameter vector $\theta^*$ that minimizes the distance (in a suitable norm) between $\hat{V}_\theta$ and $V_\mu$. To that end, we will focus on the classical TD(0) algorithm within the family of TD learning algorithms. Starting from an initial estimate $\theta_0$, at each time-step $t = 0, 1, \ldots$, this algorithm receives as observation a data tuple $X_t = (s_t, s_{t+1}, r_t = R(s_t))$ comprising of the current state, the next state reached by playing action $\mu(s_t)$, and the instantaneous reward $r_t$. Given this tuple $X_t$, let us

---

[1]For simplicity of notation, we have dropped the dependence of $P$ and $R$ on the policy $\mu$.

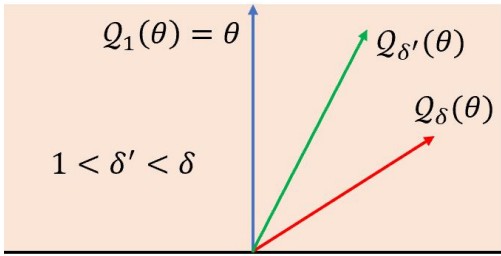

Figure 1: A geometric interpretation of the operator $\mathcal{Q}_\delta(\cdot)$ in Algorithm 1. A larger $\delta$ induces more distortion.

---

**Algorithm 1** The `EF-TD` Algorithm

---

1: **Input:** Initial estimate $\theta_0 \in \mathbb{R}^K$, initial error $e_{-1} = 0$, and step-size $\alpha \in (0, 1)$.
2: **for** $t = 0, 1, \dots$ **do**
3:    Observe tuple $X_t = (s_t, s_{t+1}, r_t)$.
4:    Compute perturbed `TD(0)` direction $h_t$:

$$h_t = \mathcal{Q}_\delta \left( e_{t-1} + g_t(\theta_t) \right). \tag{3}$$

5:    Update parameter: $\theta_{t+1} = \theta_t + \alpha h_t$.
6:    Update error: $e_t = e_{t-1} + g_t(\theta_t) - h_t$.
7: **end for**

---

define $g_t(\theta) = g(X_t, \theta)$ as:

$$g_t(\theta) \triangleq (r_t + \gamma \langle \phi(s_{t+1}), \theta \rangle - \langle \phi(s_t), \theta \rangle) \phi(s_t), \forall \theta \in \mathbb{R}^K.$$

The `TD(0)` update to the current parameter $\theta_t$ with step-size $\alpha_t \in (0, 1)$ can now be described succinctly as:

$$\theta_{t+1} = \theta_t + \alpha_t g_t(\theta_t). \tag{2}$$

Under some mild technical conditions, it was shown in (Tsitsiklis & Van Roy, 1997) that the iterates generated by `TD(0)` converge almost surely to the best linear approximator in the span of $\{\phi_k\}_{k \in [K]}$. In particular, $\theta_t \to \theta^*$, where $\theta^*$ is the unique solution of the projected Bellman equation $\Pi_D \mathcal{T}_\mu(\Phi \theta^*) = \Phi \theta^*$. Here, $D$ is a diagonal matrix with entries given by the elements of the stationary distribution $\pi$ of the kernel $P$. Moreover, $\Pi_D(\cdot)$ is the projection operator onto the subspace spanned by $\{\phi_k\}_{k \in [K]}$ with respect to the inner product $\langle \cdot, \cdot \rangle_D$. The key question we explore in this paper is: *What can be said of the convergence of TD(0) when the update direction $g_t(\theta_t)$ in equation 2 is replaced by a distorted/compressed version of it?* In the next section, we will make the notion of distortion precise, and outline the various technical challenges that make it non-trivial to answer the above question.

## 3   Error-Feedback meets TD Learning

In this section, we propose a general framework for analyzing TD learning algorithms with distorted update directions. Extreme forms of such distortion could involve replacing each coordinate of the `TD(0)` direction with just its sign, or retaining just $k \in [K]$ of the largest magnitude coordinates and zeroing out the rest. In the sequel, we will refer to these variants of `TD(0)` as `SignTD(0)` and `Topk-TD(0)`, respectively. Coupled with a memory mechanism known as error-feedback, it is known that analogous variants of `SGD` exhibit, almost remarkably, the same behavior as `SGD` itself (Stich et al., 2018). Inspired by these findings, we ask: *Does compressed TD(0) with error-feedback exhibit behavior similar to that of TD(0)?*

Our motivation behind studying the above question is to build an understanding of: (i) communication-efficient versions of MARL algorithms where agents exchange compressed model-differentials, i.e., the gradient-like updates (see Section 6); and (ii) the *robustness* of TD learning algorithms to *structured perturbations*. It is

important to reiterate here that our motivation mirrors that of studying perturbed versions of `SGD` in the context of optimization.

**Description of `EF-TD`.** In Algorithm 1, we propose the compressed `TD(0)` algorithm with error-feedback (`EF-TD`). Compared to equation 2, we note that the parameter $\theta_t$ is updated based on $h_t$ - a compressed version of the actual `TD(0)` direction $g_t(\theta_t)$, where the compression is due to the operator $\mathcal{Q}_\delta : \mathbb{R}^K \to \mathbb{R}^K$. The information lost up to time $t-1$ due to compression is accumulated in the memory variable $e_{t-1}$, and injected back into the current step as in equation 3. In line with compressors used for optimization, we consider a fairly general operator $\mathcal{Q}_\delta$ that is required to only satisfy the following contraction property for some $\delta \geq 1$:

$$\|\mathcal{Q}_\delta(\theta) - \theta\|_2^2 \leq \left(1 - \frac{1}{\delta}\right)\|\theta\|_2^2, \forall \theta \in \mathbb{R}^K. \tag{4}$$

**A distortion perspective.** For $\theta \neq 0$, it is easy to verify based on equation 4 that $\langle \mathcal{Q}_\delta(\theta), \theta \rangle \geq 1/(2\delta)\|\theta\|_2^2 > 0$, i.e., the angle between $\mathcal{Q}_\delta(\theta)$ and $\theta$ is acute. This provides an alternative view to the compression perspective: one can think of $\mathcal{Q}_\delta(\theta)$ as a tilted version of $\theta$, with a larger $\delta$ implying more tilt, and $\delta = 1$ implying no distortion at all. See Figure 1 for a visual interpretation of this point.

The contraction property in equation 4 is satisfied by several popular quantization/sparsification schemes. These include the sign operator, and the Top-$k$ operator that selects $k$ coordinates with the largest magnitude, zeroing out the rest. Importantly, note that the operator $\mathcal{Q}_\delta$ does not necessarily yield an unbiased version of its argument. In optimization, error-feedback serves to counter the bias introduced by $\mathcal{Q}_\delta$. In fact, without error-feedback, algorithms like `SignSGD` can converge very slowly, or not converge at all (Karimireddy et al., 2019). In a similar spirit, we empirically observe in Fig. 2 (later in Section 8) that without error-feedback, `SignTD(0)` can end up making little to no progress towards $\theta^*$. However, understanding whether error-feedback guarantees convergence to $\theta^*$ is quite non-trivial. We now provide some intuition as to why this is the case.

**Need for Technical Novelty.** Error-feedback ensures that past (pseudo)-gradient information is injected with some delay. Thus, for optimization, error-feedback essentially leads to a delayed SGD algorithm. As long as the objective function is smooth, the gradient does not change much, and the delay-effect can be controlled. *This intuition does not carry over to our setting since the TD(0) update direction is not a stochastic gradient of any fixed objective function.*[2] Thus, controlling the effect of the compressor-induced delay requires new techniques for our setting. Moreover, unlike the SGD noise, the data tuples $\{X_t\}$ are part of the same Markov trajectory, introducing further complications. Finally, since the parameter updates of Algorithm 1 are intricately linked with the error variable $e_t$, to analyze their joint behavior, we need to perform a more refined Lyapunov drift analysis relative to the standard `TD(0)` analysis. Despite these challenges, in the sequel, we will establish that `EF-TD` retains almost the same convergence guarantees as vanilla `TD(0)`.

**Remark 1.** *It is important to emphasize that even though the error vector $e_t$ in `EF-TD` might be dense, $e_t$ never gets transmitted. The communication-efficient aspect of `EF-TD` stems from the fact that it only requires communicating the compressed update direction $h_t$ - a vector that can be represented by using just a few bits (depending on the level of compression). This fact is further clarified in our description of the multi-agent version of `EF-TD` in Section 6 (see line 6 of Algorithm 2 in Section 6).*

## 4 Analysis of `EF-TD` under Markovian Sampling

The goal of this section is to provide a rigorous finite-time analysis of `EF-TD` under Markovian sampling. To that end, we need to introduce a few concepts and make certain standard assumptions. We start by assuming that all rewards are uniformly bounded by some $\bar{r} > 0$, i.e., $|R(s)| \leq \bar{r}, \forall s \in \mathcal{S}$. This ensures that the value functions exist. Next, we state a standard assumption that shows up in the finite-time analysis of iterative RL algorithms (Bhandari et al., 2018; Srikant & Ying, 2019; Chen et al., 2019; Patil et al., 2023; Doan et al., 2019; Khodadadian et al., 2022).

**Assumption 1.** *The Markov chain induced by the policy $\mu$ is aperiodic and irreducible.*

---

[2]As observed by Bhandari et al. (2018) and Liu & Olshevsky (2021b), this can be seen from the fact that the derivative of the TD update direction produces a matrix that is not necessarily symmetric, unlike the symmetric Hessian matrix of a fixed objective function.

The above assumption implies that the Markov chain induced by $\mu$ admits a unique stationary distribution $\pi$ (Levin & Peres, 2017). Let $\Sigma = \Phi^\top D \Phi$. Since $\Phi$ is full column rank, $\Sigma$ is full rank with a strictly positive smallest eigenvalue $\omega < 1$. To appreciate the implication of the above assumption, let us define the "steady-state" version of the TD update direction as follows: for a fixed $\theta \in \mathbb{R}^K$, let

$$\bar{g}(\theta) \triangleq \mathbb{E}_{s_t \sim \pi, s_{t+1} \sim P_\mu(\cdot|s_t)} \left[ g(X_t, \theta) \right].$$

We now introduce the notion of the mixing time $\tau_\epsilon$.

**Definition 1.** *Define* $\tau_\epsilon \triangleq \min\{t \geq 1 : \|\mathbb{E}\left[g(X_k, \theta)|X_0\right] - \bar{g}(\theta)\|_2 \leq \epsilon \left(\|\theta\|_2 + 1\right), \forall k \geq t, \forall \theta \in \mathbb{R}^K, \forall X_0\}$.

Assumption 1 implies that the total variation distance between the conditional distribution $\mathbb{P}\left(s_t = \cdot|s_0 = s\right)$ and the stationary distribution $\pi$ decays geometrically fast for all $t \geq 0$, regardless of the initial state $s \in \mathcal{S}$ (Levin & Peres, 2017). As a consequence of this geometric mixing of the Markov chain, it is not hard to show that $\tau_\epsilon$ in Definition 1 is $O\left(\log(1/\epsilon)\right)$; see, for instance, Chen et al. (2019). The precision that suffices for all results in this paper is $\epsilon = \alpha^2$, where recall that $\alpha$ is the step-size. Henceforth, we will simply use $\tau$ as a shorthand for $\tau_\alpha$. Finally, let us define $\sigma \triangleq \max\{1, \bar{r}, \|\theta^*\|_2\}$ as the variance of our noise model, and $d_t \triangleq \|\theta_t - \theta^*\|_2$. We can now state the first main result of this paper.

**Theorem 1.** *Suppose Assumption 1 holds. There exist universal constants* $c, C \geq 1$ *such that the iterates generated by* `EF-TD` *with step-size* $\alpha \leq \frac{\omega(1-\gamma)}{c \max\{\delta, \tau\}}$ *satisfy the following* $\forall T \geq \tau$:

$$\mathbb{E}\left[d_T^2\right] \leq C_1 \left(1 - \frac{\alpha\omega(1-\gamma)}{C\tau}\right)^{T-\tau} + O\left(\frac{\alpha(\tau+\delta)\sigma^2}{\omega(1-\gamma)}\right), \text{ where } C_1 = O(d_0^2 + \sigma^2). \quad (5)$$

The proof of Theorem 1 is provided in Appendix D. We now discuss the key implications of this result.

**Discussion.** Theorem 1 tells us that `EF-TD` guarantees linear convergence of the iterates to a ball around $\theta^*$, where the size of the ball scales with the variance $\sigma^2$; this exactly matches the behavior of vanilla TD (Bhandari et al., 2018; Srikant & Ying, 2019). Since the step-size $\alpha$ scales inversely with the distortion factor $\delta$, we observe from equation 5 that the exponent of linear convergence gets slackened by $\delta$; once again, this is consistent with analogous results for SGD with (biased) compression (Beznosikov et al., 2020). The variance term, namely the second term in equation 5, has the *exact same dependence on* $\tau, \omega,$ *and* $\gamma$ *as one observes for vanilla TD (Bhandari et al., 2018, Theorem 3).* Observe that in this term, the effects of $\tau$ and $\delta$ in inflating the variance are additive. Moreover, even in the absence of compression, the dependence of the variance term on the mixing time $\tau$ is known to be *unavoidable* (Nagaraj et al., 2020). *This immediately leads to the following interesting conclusion: when the underlying Markov chain induced by the policy mixes "slowly", i.e., has a large mixing time* $\tau$, *one can afford to be more aggressive in terms of compression, i.e., use a larger* $\delta$, *since this would lead to a variance bound that is no worse than in the uncompressed setting.* Said differently, Theorem 1 reveals that slowly-mixing Markov chains have a higher tolerance to distortions. This observation is novel since no prior work has studied the effect of distortions to RL algorithms under Markovian sampling. It is also interesting to note here that the phenomenon described above shows up in other contexts too: for instance, the authors in Gandikota et al. (2021) showed that certain quantization mechanisms in optimization automatically come with privacy guarantees.

**Theorem 1 is significant in that it is the first result to reveal that coupled with error-feedback, `EF-TD` is robust to extreme distortions.** We will corroborate this phenomenon empirically as well in Section 8. The other key takeaway from Theorem 1 is that the scope of the error-feedback mechanism - now popularly used in distributed learning - extends to stochastic approximation problems well beyond just static optimization settings.

In Appendix B, we analyze a simpler steady-state version of `EF-TD` to help build intuition. There, we also provide an analysis of compressed TD learning algorithms that do not employ any error-feedback mechanism. Our analysis reveals that such schemes can also converge, provided the compression parameter $\delta$ satisfies a restrictive condition. Notably, as evident from the statement of Theorem 1, we do not need to impose any restrictions on $\delta$ for the convergence of `EF-TD`.

# 5    Compressed Nonlinear Stochastic Approximation with Error-Feedback

For the TD(0) algorithm, the update direction $g_t(\theta)$ is affine in the parameter $\theta$, i.e., $g_t(\theta)$ is of the form $A(X_t)\theta - b(X_t)$. As such, the recursion in equation 2 can be seen as an instance of linear stochastic approximation (SA), where the end goal is to use the data samples $\{X_t\}$ to find a $\theta$ that solves the linear equation $A\theta = b$; here, $A$ and $b$ are the steady-state versions of $A(X_t)$ and $b(X_t)$, respectively. The more involved TD($\lambda$) algorithms within the TD learning family also turn out to be instances of linear SA. Instead of deriving versions of our results for TD($\lambda$) algorithms in particular, we will instead consider a much more general nonlinear SA setting. Accordingly, we now study a variant of Algorithm 1 where $g(X_t, \theta)$ is a general nonlinear map, and as before, $X_t$ comes from a finite-state Markov chain that is assumed to be aperiodic and irreducible. We assume that the nonlinear map satisfies the following regularity conditions.

**Assumption 2.** *There exist $L, \sigma \geq 1$ s.t. the following hold for any $X$ in the space of data tuples: (i) $\|g(X, \theta_1) - g(X, \theta_2)\|_2 \leq L\|\theta_1 - \theta_2\|_2, \forall \theta_1, \theta_2 \in \mathbb{R}^K$, and (ii) $\|g(X, \theta)\|_2 \leq L(\|\theta\|_2 + \sigma), \forall \theta \in \mathbb{R}^K$.*

**Assumption 3.** *Let $\bar{g}(\theta) \triangleq \mathbb{E}_{X_t \sim \pi}[g(X_t, \theta)], \forall \theta \in \mathbb{R}^K$, where $\pi$ is the stationary distribution of the Markov process $\{X_t\}$. The equation $\bar{g}(\theta) = 0$ has a solution $\theta^*$, and $\exists \beta > 0$ s.t.*

$$\langle \theta - \theta^*, \bar{g}(\theta) - \bar{g}(\theta^*) \rangle \leq -\beta \|\theta - \theta^*\|_2^2, \forall \theta \in \mathbb{R}^K. \tag{6}$$

In words, Assumption 2 says that $g(X, \theta)$ is globally uniformly (w.r.t. $X$) Lipschitz in the parameter $\theta$. Assumption 3 is a strong monotone property of the map $-\bar{g}(\theta)$ that guarantees that the iterates generated by the steady-state version of equation 2 converge exponentially fast to $\theta^*$. To provide some context, consider the TD(0) setting. Under feature normalization and the bounded rewards assumption, the global Lipschitz property is immediate. Moreover, Assumption 3 corresponds to negative-definiteness of the steady-state matrix $A = \mathbb{E}_{X_t \sim \pi}[A(X_t)]$; this negative-definite property is also easy to verify (Srikant & Ying, 2019). For optimization, Assumptions 2 and 3 simply correspond to $L$-smoothness and $\beta$-strong-convexity of the loss function, respectively. For simplicity, we state the main result of this section for $L = 2$; the analysis for a general $L$ follows identical arguments.

**Theorem 2.** *Suppose Assumption 2 holds with $L = 2$, and Assumption 3 holds. Let $\bar{\beta} = \min\{\beta, 1/\beta\}$. There exist universal constants $c, C \geq 1$ such that Algorithm 1 with step-size $\alpha \leq \frac{\bar{\beta}}{c\max\{\delta,\tau\}}$ guarantees*

$$\mathbb{E}\left[d_T^2\right] \leq C_1 \left(1 - \frac{\alpha\beta}{C\tau}\right)^{T-\tau} + O\left(\frac{\alpha(\tau + \delta)\sigma^2}{\beta}\right), \forall T \geq \tau, \text{ where } C_1 = O(d_0^2 + \sigma^2). \tag{7}$$

**Discussion.** We note that the guarantee in Theorem 2 mirrors that in Theorem 1, and represents our setting in its full generality, accounting for nonlinear SA, Markovian noise, compression, and error-feedback. Providing an explicit finite-time analysis for this setting is one of the main contributions of our paper. Now let us comment on the applications of this result. As noted earlier, our result applies to TD($\lambda$) algorithms and SGD under Markovian noise (Doan, 2022); the effect of compression and error-feedback was previously unknown for both these settings. More importantly, certain instances of Q-learning with linear function approximation can also be captured via Assumptions 2 and 3 (Chen et al., 2019). *The key implication is that our analysis framework is not just limited to policy evaluation, but rather extends gracefully to decision-making (control) problems.* This speaks to the significance of Theorem 2.

# 6    Communication-Efficient Multi-Agent Reinforcement Learning (MARL)

A key motivation of our work is to develop communication-efficient algorithms for MARL. This is particularly relevant for networked/federated versions of RL problems where communication imposes a major bottleneck (Qi et al., 2021). To that end, we consider a collaborative MARL setting that has appeared in various recent works (Qi et al., 2021; Doan et al., 2019; Liu & Olshevsky, 2021a; Jin et al., 2022; Khodadadian et al., 2022; Shen et al., 2023). The setup comprises of $M$ agents, all of whom interact with the *same* MDP, and can communicate via a central server. Every agent seeks to evaluate the *same* policy $\mu$. The purpose of collaboration is as in the standard FL setting: to achieve a $M$-fold reduction in sample-complexity by leveraging information from all agents. In particular, we ask: *By exchanging compressed information via the*

---

**Algorithm 2** Multi-Agent `EF-TD`

---

1: **Input:** Initial estimate $\theta_0 \in \mathbb{R}^K$, initial errors $e_{i,-1} = 0, \forall i \in [M]$, and step-size $\alpha \in (0,1)$.
2: **for** $t = 0, 1, \ldots$ **do**
3:     Server sends $\theta_t$ to all agents.
4:     **for** $i \in [M]$ **do**
5:         Observe tuple $X_{i,t} = (s_{i,t}, s_{i,t+1}, r_{i,t})$.
6:         Compute compressed `TD`(0) direction $h_{i,t} = \mathcal{Q}_\delta\left(e_{i,t-1} + g_{i,t}(\theta_t)\right)$, and send $h_{i,t}$ to server.
7:         Update error: $e_{i,t} = e_{i,t-1} + g_{i,t}(\theta_t) - h_{i,t}$.
8:     **end for**
9:     Server updates the model as follows: $\theta_{t+1} = \theta_t + \alpha \bar{h}_t$, where $h_t = (1/M)\sum_{i \in [M]} h_{i,t}$.
10: **end for**

---

*server, is it possible to expedite the process of learning by achieving a linear speedup in the number of agents?* While such questions have been extensively studied for supervised learning, we are unaware of any work that addresses them in the context of MARL. We further note that even without compression or error-feedback, establishing linear speedups under Markovian sampling is highly non-trivial, and the only other paper that does so is the very recent work of Khodadadian et al. (2022). As we explain later in Section 7, our proof technique departs significantly from (Khodadadian et al., 2022).

We propose and analyze a natural multi-agent version of `EF-TD`, outlined as Algorithm 2. In a nutshell, multi-agent `EF-TD` operates as follows. At each time-step, the server sends down a model $\theta_t$; each agent $i$ observes a local data sample, computes and uploads the compressed direction $h_{i,t}$, and updates its local error. The server then updates the model. Transmitting compressed `TD`(0) pseudo-gradients is consistent with both works in FL (Karimireddy et al., 2020), and in MARL (Qi et al., 2021; Doan et al., 2019; Khodadadian et al., 2022), where the agents essentially exchange model-differentials (i.e., the update directions), keeping their raw data private. It is worth noting here that while all agents play the same policy, the realizations of the data tuples $\{X_{i,t}\}$ may vary across agents. Let $\tilde{\theta}_t = \theta_t + \alpha \bar{e}_{t-1}$, where $\bar{e}_t = (1/M)\sum_{i \in [M]} e_{i,t}$, and define $\tilde{d}_t \triangleq \|\tilde{\theta}_t - \theta^*\|_2$. We can now state our main convergence result for Algorithm 2.

**Theorem 3.** *Suppose Assumption 1 holds. There exist universal constants $c, C \geq 1$ such that with step-size $\alpha \leq \frac{\omega(1-\gamma)}{c\max\{\delta,\tau\}}$, and $C_1 = O(d_0^2 + \sigma^2)$, Algorithm 2 guarantees the following $\forall T \geq 2\tau$:*

$$\mathbb{E}\left[\tilde{d}_T^2\right] \leq \underbrace{C_1\left(1 - \frac{\alpha\omega(1-\gamma)}{C\tau}\right)^{T-2\tau}}_{T_1} + \underbrace{O\left(\frac{\alpha\tau}{\omega(1-\gamma)}\right)\frac{\sigma^2}{M}}_{T_2} + \underbrace{O\left(\frac{\alpha^2\max\{\delta,\tau\}\delta}{\omega^2(1-\gamma)^2}\right)\sigma^2}_{T_3}. \tag{8}$$

The proof of Theorem 3 is deferred to Appendix E. There are several key messages conveyed by Theorem 3. We discuss them below.

**Message 1: Linear Speedup.** Observe that the noise variance terms in equation 8 are $T_2$ and $T_3$, where $T_3$ is $O(\alpha^2)$, i.e., a higher-order term in $\alpha$. Thus, for small $\alpha$, $T_2$ is the dominant noise term. Compared to the noise term for vanilla TD in (Bhandari et al., 2018, Theorem 3), $T_2$ in our bound has exactly the same dependence on $\tau$, $\omega$, and $\gamma$, and importantly, exhibits an inverse scaling w.r.t. $M$, *implying a $M$-fold reduction in the variance $\sigma^2$ relative to the single agent setting.* This is precisely what we wanted. For exact convergence, we can set $\alpha = O(\log(MT^2)/T)$ to make $T_1$ and $T_3$ of order $\tilde{O}(1/T^2)$, and the dominant term $T_2 = \tilde{O}(\sigma^2/(MT))$. Thus, relative to the $O(\sigma^2/T)$ rate of `TD`(0), *we achieve a linear speedup w.r.t. the number of agents $M$ in our dominant term $T_2$.*

**Message 2: Communication-efficiency comes (nearly) for free.** The second key thing to note is that the dominant $T_2$ term is *completely unaffected by the compression factor $\delta$.* Indeed, the compression factor $\delta$ only affects higher-order terms that decay much faster than $T_2$. *This means that we can significantly ramp up $\delta$, thereby communicating very little, while preserving the asymptotic rate of convergence of `TD(0)` and achieving optimal speedup in the number of agents, i.e., communication-efficiency comes almost for free.* For instance, suppose $\mathcal{Q}_\delta(\cdot)$ is a top-1 operator, i.e., $h_{i,t}$ has only one non-zero entry which can be encoded using

$\tilde{O}(1)$ bits. In this case, although $\delta = K$, where $K$ is the potentially large number of features, the dominant $\tilde{O}\left(\sigma^2/(MT)\right)$ term remains unaffected by $K$. **Thus, in the context of MARL, our work is the first to show that asymptotically optimal rates can be achieved by transmitting just $\tilde{O}(1)$ bits per-agent per time-step**.

**Message 3: Tight Dependence on the Mixing Time.** Compared to Khodadadian et al. (2022) - the only other paper in MARL that provides a linear speedup under Markovian sampling - our dominant term $T_2$ has a tighter $O(\tau)$ dependence on the mixing time $\tau$ as compared to the $O(\tau^2)$ dependence in Khodadadian et al. (2022, Theorem 4.1). It should be noted here that the $O(\tau)$ dependence is known to be information-theoretically optimal (Nagaraj et al., 2020).

A few remarks are in order before we end this section.

**Remark 2.** *We note that the bound in Theorem 3 is stated for $\tilde{\theta}_T$, not $\theta_T$. Since $\tilde{\theta}_T = \theta_T + \alpha\bar{e}_{T-1}$, to output $\tilde{\theta}_T$, the server needs to query $e_{i,T-1}$ from each agent $i$ only once at time $T-1$. We believe that this one extra step of communication can also be avoided via a slightly sharper analysis. We provide such an analysis in Appendix F, albeit under a common i.i.d. sampling model (Dalal et al., 2018; Doan et al., 2019; Liu & Olshevsky, 2021a).*

**Remark 3.** *While our MARL result in Theorem 3 is for TD learning, in light of the developments in Section 5, we note that one can develop analogs of Theorem 3 for multi-agent Q-learning as well.*

**Remark 4.** *Suppose we set $M = 1$ in equation 8, and compare the resulting bound with that in equation 5. While the effect of the distortion $\delta$ shows up as a higher-order $O(\alpha^2)$ term in the former, it manifests as a $O(\alpha)$ term in the latter. The difference in these bounds can be attributed to the fact that we use a finer Lyapunov function to prove Theorem 3 relative to that which we use to prove Theorem 1. We do so primarily for the sake of exposition: the relatively simpler proof of Theorem 1 (compared to Theorem 3) helps build much of the key intuition needed to understand the proof of Theorem 3.*

## 7 Technical Challenges in Analysis and Overview of our Proof Techniques

We now discuss the novel steps in our analysis. Complete proof details are provided in the Appendix.

**Proof Sketch for Theorem 1.** Inspired by the perturbed iterate framework of Mania et al. (2015), our first step is to define the perturbed iterate $\tilde{\theta}_t = \theta_t + \alpha e_{t-1}$. Simple calculations reveal: $\tilde{\theta}_{t+1} = \tilde{\theta}_t + \alpha g_t(\theta_t)$. Notice that this recursion is almost the same as equation 2, other than the fact that the TD(0) update direction is evaluated at $\theta_t$ instead of $\tilde{\theta}_t$. Thus, to analyze the above recursion, we need to account for the gap between $\theta_t$ and $\tilde{\theta}_t$, the cause of which is the memory variable $e_{t-1}$. Accordingly, our next key step is to construct a novel Lyapunov function $\psi_t$ that captures the *joint dynamics* of $\tilde{\theta}_t$ and $e_{t-1}$:

$$\psi_t \triangleq \mathbb{E}[\tilde{d}_t^2 + \alpha^2\|e_{t-1}\|_2^2], \text{ where } \tilde{d}_t^2 = \|\tilde{\theta}_t - \theta^*\|_2^2. \tag{9}$$

*As far as we are aware, a potential function of the above form has not been analyzed in prior RL work.* Our goal now is to exploit the geometric mixing property in Assumption 1 to establish that $\psi_t$ decays over time (up to noise terms). This is precisely where the intricate coupling between the parameter $\tilde{\theta}_t$, the memory variable $e_t$, and the Markovian data tuples $\{X_t\}$ makes the analysis quite challenging. Let us elaborate. To exploit mixing-time arguments, we need to condition sufficiently into the past and bound drift terms of the form $\|\tilde{\theta}_t - \tilde{\theta}_{t-\tau}\|_2$. This is where the coupling between $\tilde{\theta}_t$ and $e_t$ introduces non-standard delay terms, precluding the direct use of prior approaches in RL (Bhandari et al., 2018; Srikant & Ying, 2019; Chen et al., 2019). This difficulty does not show up in compressed optimization since one deals with i.i.d. data, precluding the need for mixing-time arguments. A workaround here is to employ a projection step (as in Bhandari et al. (2018); Doan et al. (2019)) to simplify the analysis. This is not satisfying for two reasons: (i) as we show in the Appendix, the simplification in the analysis comes at the cost of a sub-optimal dependence on the distortion $\delta$; and (ii) to project, one needs prior knowledge of the set containing $\theta^*$; this turns out to be unnecessary. Our key innovation here to bound the drift $\|\tilde{\theta}_t - \tilde{\theta}_{t-\tau}\|$ is the following technical lemma.

**Lemma 1.** *(**Relating Drift to Past Shocks**) For* `EF-TD`, *the following is true $\forall t \geq \tau$:*

$$\mathbb{E}[\|\tilde{\theta}_t - \tilde{\theta}_{t-\tau}\|_2^2] = O(\alpha^2\tau^2)\max_{t-\tau\leq\ell\leq t}\psi_\ell + O(\alpha^2\tau^2\sigma^2). \tag{10}$$

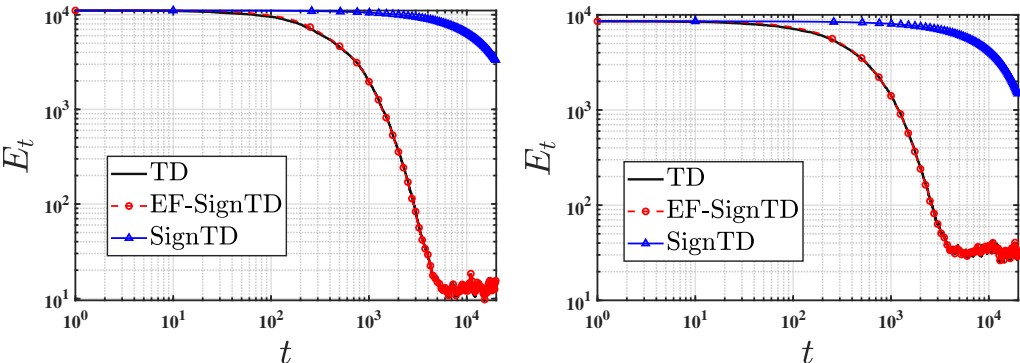

Figure 2: Plots of the mean-squared error $E_t = \|\theta_T - \theta^*\|_2^2$ for vanilla `TD(0)` without compression, and `SignTD(0)` with (`EF-SignTD`) and without (`SignTD`) error-feedback. (**Left**) Discount factor $\gamma = 0.5$. (**Right**) Discount factor $\gamma = 0.9$.

The above lemma relates the drift to *damped* "shocks" (delay terms) from the past, where (i) the amplitude of the shocks is captured by our Lyapunov function; (ii) the duration of these shocks is the mixing time $\tau$; and (iii) the damping arises from the fact that shocks are scaled by $\alpha^2$. In attempting to overcome one difficulty, we have created another for ourselves via the "max" term in equation 10. This is where we make a connection to the analysis of the `IAG` algorithm in Gurbuzbalaban et al. (2017) where a similar "max" term shows up. Via this connection, we are able to analyze the following final recursion:

$$\psi_{t+1} \leq (1 - c\alpha\omega(1-\gamma))\psi_t + O(\alpha^2\tau) \max_{t-\tau \leq \ell \leq t} \psi_\ell + O(\alpha^2)(\tau + \delta)\sigma^2, \text{ where } c < 1. \tag{11}$$

**Proof Sketch for Theorem 3.** To establish the linear speedup property, we need a finer Lyapunov function (and much finer analysis) relative to Theorem 1. Accordingly, we construct:

$$\Xi_t \triangleq \mathbb{E}\left[\|\tilde{\theta}_t - \theta^*\|_2^2\right] + C\alpha^3 E_{t-1}, \text{ where } E_t \triangleq \frac{1}{M}\mathbb{E}\left[\sum_{i=1}^{M} \|e_{i,t}\|_2^2\right], \tag{12}$$

and $C$ is a suitable design parameter. Using the techniques in Bhandari et al. (2018), Srikant & Ying (2019), and Chen et al. (2019) to bound $\|g_{i,t}(\theta)\|^2$ unfortunately do not yield the desired linear speedup. Moreover, it is unclear whether the Generalized Moreau Envelope framework in Khodadadian et al. (2022) can be extended to analyze equation 12. As such, we depart from these works by establishing the following key result by carefully exploiting the geometric mixing property.

**Lemma 2.** *Let $z_t(\theta_t) \triangleq (1/M)\sum_{i\in[M]} g_{i,t}(\theta_t)$. Under Assumption 1, the following bound holds for Algorithm 2: $\mathbb{E}[\|z_t(\theta_t)\|_2^2] = O(1)\mathbb{E}[\tilde{d}_t^2] + O(\alpha^2)E_{t-1} + O(1/M + \alpha^4)\sigma^2, \forall t \geq \tau$.*

The above norm bound on the average TD direction turns out to be the main ingredient in bounding the drift terms for Algorithm 2, and eventually establishing a recursion of the form in equation 11 for $\Xi_t$.

## 8 Simulations

To corroborate our theory, we construct a MDP with 100 states, and use a feature matrix $\Phi$ with $K = 10$ independent basis vectors. Using this MDP, we generate the state transition matrix $P_\mu$ and reward vector $R_\mu$ associated with a fixed policy $\mu$. We compare the performance of the vanilla uncompressed `TD(0)` algorithm with linear function approximation to `SignTD(0)` with and without error-feedback. We perform 30 independent trials, and average the errors from each of these trials to report the mean-squared error $E_t = \|\theta_T - \theta^*\|_2^2$ for each of the algorithms mentioned above. The results of this experiment are reported in Fig. 2. We observe that in the absence of error-feedback, `SignTD(0)` can make little to no progress towards $\theta^*$. In contrast, the behavior of `SignTD(0)` with error-feedback almost exactly matches that of vanilla uncompressed `TD(0)`. Our

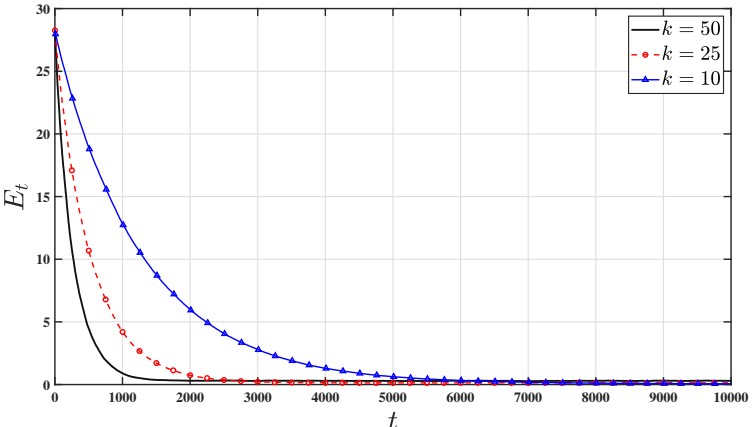

Figure 3: Plot of the mean-squared error $E_t = \|\theta_T - \theta^*\|_2^2$ for `EF-TD` (Algo. 1), with $\mathcal{Q}_\delta(\cdot)$ chosen to be the top-$k$ operator. We study the effect of varying the number of components transmitted $k$.

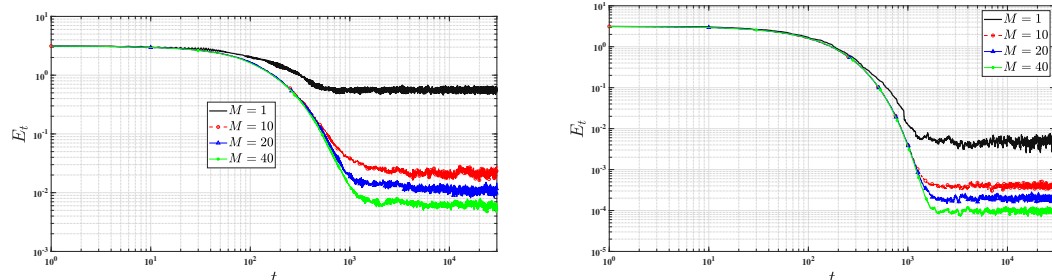

Figure 4: Plots of the mean-squared error $E_t = \|\theta_T - \theta^*\|_2^2$ for multi-agent `EF-TD` (Algorithm 2). (**Left**) $\mathcal{Q}_\delta(\cdot)$ is the sign operator. (**Right**) $\mathcal{Q}_\delta(\cdot)$ is the Top-$k$ operator with $k = 2$.

results align with similar observations made in the context of optimization, where `SignSGD` with error-feedback retains almost the same behavior as `SGD` with no compression (Stich et al., 2018; Karimireddy et al., 2019). We provide some additional experiments below.

• **Simulation of `Topk-TD(0)` with Error-Feedback**. We simulate the behavior of `EF-TD` with the operator $\mathcal{Q}_\delta(\cdot)$ chosen to be a top-$k$ operator. We consider the same MDP as above, with the rewards for this experiment chosen from the interval $\begin{bmatrix} 0 & 1 \end{bmatrix}$. The size of the parameter vector $K$ is 50, and the discount factor $\gamma$ is set to be 0.5. We vary the level of distortion introduced by $\mathcal{Q}_\delta(\cdot)$ by changing the number of components $k$ transmitted. Note that $\delta = K/k$. As one might expect, increasing the distortion $\delta$ by transmitting fewer components impacts the exponential decay rate; this is clearly reflected in Fig. 3.

**Simulation of Multi-Agent `EF-TD`**. To simulate the behavior of multi-agent `EF-TD` (Algorithm 2), we consider the same MDP on 100 states as before with rewards in [0 1]. The dimension $K$ of the parameter vector is set to 10 and the discount factor $\gamma$ to 0.3. We consider two cases: one where $\mathcal{Q}_\delta(\cdot)$ is the sign operator, and another where it is a top-2 operator, i.e., only two components are transmitted by each agent at every time-step. We report our findings in Fig. 4, where we vary the number of agents $M$, and observe that for both the sign- and top-$k$ operator experiments, the residual mean-squared error goes down as we scale up $M$. Since the residual error is essentially an indicator of the noise variance, our plots display a clear effect of variance reduction by increasing the number of agents. Our observations thus align with Theorem 3.

## 9   Conclusion and Future Work

We contributed to the development of a robustness theory for iterative RL algorithms subject to general compression schemes. In particular, we proposed and analyzed `EF-TD` - a compressed version of the classical `TD`(0) algorithm coupled with error-compensation. We then significantly generalized our analysis to nonlinear stochastic approximation and multi-agent settings. Concretely, our work conveys the following key messages: (i) compressed TD learning algorithms with error-feedback can be just as robust as their optimization counterparts; (ii) the popular error-feedback mechanism extends gracefully beyond the static optimization problems it has been explored for thus far; and (iii) linear convergence speedups in multi-agent TD learning can be achieved with very little communication. Our work opens up several research directions. Studying alternate quantization schemes for RL and exploring other RL algorithms (beyond TD and Q-learning) are immediate next steps. A more open-ended question is the following. `SignSGD` with momentum (Bernstein et al., 2018) is known to exhibit convergence behavior very similar to that of adaptive optimization algorithms such as `ADAM` (Balles & Hennig, 2018), explaining their use in fast training of deep neural networks. In a similar spirit, can we make connections between `SignTD` and adaptive RL algorithms? This is an interesting question to explore since its resolution can potentially lead to faster RL algorithms.

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

## Contents

# A    Preliminary Results and Facts

In this section, we will compile and derive some preliminary results that will play a key role in our subsequent analysis. In what follows, unless otherwise stated, we will use $\|\cdot\|$ to refer to the standard Euclidean norm. The next three results are from Bhandari et al. (2018).

**Lemma 3.** *For all $\theta_1, \theta_2 \in \mathbb{R}^K$, we have:*

$$\sqrt{\omega}\|\theta_1 - \theta_2\| \leq \|\hat{V}_{\theta_1} - \hat{V}_{\theta_2}\|_D \leq \|\theta_1 - \theta_2\|.$$

We remind the reader here that in the above result, $\omega$ is the smallest eigenvalue of the matrix $\Sigma = \Phi^\top D\Phi$. Before stating the next result, we recall the definition of the steady-state TD update direction: for a fixed $\theta \in \mathbb{R}^K$, let

$$\bar{g}(\theta) \triangleq \mathbb{E}_{s_t \sim \pi, s_{t+1} \sim P_\mu(\cdot|s_t)} \left[ g(X_t, \theta) \right].$$

The next lemma provides intuition as to why the expected steady-state TD(0) update direction $\bar{g}(\theta)$ acts like a "pseudo-gradient", driving the TD(0) iterates towards the minimizer $\theta^*$ of the projected Bellman equation.

**Lemma 4.** *For any $\theta \in \mathbb{R}^K$, the following holds:*

$$\langle \theta^* - \theta, \bar{g}(\theta) \rangle \geq (1-\gamma)\|\hat{V}_{\theta^*} - \hat{V}_\theta\|_D^2.$$

We will have occasion to use the following upper bound on the norm of the steady-state TD(0) update direction.

**Lemma 5.** *For any $\theta \in \mathbb{R}^K$, the following holds:*

$$\|\bar{g}(\theta)\| \leq 2\|\hat{V}_{\theta^*} - \hat{V}_\theta\|_D.$$

The following bound on the norm of the random TD(0) update direction will also be invoked several times in our analysis (Srikant & Ying, 2019).

**Lemma 6.** *For any $\theta \in \mathbb{R}^K$, the following holds $\forall t \geq 0$:*

$$\|g_t(\theta)\| \leq 2\|\theta\| + 2\bar{r} \leq 2\|\theta\| + 2\sigma, \tag{13}$$

*where $\sigma = \max\{1, \bar{r}, \|\theta^*\|\}$.*

As mentioned earlier in Section 3, compressed SGD with error-feedback essentially acts like delayed SGD. Thus, for smooth functions where the gradients do not change much, the effect of the delay can be effectively controlled. Unlike this optimization setting, we do not have a fixed objective function at our disposal. So how do we leverage any kind of smoothness property? Fortunately for us, the steady-state TD(0) update direction does satisfy a Lipschitz property; we prove this fact below.

**Lemma 7.** *(Lipschitz property of steady-state TD(0) update direction) For all $\theta_1, \theta_2 \in \mathbb{R}^K$, we have:*

$$\|\bar{g}(\theta_1) - \bar{g}(\theta_2)\| \leq \|\theta_1 - \theta_2\|.$$

*Proof.* We will make use of the explicit affine form of $\bar{g}(\theta)$ shown below (Tsitsiklis & Van Roy, 1997):

$$\bar{g}(\theta) = \Phi^\top D \left(\mathcal{T}_\mu \Phi\theta - \Phi\theta\right) = \bar{A}\theta - \bar{b}, \text{ where } \bar{A} = \Phi^\top D \left(\gamma P_\mu - I\right)\Phi, \text{ and } \bar{b} = -\Phi^\top DR_\mu. \tag{14}$$

In Bhandari et al. (2018), it was shown that $\bar{A}^\top\bar{A} \preceq \Sigma$, where $\Sigma = \Phi^\top D\Phi$. Furthermore, due to feature normalization, it is easy to see that $\lambda_{\max}(\Sigma) \leq 1$. Using these properties, we have:

$$\begin{aligned}
\|\bar{g}(\theta_1) - \bar{g}(\theta_2)\|^2 &= (\theta_1 - \theta_2)^\top \bar{A}^\top \bar{A} (\theta_1 - \theta_2) \\
&\leq \lambda_{\max}(\bar{A}^\top\bar{A})\|\theta_1 - \theta_2\|^2 \\
&\leq \lambda_{\max}(\Sigma)\|\theta_1 - \theta_2\|^2 \\
&\leq \|\theta_1 - \theta_2\|^2,
\end{aligned} \tag{15}$$

which leads to the desired claim. For the first inequality above, we used the Rayleigh-Ritz theorem (Horn & Johnson, 2012, Theorem 4.2.2). $\qquad\square$

An immediate consequence of the above result, in tandem with the fact that $\bar{g}(\theta^*) = 0$, is the following upper bound on the norm of the steady-state $\texttt{TD}(0)$ update direction: $\forall \theta \in \mathbb{R}^K$, we have

$$\|\bar{g}(\theta)\| = \|\bar{g}(\theta) - \bar{g}(\theta^*)\| \leq \|\theta - \theta^*\| \leq \|\theta\| + \sigma. \tag{16}$$

Essentially, this shows that the bound in Lemma 6 applies to the steady-state $\texttt{TD}(0)$ update direction as well. Next, we prove an analog of the Lipschitz property in Lemma 7 for the random $\texttt{TD}(0)$ update direction.

**Lemma 8.** *(**Lipschitz property of the noisy $\texttt{TD}(0)$ update direction**) For all $\theta_1, \theta_2 \in \mathbb{R}^K$, we have:*

$$\|g_t(\theta_1) - g_t(\theta_2)\| \leq 2\|\theta_1 - \theta_2\|.$$

*Proof.* As in the proof of Lemma 7, we will use the fact that the $\texttt{TD}(0)$ update direction is an affine function of the parameter $\theta$. In particular, we have

$$g_t(\theta) = A(X_t)\theta - b(X_t), \text{ where } A(X_t) = \gamma\Phi(s_t)\Phi^\top(s_{t+1}) - \Phi(s_t)\Phi^\top(s_t), \text{ and } b_t = -\phi(s_t)r_t.$$

Thus, we have

$$\begin{aligned}
\|g_t(\theta_1) - g_t(\theta_2)\| &= \|A(X_t)(\theta_1 - \theta_2)\| \\
&\leq \|A(X_t)\|\|\theta_1 - \theta_2\| \\
&\leq \left(\gamma\|\Phi(s_t)\|\|\Phi(s_{t+1})\| + \|\Phi(s_t)\|^2\right)\|\theta_1 - \theta_2\| \\
&\leq 2\|\theta_1 - \theta_2\|,
\end{aligned} \tag{17}$$

where for the last step we used that $\|\Phi(s)\| \leq 1, \forall s \in \mathcal{S}$. $\qquad\square$

In addition to the above results, we will make use of the following facts.

- Given any two vectors $x, y \in \mathbb{R}^K$, the following holds for any $\eta > 0$:

$$\|x + y\|^2 \leq (1 + \eta)\|x\|^2 + \left(1 + \frac{1}{\eta}\right)\|y\|^2. \tag{18}$$

- Given $m$ vectors $x_1, \ldots, x_m \in \mathbb{R}^K$, the following is a simple application of Jensen's inequality:

$$\left\|\sum_{i=1}^m x_i\right\|^2 \leq m\sum_{i=1}^m \|x_i\|^2. \tag{19}$$

# B   Building Intuition: Analysis of Mean-Path `EF-TD`

Since the dynamics of `EF-TD` are quite complex, and have not been studied before, we provide an analysis in this section for the simplest setting in our paper: the noiseless steady-state version of `EF-TD` where $g_t(\theta_t)$ in line 4 of Algorithm 1 is replaced by $\bar{g}(\theta_t)$. We refer to this variant as the mean-path version of `EF-TD`, and compile its governing equations below:

$$
\begin{aligned}
h_t &= \mathcal{Q}_\delta \left( e_{t-1} + \bar{g}(\theta_t) \right), \\
\theta_{t+1} &= \theta_t + \alpha h_t, \\
e_t &= e_{t-1} + \bar{g}(\theta_t) - h_t,
\end{aligned}
\tag{20}
$$

where the above equations hold for $t = 0, 1, \ldots$, with $\theta_0 \in \mathbb{R}^K$, and $e_{-1} = 0$. The goal of this section is to provide an analysis of mean-path `EF-TD`, and, in the process, outline some of the key ideas that will aid us in the more involved settings to follow. We have the following result.

**Theorem 4.** *(**Noiseless Setting**) There exist universal constants $c, C \geq 1$, such that the iterates generated by the mean-path version of `EF-TD` with step-size $\alpha = (1-\gamma)/(c\delta)$ satisfy the following after $T$ iterations:*

$$
\|\theta_T - \theta^*\|_2^2 \leq 2 \left( 1 - \frac{(1-\gamma)^2 \omega}{C\delta} \right)^T \|\theta_0 - \theta^*\|_2^2.
\tag{21}
$$

**Discussion.** Theorem 4 reveals linear convergence of the iterates to $\theta^*$. When $\delta = 1$, i.e., when there is no distortion to the TD(0) direction, the linear rate of convergence exactly matches that in Bhandari et al. (2018). Moreover, when $\delta > 1$, the slowdown in the linear rate by a factor of $\delta$ is also exactly consistent with analogous results for SGD with (biased) compression (Beznosikov et al., 2020). Thus, our result captures - in a transparent way - precisely what one could have hoped for.

To proceed with the analysis of mean-path `EF-TD`, we will make use of the perturbed iterate framework from Mania et al. (2015). In particular, let us define the perturbed iterate $\tilde{\theta}_t \triangleq \theta_t + \alpha e_{t-1}$. Using equation 20, we then obtain:

$$
\begin{aligned}
\tilde{\theta}_{t+1} &= \theta_{t+1} + \alpha e_t \\
&= \theta_t + \alpha h_t + \alpha \left( e_{t-1} + \bar{g}(\theta_t) - h_t \right) \\
&= \tilde{\theta}_t + \alpha \bar{g}(\theta_t).
\end{aligned}
\tag{22}
$$

The final recursion above looks almost like the the standard steady-state TD(0) update direction, other than the fact that $\bar{g}(\theta_t)$ is evaluated at $\theta_t$, and not $\tilde{\theta}_t$. To account for this "mismatch" introduced by the memory-variable $e_{t-1}$, we will analyze the following composite Lyapunov function:

$$
\psi_t \triangleq \|\tilde{\theta}_t - \theta^*\|^2 + \alpha^2 \|e_{t-1}\|^2.
\tag{23}
$$

Note that the above energy function captures the *joint dynamics* of the perturbed iterate and the memory variable. Our goal is to prove that this energy function decays exponentially over time. To that end, we start by establishing a bound on $\|\tilde{\theta}_{t+1} - \theta^*\|^2$ in the following lemma.

**Lemma 9.** *(**Bound on Perturbed Iterate**) Suppose the step-size $\alpha$ is chosen such that $\alpha \leq (1-\gamma)/8$. Then, the iterates generated by the mean-path version of `EF-TD` satisfy:*

$$
\|\tilde{\theta}_{t+1} - \theta^*\|^2 \leq \|\tilde{\theta}_t - \theta^*\|^2 - \frac{\alpha(1-\gamma)}{4} \|\hat{V}_{\tilde{\theta}_t} - \hat{V}_{\theta^*}\|_D^2 + \frac{5\alpha^3}{(1-\gamma)} \|e_{t-1}\|^2.
\tag{24}
$$

*Proof.* Subtracting $\theta^*$ from each side of equation 22 and then squaring both sides yields:

$$
\begin{aligned}
\|\tilde{\theta}_{t+1} - \theta^*\|^2 &= \|\tilde{\theta}_t - \theta^*\|^2 + 2\alpha \langle \tilde{\theta}_t - \theta^*, \bar{g}(\theta_t) \rangle + \alpha^2 \|\bar{g}(\theta_t)\|^2 \\
&= \|\tilde{\theta}_t - \theta^*\|^2 + 2\alpha \langle \theta_t - \theta^*, \bar{g}(\theta_t) \rangle + \alpha^2 \|\bar{g}(\theta_t)\|^2 + 2\alpha \langle \tilde{\theta}_t - \theta_t, \bar{g}(\theta_t) \rangle \\
&\overset{(a)}{\leq} \|\tilde{\theta}_t - \theta^*\|^2 + 2\alpha \langle \theta_t - \theta^*, \bar{g}(\theta_t) \rangle + \alpha \left( \alpha + \frac{1}{\eta} \right) \|\bar{g}(\theta_t)\|^2 + \alpha \eta \|\tilde{\theta}_t - \theta_t\|^2 \\
&\overset{(b)}{=} \|\tilde{\theta}_t - \theta^*\|^2 + 2\alpha \langle \theta_t - \theta^*, \bar{g}(\theta_t) \rangle + \alpha \left( \alpha + \frac{1}{\eta} \right) \|\bar{g}(\theta_t)\|^2 + \alpha^3 \eta \|e_{t-1}\|^2.
\end{aligned}
\tag{25}
$$

For (a), we used the fact that for any two vectors $x, y \in \mathbb{R}^K$, the following holds for all $\eta > 0$,

$$\langle x, y \rangle \leq \frac{1}{2\eta} \|x\|^2 + \frac{\eta}{2} \|y\|^2.$$

We will pick an appropriate $\eta$ shortly. For (b), we used the fact that from the definition of the perturbed iterate, it holds that $\tilde{\theta}_t - \theta_t = \alpha e_{t-1}$. We will now make use of Lemma's 4 and 5. We proceed as follows.

$$
\begin{aligned}
\|\tilde{\theta}_{t+1} - \theta^*\|^2 &\leq \|\tilde{\theta}_t - \theta^*\|^2 + 2\alpha \langle \theta_t - \theta^*, \bar{g}(\theta_t) \rangle + \alpha \left( \alpha + \frac{1}{\eta} \right) \|\bar{g}(\theta_t)\|^2 + \alpha^3 \eta \|e_{t-1}\|^2 \\
&\overset{(a)}{\leq} \|\tilde{\theta}_t - \theta^*\|^2 - 2\alpha(1-\gamma)\|\hat{V}_{\theta_t} - \hat{V}_{\theta^*}\|_D^2 + \alpha \left( \alpha + \frac{1}{\eta} \right) \|\bar{g}(\theta_t)\|^2 + \alpha^3 \eta \|e_{t-1}\|^2 \\
&\overset{(b)}{\leq} \|\tilde{\theta}_t - \theta^*\|^2 - \alpha \left( 2(1-\gamma) - 4\left( \alpha + \frac{1}{\eta} \right) \right) \|\hat{V}_{\theta_t} - \hat{V}_{\theta^*}\|_D^2 + \alpha^3 \eta \|e_{t-1}\|^2 \quad (26) \\
&\overset{(c)}{\leq} \|\tilde{\theta}_t - \theta^*\|^2 - \alpha(1-\gamma) \left( 1 - \frac{4\alpha}{(1-\gamma)} \right) \|\hat{V}_{\theta_t} - \hat{V}_{\theta^*}\|_D^2 + \frac{4\alpha^3}{(1-\gamma)} \|e_{t-1}\|^2 \\
&\overset{(d)}{\leq} \|\tilde{\theta}_t - \theta^*\|^2 - \frac{\alpha(1-\gamma)}{2} \|\hat{V}_{\theta_t} - \hat{V}_{\theta^*}\|_D^2 + \frac{4\alpha^3}{(1-\gamma)} \|e_{t-1}\|^2.
\end{aligned}
$$

In the above steps, (a) follows from Lemma 4; (b) follows from Lemma 5; (c) follows by setting $\eta = 4/(1-\gamma)$; and (d) is a consequence of the fact that $\alpha \leq (1-\gamma)/8$. To complete the proof, we need to relate $\|\hat{V}_{\theta_t} - \hat{V}_{\theta^*}\|_D^2$ to $\|\hat{V}_{\tilde{\theta}_t} - \hat{V}_{\theta^*}\|_D^2$. We do so by using the fact that for any $x, y \in \mathbb{R}^n$, it holds that $\|x + y\|_D^2 \leq 2\|x\|_D^2 + 2\|y\|_D^2$. This yields:

$$
\begin{aligned}
\|\hat{V}_{\tilde{\theta}_t} - \hat{V}_{\theta^*}\|_D^2 &\leq 2\|\hat{V}_{\theta_t} - \hat{V}_{\theta^*}\|_D^2 + 2\|\hat{V}_{\tilde{\theta}_t} - \hat{V}_{\theta_t}\|_D^2 \\
&\overset{(a)}{\leq} 2\|\hat{V}_{\theta_t} - \hat{V}_{\theta^*}\|_D^2 + 2\|\tilde{\theta}_t - \theta_t\|^2 \quad (27) \\
&\leq 2\|\hat{V}_{\theta_t} - \hat{V}_{\theta^*}\|_D^2 + 2\alpha^2 \|e_{t-1}\|^2,
\end{aligned}
$$

where for (a), we used Lemma 3. Rearranging and simplifying, we obtain:

$$-\|\hat{V}_{\theta_t} - \hat{V}_{\theta^*}\|_D^2 \leq -\frac{1}{2}\|\hat{V}_{\tilde{\theta}_t} - \hat{V}_{\theta^*}\|_D^2 + \alpha^2 \|e_{t-1}\|^2.$$

Plugging the above inequality in equation 26 leads to equation 24. This completes the proof. $\qquad \square$

The last term in equation 24 is one which does not show up in the standard analysis of `TD(0)`, and is unique to our setting. In our next result, we control this extra term (depending on the memory variable) by appealing to the contraction property of the compression operator $\mathcal{Q}_\delta(\cdot)$ in equation 4.

**Lemma 10.** *(**Bound on Memory Variable**) For the mean-path version of `EF-TD`, the following holds:*

$$\|e_t\|^2 \leq \left( 1 - \frac{1}{2\delta} + 4\alpha^2 \delta \right) \|e_{t-1}\|^2 + 16\delta \|\hat{V}_{\tilde{\theta}_t} - \hat{V}_{\theta^*}\|_D^2. \quad (28)$$

*Proof.* We begin as follows:

$$
\begin{aligned}
\|e_t\|^2 &= \|e_{t-1} + \bar{g}(\theta_t) - h_t\|^2 \\
&= \|e_{t-1} + \bar{g}(\theta_t) - \mathcal{Q}_\delta \left( e_{t-1} + \bar{g}(\theta_t) \right)\|^2 \\
&\overset{(a)}{\leq} \left( 1 - \frac{1}{\delta} \right) \|e_{t-1} + \bar{g}(\theta_t)\|^2 \quad (29) \\
&\overset{(b)}{\leq} \left( 1 - \frac{1}{\delta} \right) \left( 1 + \frac{1}{\eta} \right) \|e_{t-1}\|^2 + \left( 1 - \frac{1}{\delta} \right) (1 + \eta) \|\bar{g}(\theta_t)\|^2,
\end{aligned}
$$

for some $\eta > 0$ to be chosen by us shortly. Here, for (a), we used the contraction property of $\mathcal{Q}_\delta(\cdot)$ in equation 4; for (b), we used the relaxed triangle inequality equation 18. To ensure that $\|e_t\|^2$ contracts over time, we want

$$\left(1 - \frac{1}{\delta}\right)\left(1 + \frac{1}{\eta}\right) < 1 \implies \eta > (\delta - 1).$$

Accordingly, suppose $\eta = \delta - 1$. Simple calculations then yield

$$\left(1 - \frac{1}{\delta}\right)\left(1 + \frac{1}{\eta}\right) = \left(1 - \frac{1}{2\delta}\right); \quad \left(1 - \frac{1}{\delta}\right)(1 + \eta) < 2\delta.$$

Plugging these bounds back in equation 29, we obtain

$$
\begin{aligned}
\|e_t\|^2 &\leq \left(1 - \frac{1}{2\delta}\right)\|e_{t-1}\|^2 + 2\delta\|\bar{g}(\theta_t)\|^2 \\
&\leq \left(1 - \frac{1}{2\delta}\right)\|e_{t-1}\|^2 + 2\delta\|\bar{g}(\theta_t) - \bar{g}(\tilde{\theta}_t) + \bar{g}(\tilde{\theta}_t)\|^2 \\
&\leq \left(1 - \frac{1}{2\delta}\right)\|e_{t-1}\|^2 + 4\delta\|\bar{g}(\theta_t) - \bar{g}(\tilde{\theta}_t)\|^2 + 4\delta\|\bar{g}(\tilde{\theta}_t)\|^2 \\
&\overset{(a)}{\leq} \left(1 - \frac{1}{2\delta}\right)\|e_{t-1}\|^2 + 4\delta\|\theta_t - \tilde{\theta}_t\|^2 + 4\delta\|\bar{g}(\tilde{\theta}_t)\|^2 \\
&= \left(1 - \frac{1}{2\delta} + 4\alpha^2\delta\right)\|e_{t-1}\|^2 + 4\delta\|\bar{g}(\tilde{\theta}_t)\|^2 \\
&\overset{(b)}{\leq} \left(1 - \frac{1}{2\delta} + 4\alpha^2\delta\right)\|e_{t-1}\|^2 + 16\delta\|\hat{V}_{\tilde{\theta}_t} - \hat{V}_{\theta^*}\|_D^2.
\end{aligned}
\tag{30}
$$

In the above steps, for (a) we used the Lipschitz property of the steady-state $\mathtt{TD}(0)$ update direction, namely Lemma 7; and for (b), we used Lemma 5. This concludes the proof. $\qquad\square$

Lemmas 9 and 10 reveal that the error-dynamics of the perturbed iterate and the memory variable are coupled with each other. As such, they cannot be studied in isolation. This precisely motivates the choice of the Lyapunov function $\psi_t$ in equation 23. We are now ready to complete the proof of Theorem 4.

*Proof.* (**Proof of Theorem 4**) Using the bounds from Lemmas 9 and 10, and recalling the definition of the Lyapunov function $\psi_t$ from equation 23, we have

$$
\begin{aligned}
\psi_{t+1} &\leq \|\tilde{\theta}_t - \theta^*\|^2 - \frac{\alpha(1-\gamma)}{4}\left(1 - \frac{64\alpha\delta}{(1-\gamma)}\right)\|\hat{V}_{\tilde{\theta}_t} - \hat{V}_{\theta^*}\|_D^2 + \alpha^2\left(1 - \frac{1}{2\delta} + 4\alpha^2\delta + \frac{5\alpha}{(1-\gamma)}\right)\|e_{t-1}\|^2 \\
&\leq \underbrace{\left(1 - \frac{\alpha\omega(1-\gamma)}{4}\left(1 - \frac{64\alpha\delta}{(1-\gamma)}\right)\right)}_{\mathcal{A}_1}\|\tilde{\theta}_t - \theta^*\|^2 + \alpha^2 \underbrace{\left(1 - \frac{1}{2\delta} + 4\alpha^2\delta + \frac{5\alpha}{(1-\gamma)}\right)}_{\mathcal{A}_2}\|e_{t-1}\|^2,
\end{aligned}
\tag{31}
$$

where we used Lemma 3 in the last step. Our goal is to establish an inequality of the form $\psi_{t+1} \leq \nu\psi_t$ for some $\nu < 1$. To that end, the next step of the proof is to pick the step-size $\alpha$ in a way such that $\max\{\mathcal{A}_1, \mathcal{A}_2\} < 1$. Accordingly, with $\alpha = (1-\gamma)/(128\delta)$, we have that

$$\mathcal{A}_1 = \left(1 - \frac{(1-\gamma)^2\omega}{1024\delta}\right); \text{ and } \mathcal{A}_2 \leq \left(1 - \frac{1}{4\delta}\right).$$

Furthermore, it is easy to check that $\mathcal{A}_2 \leq \mathcal{A}_1$. Combining these observations with equation 31, we obtain:

$$\psi_{t+1} \leq \left(1 - \frac{(1-\gamma)^2\omega}{1024\delta}\right)\underbrace{\left(\|\tilde{\theta}_t - \theta^*\|^2 + \alpha^2\|e_{t-1}\|^2\right)}_{\psi_t}.$$

Unrolling the above recursion yields:

$$
\begin{aligned}
\psi_T &\le \left(1 - \frac{(1-\gamma)^2 \omega}{1024\delta}\right)^T \psi_0 \\
&= \left(1 - \frac{(1-\gamma)^2 \omega}{1024\delta}\right)^T \|\theta_0 - \theta^*\|^2,
\end{aligned}
\tag{32}
$$

where for the last step, we used the fact that $e_{-1} = 0$. To conclude the proof, it suffices to notice that:

$$
\begin{aligned}
\|\theta_T - \theta^*\|^2 &= \|\theta_T - \tilde{\theta}_T + \tilde{\theta}_T - \theta^*\|^2 \\
&\le 2\|\tilde{\theta}_T - \theta^*\|^2 + 2\|\theta_T - \tilde{\theta}_T\|^2 \\
&= 2\|\tilde{\theta}_T - \theta^*\|^2 + 2\alpha^2 \|e_{T-1}\|^2 \\
&= 2\psi_T.
\end{aligned}
\tag{33}
$$

$\square$

### B.1 Can Compressed TD Methods Without Error-Feedback Still Converge?

Earlier in this section, we provided intuition as to why `EF-TD` converges by studying its dynamics in the steady-state. One might ask: *Can compressed TD algorithms without error-feedback still converge?* If so, under what conditions? We turn to answering these questions in this subsection. In what follows, we will show that compressed TD without error-feedback can still converge, provided certain restrictive conditions on the compression parameter $\delta$ are met. Notably, these conditions are no longer needed when one employs error-feedback. To convey the key ideas, we consider a mean-path version of compressed TD shown below:

$$
\theta_{t+1} = \theta_t + \alpha \mathcal{Q}_\delta(\bar{g}(\theta_t)),
\tag{34}
$$

where $\mathcal{Q}_\delta(\cdot)$ is the compression operator in equation 4. From the above display, we immediately have

$$
\|\theta_{t+1} - \theta^*\|^2 = \|\theta_t - \theta^*\|^2 + 2\alpha\langle\theta_t - \theta^*, \mathcal{Q}_\delta(\bar{g}(\theta_t))\rangle + \alpha^2\|\mathcal{Q}_\delta(\bar{g}(\theta_t))\|^2.
\tag{35}
$$

Among the three terms on the R.H.S. of the above equation, notice that the only term that can lead to a decrease in the iterate error $\|\theta_{t+1} - \theta^*\|^2$ is clearly $2\alpha\langle\theta_t - \theta^*, \mathcal{Q}_\delta(\bar{g}(\theta_t))\rangle$. As such, let us fix a $\theta \in \mathbb{R}^K$, and investigate what we can say about $\langle\theta - \theta^*, \mathcal{Q}_\delta(\bar{g}(\theta))\rangle$. First, notice that if there is no compression, i.e., $\delta = 1$, then $\mathcal{Q}_\delta(\bar{g}(\theta)) = \bar{g}(\theta)$, and we know from Lemmas 3 and 4 that

$$
\langle\theta - \theta^*, \bar{g}(\theta)\rangle \le -\beta\|\theta - \theta^*\|^2,
\tag{36}
$$

where $\beta = \omega(1-\gamma) \in (0,1)$. It is precisely the above key property that causes uncompressed TD to converge to $\theta^*$. Now let us observe:

$$
\begin{aligned}
\langle\theta - \theta^*, \mathcal{Q}_\delta(\bar{g}(\theta))\rangle &= \langle\theta - \theta^*, \bar{g}(\theta)\rangle + \langle\theta - \theta^*, \mathcal{Q}_\delta(\bar{g}(\theta)) - \bar{g}(\theta)\rangle \\
&\le \langle\theta - \theta^*, \bar{g}(\theta)\rangle + \|\theta - \theta^*\|\|\mathcal{Q}_\delta(\bar{g}(\theta)) - \bar{g}(\theta)\| \\
&\overset{(a)}{\le} \langle\theta - \theta^*, \bar{g}(\theta)\rangle + \sqrt{\left(1 - \frac{1}{\delta}\right)}\|\theta - \theta^*\|\|\bar{g}(\theta)\| \\
&\overset{(b)}{\le} \langle\theta - \theta^*, \bar{g}(\theta)\rangle + \sqrt{\left(1 - \frac{1}{\delta}\right)}\|\theta - \theta^*\|^2 \\
&\overset{(c)}{\le} -\left(\beta - \sqrt{\left(1 - \frac{1}{\delta}\right)}\right)\|\theta - \theta^*\|^2.
\end{aligned}
\tag{37}
$$

In the above steps, (a) follows from equation 4, (b) follows from equation 16, and (c) from equation 36. Comparing equation 37 to equation 36, we conclude that for the distorted TD direction $\mathcal{Q}_\delta(\bar{g}(\theta))$ to ensure progress towards $\theta^*$, we need the following condition to hold:

$$\boxed{\sqrt{\left(1 - \frac{1}{\delta}\right)} < \beta.} \tag{38}$$

Simplifying, the above condition amounts to

$$\boxed{\delta < \frac{1}{(1 - \beta^2)}.} \tag{39}$$

The parameter $\beta \in (0, 1)$ gets fixed when one fixes an MDP, the policy to be evaluated, and the feature vectors for linear function approximation. The condition for contraction/convergence in equation 39 tells us that this parameter $\beta$ limits the extent of compression $\delta$. Said differently, one cannot choose the compression level $\delta$ to be arbitrarily large; rather it is dictated by the problem-dependent parameter $\beta$. *It is important to note here that no such restriction on $\delta$ is necessary when one uses error-feedback, as revealed by our analysis for mean-path `EF-TD`. This highlights the benefit of using error-feedback in the context of compressed TD learning.* With these observations in place, let us return to our analysis of the update rule in equation 34. For ease of notation, let us define

$$\zeta \triangleq \left(\beta - \sqrt{\left(1 - \frac{1}{\delta}\right)}\right),$$

and note that if the compression parameter $\delta$ satisfies the condition in equation 39, then $\zeta > 0$. Plugging the bound from equation 37 in equation 35, we obtain

$$\begin{aligned}
\|\theta_{t+1} - \theta^*\|^2 &= \|\theta_t - \theta^*\|^2 + 2\alpha\langle\theta_t - \theta^*, \mathcal{Q}_\delta(\bar{g}(\theta_t))\rangle + \alpha^2\|\mathcal{Q}_\delta(\bar{g}(\theta_t))\|^2 \\
&\leq (1 - 2\alpha\zeta)\|\theta_t - \theta^*\|^2 + \alpha^2\|\mathcal{Q}_\delta(\bar{g}(\theta_t)) - \bar{g}(\theta_t) + \bar{g}(\theta_t)\|^2 \\
&\leq (1 - 2\alpha\zeta)\|\theta_t - \theta^*\|^2 + 2\alpha^2\|\mathcal{Q}_\delta(\bar{g}(\theta_t)) - \bar{g}(\theta_t)\|^2 + 2\alpha^2\|\bar{g}(\theta_t)\|^2 \\
&\overset{(a)}{\leq} (1 - 2\alpha\zeta)\|\theta_t - \theta^*\|^2 + 2\left(2 - \frac{1}{\delta}\right)\alpha^2\|\bar{g}(\theta_t)\|^2 \\
&\overset{(b)}{\leq} \left(1 - 2\alpha\zeta + 4\alpha^2\right)\|\theta_t - \theta^*\|^2,
\end{aligned} \tag{40}$$

where (a) follows from equation 4 and (b) from equation 16. Thus, with $\alpha \leq \zeta/4$, we have

$$\|\theta_{t+1} - \theta^*\|^2 \leq (1 - \alpha\zeta)\|\theta_t - \theta^*\|^2.$$

We conclude that when the compression parameter $\delta$ satisfies the condition in equation 39, and the step-size is chosen to be suitably small, the compressed TD update rule in equation 34 does converge linearly to $\theta^*$.

# C   Warm-Up: Analysis of `EF-TD` with a Projection Step

Before attempting to prove Theorem 1, it is instructive to analyze the behavior of `EF-TD` with a projection step. The benefit of this projection step is that it makes it relatively easier to argue that the iterates generated by `EF-TD` remain uniformly bounded; nonetheless, as we shall soon see, the analysis remains quite non-trivial even in light of this simplification. Let us now jot down the governing equations of the dynamics we plan to study.

$$
\begin{aligned}
h_t &= \mathcal{Q}_\delta \left( e_{t-1} + g_t(\theta_t) \right), \\
\theta_{t+1} &= \Pi_{2,\mathcal{B}} \left( \theta_t + \alpha h_t \right), \\
e_t &= e_{t-1} + g_t(\theta_t) - h_t,
\end{aligned}
\tag{41}
$$

where $\Pi_{2,\mathcal{B}}(\cdot)$ denotes the standard Euclidean projection on to a convex compact subset $\mathcal{B} \subset \mathbb{R}^K$ that is assumed to contain the fixed point $\theta^*$. We also note here that a projection step of the form in equation 41 is common in the literature on stochastic approximation (Borkar, 2009) and RL (Bhandari et al., 2018; Doan et al., 2019).

Our main result concerning the performance of the projected version of `EF-TD` is the following.

**Theorem 5.** *Suppose Assumption 1 holds. There exists a universal constant $c \geq 1$ such that the iterates generated by the projected version of* `EF-TD` *(i.e., equation 41) with step-size $\alpha \leq (1-\gamma)/c$ satisfy the following after $T \geq \tau$ iterations:*

$$
\mathbb{E}\left[ \|\theta_T - \theta^*\|_2^2 \right] \leq C_1 \left( 1 - \alpha\omega(1-\gamma) \right)^{T-\tau} + O\left( \frac{\alpha\tau\delta^2 G^2}{\omega(1-\gamma)} \right),
\tag{42}
$$

*where $C_1 = O(\alpha^2\delta^2 G^2 + G^2)$, and $G$ is the radius of the convex compact set $\mathcal{B}$.*

**Main Takeaway.** We note that the nature of the above guarantee is similar to that of Theorem 1. That said, while the noise term in Theorem 1 is $O(\tau + \delta)$, it is $O(\tau\delta^2)$ in Theorem 5. In words, with the somewhat cruder bounds we obtain via projection, we end up with a looser dependency on the distortion parameter $\delta$. Moreover, the mixing time $\tau$ and the distortion parameter $\delta$ show up in multiplicative form in Theorem 5. In Section D, we will provide a finer analysis (without the need for projection) that yields the tighter $O(\tau + \delta)$ bound.

We now proceed with the proof of Theorem 5. Let us start by defining the projection error $e_{p,t}$ at time-step $t$ as follows: $e_{p,t} = \theta_t - (\theta_{t-1} + \alpha h_{t-1})$. We also define an intermediate sequence $\{\bar{\theta}_t\}$ as follows: $\bar{\theta}_t \triangleq \theta_{t-1} + \alpha h_{t-1}$. Thus, $\theta_t - \bar{\theta}_t = e_{p,t}$. Next, inspired by the perturbed iterate framework in Mania et al. (2015), we define a modified *perturbed iterate* as follows:

$$
\tilde{\theta}_t \triangleq \bar{\theta}_t + \alpha e_{t-1}.
\tag{43}
$$

Based on the above definitions, observe that

$$
\begin{aligned}
\tilde{\theta}_{t+1} &= \bar{\theta}_{t+1} + \alpha e_t \\
&= \theta_t + \alpha h_t + \alpha \left( e_{t-1} + g_t(\theta_t) - h_t \right) \\
&= \theta_t + \alpha g_t(\theta_t) + \alpha e_{t-1} \\
&= \bar{\theta}_t + \alpha e_{t-1} + \alpha g_t(\theta_t) + e_{p,t} \\
&= \tilde{\theta}_t + \alpha g_t(\theta_t) + e_{p,t}.
\end{aligned}
\tag{44}
$$

Subtracting $\theta^*$ from each side of equation 44 and then squaring both sides, we obtain:

$$
\|\tilde{\theta}_{t+1} - \theta^*\|^2 = \|\tilde{\theta}_t - \theta^*\|^2 + \underbrace{2\alpha\langle \tilde{\theta}_t - \theta^*, g_t(\theta_t) \rangle}_{\mathcal{C}_1} + \underbrace{\alpha^2 \|g_t(\theta_t)\|^2}_{\mathcal{C}_2} + \underbrace{2\langle \tilde{\theta}_t - \theta^* + \alpha g_t(\theta_t), e_{p,t} \rangle}_{\mathcal{C}_3} + \underbrace{\|e_{p,t}\|^2}_{\mathcal{C}_4}.
\tag{45}
$$

In what follows, we outline the key steps of our proof that involve bounding each of the terms $\mathcal{C}_1 - \mathcal{C}_4$.

● **Step 1.** The dynamics of the model parameter $\theta_t$, the Markov variable $X_t$, the memory variable $e_t$, and the projection error $e_{p,t}$ are all closely coupled, leading to a dynamical system far more complex than the

standard TD(0) system. To start unravelling this complex dynamical system, our key strategy is to *disentangle the memory variable and the projection error from the perturbed iterate and the Markov data tuple*. To do so, we derive uniform bounds on $e_t, h_t$, and $e_{p,t}$ by exploiting the contraction property in equation 4. This is achieved in Lemma 12.

● **Step 2.** Using the uniform bounds from the previous step in tandem with properties of the Euclidean projection operator, we control terms $\mathcal{C}_2 - \mathcal{C}_4$ in Lemma 13.

● **Step 3.** Bounding $\mathcal{C}_1$ takes the most work. For this step, we exploit the idea of conditioning on the system state sufficiently into the past, and using the geometric mixing property of the Markov chain. As we shall see, conditioning into the past creates the need to control $\|\theta_t - \theta_{t-\tau}\|, \forall t \geq \tau$, where $\tau$ is the mixing time. This is done in Lemma 14. Using the result from Lemma 14, we bound $\mathcal{C}_1$ in Lemma 15.

At the end of the three steps above, what we wish to establish is a recursion of the following form $\forall t \geq \tau$:

$$\mathbb{E}\left[\|\tilde{\theta}_{t+1} - \theta^*\|^2\right] \leq (1 - \alpha\omega(1 - \gamma))\,\mathbb{E}\left[\|\tilde{\theta}_t - \theta^*\|^2\right] + O(\alpha^2\tau\delta^2G^2).$$

To proceed with Step 1, we recall the following result from Nedic et al. (2010).

**Lemma 11.** *Let $\mathcal{B}$ be a nonempty, closed, convex set in $\mathbb{R}^K$. Then, for any $x \in \mathbb{R}^K$, we have:*

(a) $\langle \Pi_{2,\mathcal{B}}(x) - x, x - y \rangle \leq -\|\Pi_{2,\mathcal{B}}(x) - x\|^2, \forall y \in \mathcal{B}.$

(b) $\|\Pi_{\mathcal{B}}(x) - y\|^2 \leq \|x - y\|^2 - \|\Pi_{\mathcal{B}}(x) - x\|^2, \forall y \in \mathcal{B}.$

To lighten notation, let us assume without loss of generality that all rewards are uniformly bounded by 1. Our results can be trivially extended to the case where the uniform bound is some finite number $R_{max}$. To make the calculations cleaner, we also assume that the projection radius $G$ is greater than 1. We have the following key result that provides uniform bounds on the memory variable and the projection error.

**Lemma 12.** *(**Uniform bounds on memory variable and projection error**) For the dynamics in equation 41, the following hold $\forall t \geq 0$:*

(a) $\|e_t\| \leq 6\delta G.$

(b) $\|h_t\| \leq 15\delta G.$

(c) $\|e_{p,t}\| \leq 15\alpha\delta G.$

*Proof.* We start by noting that for all $t \geq 0$,

$$\|g_t(\theta_t)\| = \|A(X_t)\theta_t - b(X_t)\| \leq \|A(X_t)\|\|\theta_t\| + \|b(X_t)\| \leq 2G + 1 \leq 3G, \tag{46}$$

where we used (i) the feature normalization property; (ii) the fact that the rewards are uniformly bounded by 1; and (iii) the fact that due to projection, $\|\theta_t\| \leq 1, \forall t \geq 0$. Next, observe that

$$\begin{aligned}
\|e_t\|^2 &= \|e_{t-1} + g_t(\theta_t) - h_t\|^2 \\
&= \|e_{t-1} + g_t(\theta_t) - \mathcal{Q}_\delta\left(e_{t-1} + g_t(\theta_t)\right)\|^2 \\
&\overset{(a)}{\leq} \left(1 - \frac{1}{\delta}\right)\|e_{t-1} + g_t(\theta_t)\|^2 \\
&\overset{(b)}{\leq} \left(1 - \frac{1}{\delta}\right)\left(1 + \frac{1}{\eta}\right)\|e_{t-1}\|^2 + \left(1 - \frac{1}{\delta}\right)(1 + \eta)\|g_t(\theta_t)\|^2,
\end{aligned} \tag{47}$$

for some $\eta > 0$ to be chosen by us shortly. Here, for (a), we used the contraction property of $\mathcal{Q}_\delta(\cdot)$ in equation 4; for (b), we used the relaxed triangle inequality in equation 18. To ensure that $\|e_t\|^2$ contracts over time, we want

$$\left(1 - \frac{1}{\delta}\right)\left(1 + \frac{1}{\eta}\right) < 1 \implies \eta > (\delta - 1).$$

Accordingly, suppose $\eta = \delta - 1$. Simple calculations then yield

$$\left(1 - \frac{1}{\delta}\right)\left(1 + \frac{1}{\eta}\right) = \left(1 - \frac{1}{2\delta}\right); \qquad \left(1 - \frac{1}{\delta}\right)(1 + \eta) < 2\delta.$$

Plugging these bounds back in equation 47 and using equation 46, we obtain

$$\|e_t\|^2 \leq \left(1 - \frac{1}{2\delta}\right)\|e_{t-1}\|^2 + 2\delta\|g_t(\theta_t)\|^2$$

$$\leq \left(1 - \frac{1}{2\delta}\right)\|e_{t-1}\|^2 + 18\delta G^2.$$

Unrolling the dynamics of the memory variable thus yields:

$$
\begin{aligned}
\|e_t\|^2 &\leq \left(1 - \frac{1}{2\delta}\right)^{t+1}\|e_{-1}\|^2 + 18\delta G^2 \sum_{k=0}^{t}\left(1 - \frac{1}{2\delta}\right)^k \\
&\leq 18\delta G^2 \sum_{k=0}^{\infty}\left(1 - \frac{1}{2\delta}\right)^k \\
&= 36\delta^2 G^2,
\end{aligned}
\tag{48}
$$

where we used the fact that $e_{-1} = 0$. Thus, $\|e_t\| \leq 6\delta G$, which establishes part (a). For part (b), we notice that $h_t = e_{t-1} - e_t + g_t(\theta_t)$. This immediately yields:

$$\|h_t\| \leq \|e_t\| + \|e_{t-1}\| + \|g_t(\theta_t)\| \leq 12\delta G + 3G \leq 15\delta G,$$

where we used the fact that $\delta \geq 1$, the bound from part (a), and the uniform bound on $g_t(\theta_t)$ established earlier.

Next, for part (c), we use part (b) of Lemma 11 to observe that

$$\|e_{p,t}\|^2 = \|\Pi_{2,\mathcal{B}}(\bar{\theta}_t) - \bar{\theta}_t\|^2 \leq \|\bar{\theta}_t - \theta\|^2, \forall \theta \in \mathcal{B}.$$

Since the above bound holds for all $\theta \in \mathcal{B}$, and $\theta_{t-1} \in \mathcal{B}$, we have

$$\|e_{p,t}\|^2 \leq \|\bar{\theta}_t - \theta_{t-1}\|^2 = \alpha^2\|h_t\|^2 \leq 225\alpha^2\delta^2 G^2,$$

where we used the fact that $\bar{\theta}_t = \theta_{t-1} + \alpha h_{t-1}$ by definition, and also the bound on $\|h_t\|$ from part (b). This concludes the proof. $\qquad\square$

From the proof of the above lemma, bounds on terms $\mathcal{C}_2$ and $\mathcal{C}_4$ in equation 45 follow immediately. In our next result, we bound the term $\mathcal{C}_3$.

**Lemma 13.** *For the dynamics in equation 41, the following holds for all $t \geq 0$:*

$$2\langle \tilde{\theta}_t - \theta^* + \alpha g_t(\theta_t), e_{p,t}\rangle \leq 45\alpha^2\delta^2 G^2.$$

*Proof.* We start by decomposing the term we wish to bound into three parts:

$$
\begin{aligned}
2\langle \tilde{\theta}_t - \theta^* + \alpha g_t(\theta_t), e_{p,t}\rangle &= 2\langle \bar{\theta}_t - \theta^* + \alpha e_{t-1} + \alpha g_t(\theta_t), e_{p,t}\rangle \\
&= \underbrace{2\langle \bar{\theta}_t - \theta^*, e_{p,t}\rangle}_{\mathcal{C}_{31}} + \underbrace{2\alpha\langle e_{t-1}, e_{p,t}\rangle}_{\mathcal{C}_{32}} + \underbrace{2\alpha\langle g_t(\theta_t), e_{p,t}\rangle}_{\mathcal{C}_{33}}.
\end{aligned}
\tag{49}
$$

We now bound each of the three terms above separately. For $\mathcal{C}_{31}$, we have

$$2\langle \bar{\theta}_t - \theta^*, e_{p,t}\rangle = 2\langle \bar{\theta}_t - \theta^*, \theta_t - \bar{\theta}_t\rangle \leq -2\|\theta_t - \bar{\theta}_t\|^2 = -2\|e_{p,t}\|^2, \tag{50}$$

where we used part (a) of the projection lemma, namely Lemma 11, with $x = \bar{\theta}_t$ and $y = \theta^*$; here, note that we used the fact that $\theta^* \in \mathcal{B}$. Next, for $\mathcal{C}_{32}$, observe that

$$
\begin{aligned}
2\alpha \langle e_{t-1}, e_{p,t} \rangle &\leq \alpha^2 \|e_{t-1}\|^2 + \|e_{p,t}\|^2 \\
&\leq 36\alpha^2 \delta^2 G^2 + \|e_{p,t}\|^2,
\end{aligned}
\tag{51}
$$

where we used the bound on $\|e_{t-1}\|$ from part (a) of Lemma 12. Notice that we have kept $\|e_{p,t}\|^2$ as is in the above bound since we will cancel off its effect with one of the negative terms from the upper bound on $\mathcal{C}_{31}$. Finally, we bound the term $\mathcal{C}_{33}$ as follows:

$$
\begin{aligned}
2\alpha \langle g_t(\theta_t), e_{p,t} \rangle &\leq \alpha^2 \|g_t(\theta_t)\|^2 + \|e_{p,t}\|^2 \\
&\leq 9\alpha^2 G^2 + \|e_{p,t}\|^2,
\end{aligned}
\tag{52}
$$

where we used the uniform bound on $g_t(\theta_t)$ from equation 46. Combining the bounds in equations 50, 51, and 52, and using the fact that $\delta \geq 1$ yields the desired result. $\qquad \square$

Notice that up until now, we have not made any use of the geometric mixing property of the underlying Markov chain. We will call upon this property while bounding $\mathcal{C}_1$. But first, we need the following intermediate result.

**Lemma 14.** *For the dynamics in equation 41, the following holds for all $t \geq \tau$:*

$$
\|\theta_t - \theta_{t-\tau}\| \leq 60\alpha\tau\delta G.
\tag{53}
$$

*Proof.* Based on equation 44, observe that

$$
\begin{aligned}
\|\tilde{\theta}_{t+1} - \tilde{\theta}_t\| &\leq \alpha\|g_t(\theta_t)\| + \|e_{p,t}\| \\
&\leq 3\alpha G + 15\alpha\delta G \\
&\leq 18\alpha\delta G.
\end{aligned}
\tag{54}
$$

We also note that

$$
\tilde{\theta}_t - \tilde{\theta}_{t-\tau} = \sum_{k=t-\tau}^{t-1} \left( \tilde{\theta}_{k+1} - \tilde{\theta}_k \right).
$$

Based on equation 54, we then immediately have

$$
\begin{aligned}
\|\tilde{\theta}_t - \tilde{\theta}_{t-\tau}\| &\leq \sum_{k=t-\tau}^{t-1} \|\tilde{\theta}_{k+1} - \tilde{\theta}_k\| \\
&\leq \sum_{k=t-\tau}^{t-1} (18\alpha\delta G) \\
&\leq 18\alpha\tau\delta G.
\end{aligned}
\tag{55}
$$

Our goal is to now relate the above bound on $\|\tilde{\theta}_t - \tilde{\theta}_{t-\tau}\|$ to one on $\|\theta_t - \theta_{t-\tau}\|$. To that end, observe that

$$
\begin{aligned}
\tilde{\theta}_t &= \theta_t + \alpha e_{t-1} - e_{p,t}, \\
\tilde{\theta}_{t-\tau} &= \theta_{t-\tau} + \alpha e_{t-\tau-1} - e_{p,t-\tau}.
\end{aligned}
$$

This gives us exactly what we need:

$$
\begin{aligned}
\|\theta_t - \theta_{t-\tau}\| &\leq \|\tilde{\theta}_t - \tilde{\theta}_{t-\tau}\| + \alpha\|e_{t-1} - e_{t-\tau-1}\| + \|e_{p,t-\tau} - e_{p,t}\| \\
&\leq \|\tilde{\theta}_t - \tilde{\theta}_{t-\tau}\| + \alpha\|e_{t-1}\| + \alpha\|e_{t-\tau-1}\| + \|e_{p,t-\tau}\| + \|e_{p,t}\| \\
&\leq 18\alpha\tau\delta G + 12\alpha\delta G + 30\alpha\delta G \leq 60\alpha\tau\delta G,
\end{aligned}
\tag{56}
$$

where we used equation 55, parts (a) and (c) of Lemma 12, and assumed that the mixing time $\tau \geq 1$ to state the bounds more cleanly. This completes the proof. $\qquad \square$

We now turn towards establishing an upper bound on the term $\mathcal{C}_1$ in equation 45.

**Lemma 15.** *Suppose Assumption 1 holds. For the dynamics in equation 41, the following then holds $\forall t \geq \tau$ :*

$$\mathbb{E}\left[2\alpha\langle\tilde{\theta}_t - \theta^*, g_t(\theta_t)\rangle\right] \leq -2\alpha(1-\gamma)\mathbb{E}\left[\|V_{\theta_t} - V_{\theta^*}\|_D^2\right] + 1454\alpha^2\delta\tau G^2.$$

*Proof.* Let us first focus on bounding $T \triangleq \langle\theta_t - \theta^*, g_t(\theta_t) - \bar{g}(\theta_t)\rangle$. Trivially, observe that

$$T = \underbrace{\langle\theta_t - \theta_{t-\tau}, g_t(\theta_t) - \bar{g}(\theta_t)\rangle}_{T_1} + \underbrace{\langle\theta_{t-\tau} - \theta^*, g_t(\theta_t) - \bar{g}(\theta_t)\rangle}_{T_2}.$$

To bound $T_1$, we recall from Lemma 7 that for all $\theta_1, \theta_2 \in \mathbb{R}^K$, it holds that

$$\|\bar{g}(\theta_1) - \bar{g}(\theta_2)\| \leq \|\theta_1 - \theta_2\|.$$

Since $\bar{g}(\theta^*) = 0$, the above inequality immediately implies that $\|\bar{g}(\theta)\| \leq \|\theta\| + \|\theta^*\|, \forall\theta \in \mathbb{R}^K$. In particular, for any $\theta \in \mathcal{B}$, we then have that $\|\bar{g}(\theta)\| \leq 2G$ (since $\theta^* \in \mathcal{B}$). Using the bound on $\|\theta_t - \theta_{t-\tau}\|$ from Lemma 14, and the uniform bound on the noisy TD(0) update direction from equation 46, we then obtain

$$\begin{aligned}
T_1 &\leq (\|\theta_t - \theta_{t-\tau}\|)(\|g_t(\theta_t) - \bar{g}(\theta_t)\|) \\
&\leq (\|\theta_t - \theta_{t-\tau}\|)(\|g_t(\theta_t)\| + \|\bar{g}(\theta_t)\|) \\
&\leq 60\alpha\tau\delta G(3G + 2G) = 300\alpha\tau\delta G^2.
\end{aligned} \tag{57}$$

To bound $T_2$, we further split it into two parts as follows:

$$T_2 = \underbrace{\langle\theta_{t-\tau} - \theta^*, g_t(\theta_{t-\tau}) - \bar{g}(\theta_{t-\tau})\rangle}_{T_{21}} + \underbrace{\langle\theta_{t-\tau} - \theta^*, g_t(\theta_t) - g_t(\theta_{t-\tau}) + \bar{g}(\theta_{t-\tau}) - \bar{g}(\theta_t)\rangle}_{T_{22}}.$$

To bound $T_{22}$, we will exploit the Lipschitz property of the TD(0) update directions in tandem with Lemma 14. Specifically, observe that:

$$\begin{aligned}
T_{22} &\leq \|\theta_{t-\tau} - \theta^*\|(\|g_t(\theta_t) - g_t(\theta_{t-\tau})\| + \|\bar{g}(\theta_{t-\tau}) - \bar{g}(\theta_t)\|) \\
&\overset{(a)}{\leq} 2G(\|g_t(\theta_t) - g_t(\theta_{t-\tau})\| + \|\bar{g}(\theta_{t-\tau}) - \bar{g}(\theta_t)\|) \\
&\overset{(b)}{\leq} 6G\|\theta_t - \theta_{t-\tau}\| \\
&\overset{(c)}{\leq} 360\alpha\tau\delta G^2.
\end{aligned} \tag{58}$$

In the above steps, (a) follows from projection; (b) follows from Lemmas 7 and 8; and (c) follows from Lemma 14. It remains to bound $T_{21}$. This is precisely the only place in the entire proof that we will use the geometric mixing property of the Markov chain in Definition 1. We proceed as follows.

$$\begin{aligned}
\mathbb{E}[T_{21}] &= \mathbb{E}\left[\langle\theta_{t-\tau} - \theta^*, g_t(\theta_{t-\tau}) - \bar{g}(\theta_{t-\tau})\rangle\right] \\
&= \mathbb{E}\left[\mathbb{E}\left[\langle\theta_{t-\tau} - \theta^*, g_t(\theta_{t-\tau}) - \bar{g}(\theta_{t-\tau})\rangle|\theta_{t-\tau}, X_{t-\tau}\right]\right] \\
&= \mathbb{E}\left[\langle\theta_{t-\tau} - \theta^*, \mathbb{E}\left[g_t(\theta_{t-\tau}) - \bar{g}(\theta_{t-\tau})|\theta_{t-\tau}, X_{t-\tau}\right]\rangle\right] \\
&\leq \mathbb{E}\left[\|\theta_{t-\tau} - \theta^*\|\|\mathbb{E}\left[g_t(\theta_{t-\tau}) - \bar{g}(\theta_{t-\tau})|\theta_{t-\tau}, X_{t-\tau}\right]\|\right] \\
&\overset{(a)}{\leq} 2\alpha G(\mathbb{E}[\|\theta_{t-\tau} - \theta^*\|]) \\
&\leq 4\alpha G^2,
\end{aligned} \tag{59}$$

where (a) follows from the definition of the mixing time $\tau$. Combining the above bound with those in equations 57 and 58, we obtain

$$\mathbb{E}[T] \leq 664\alpha\tau\delta G^2, \tag{60}$$

where we used $\tau \geq 1$ and $\delta \geq 1$.

We can now go back to bounding $\mathcal{C}_1$ as follows:

$$
\begin{aligned}
\mathcal{C}_1 &= 2\alpha\langle\theta_t - \theta^* + \alpha e_{t-1} - e_{p,t}, g_t(\theta_t)\rangle \\
&= 2\alpha\langle\theta_t - \theta^*, g_t(\theta_t)\rangle + 2\alpha^2\langle e_{t-1}, g_t(\theta_t)\rangle - 2\alpha\langle e_{p,t}, g_t(\theta_t)\rangle \\
&\leq 2\alpha\langle\theta_t - \theta^*, g_t(\theta_t)\rangle + 2\alpha^2\|e_{t-1}\|\|g_t(\theta_t)\| + 2\alpha\|e_{p,t}\|\|g_t(\theta_t)\| \\
&\leq 2\alpha\langle\theta_t - \theta^*, g_t(\theta_t)\rangle + 126\alpha^2\delta G^2,
\end{aligned}
\tag{61}
$$

where we used equation 46, and parts (a) and (c) of Lemma 12. We continue as follows:

$$
\mathcal{C}_1 \leq 2\alpha\langle\theta_t - \theta^*, \bar{g}(\theta_t)\rangle + 2\alpha T + +126\alpha^2\delta G^2.
$$

Using Lemma 4 and the bound we derived on $T$ in equation 60, we finally obtain

$$
\mathbb{E}\left[\mathcal{C}_1\right] \leq -2\alpha(1-\gamma)\mathbb{E}\left[\|\hat{V}_{\theta_t} - \hat{V}_{\theta^*}\|_D^2\right] + 1454\alpha^2\delta\tau G^2,
$$

where in the last step, we used Lemma 4. $\qquad\square$

We can now complete the proof of Theorem 5.

*Proof.* (**Proof of Theorem 5**) We combine the bounds derived previously on the terms $\mathcal{C}_1$-$\mathcal{C}_4$ in Lemmas 12, 13, and 15 to obtain that $\forall t \geq \tau$,

$$
\begin{aligned}
\mathbb{E}\left[\|\tilde{\theta}_{t+1} - \theta^*\|_2^2\right] &\leq \mathbb{E}\left[\|\tilde{\theta}_t - \theta^*\|_2^2\right] - 2\alpha(1-\gamma)\mathbb{E}\left[\|\hat{V}_{\theta_t} - \hat{V}_{\theta^*}\|_D^2\right] + 1454\alpha^2\delta\tau G^2 \\
&\quad + 9\alpha^2 G^2 + 45\alpha^2\delta^2 G^2 + 225\alpha^2\delta^2 G^2 \\
&\leq \mathbb{E}\left[\|\tilde{\theta}_t - \theta^*\|_2^2\right] - 2\alpha(1-\gamma)\mathbb{E}\left[\|\hat{V}_{\theta_t} - \hat{V}_{\theta^*}\|_D^2\right] + 1733\alpha^2\delta^2\tau G^2.
\end{aligned}
\tag{62}
$$

To proceed, we need to relate $\|\hat{V}_{\theta_t} - \hat{V}_{\theta^*}\|_D^2$ to $\|\hat{V}_{\tilde{\theta}_t} - \hat{V}_{\theta^*}\|_D^2$. We do so by using the fact that for any $x, y \in \mathbb{R}^n$, it holds that $\|x + y\|_D^2 \leq 2\|x\|_D^2 + 2\|y\|_D^2$. This yields:

$$
\begin{aligned}
\|\hat{V}_{\tilde{\theta}_t} - \hat{V}_{\theta^*}\|_D^2 &\leq 2\|\hat{V}_{\theta_t} - \hat{V}_{\theta^*}\|_D^2 + 2\|\hat{V}_{\tilde{\theta}_t} - \hat{V}_{\theta_t}\|_D^2 \\
&\overset{(a)}{\leq} 2\|\hat{V}_{\theta_t} - \hat{V}_{\theta^*}\|_D^2 + 2\|\tilde{\theta}_t - \theta_t\|^2,
\end{aligned}
\tag{63}
$$

where for (a), we used Lemma 3. We thus have

$$
\begin{aligned}
-2\alpha(1-\gamma)\|\hat{V}_{\theta_t} - \hat{V}_{\theta^*}\|_D^2 &\leq -\alpha(1-\gamma)\|\hat{V}_{\tilde{\theta}_t} - \hat{V}_{\theta^*}\|_D^2 + 2\alpha(1-\gamma)\|\tilde{\theta}_t - \theta_t\|^2 \\
&\leq -\alpha(1-\gamma)\|\hat{V}_{\tilde{\theta}_t} - \hat{V}_{\theta^*}\|_D^2 + 2\alpha(1-\gamma)\|\alpha e_{t-1} - e_{p,t}\|^2 \\
&\leq -\alpha(1-\gamma)\|\hat{V}_{\tilde{\theta}_t} - \hat{V}_{\theta^*}\|_D^2 + 4\alpha(1-\gamma)\left(\alpha^2\|e_{t-1}\|^2 + \|e_{p,t}\|^2\right) \\
&\leq -\alpha(1-\gamma)\|\hat{V}_{\tilde{\theta}_t} - \hat{V}_{\theta^*}\|_D^2 + 1044\alpha^3(1-\gamma)\delta^2 G^2,
\end{aligned}
\tag{64}
$$

where we used Lemma 12. Plugging the above bound back in equation 62 yields:

$$
\begin{aligned}
\mathbb{E}\left[\|\tilde{\theta}_{t+1} - \theta^*\|_2^2\right] &\leq \mathbb{E}\left[\|\tilde{\theta}_t - \theta^*\|_2^2\right] - \alpha(1-\gamma)\mathbb{E}\left[\|\hat{V}_{\tilde{\theta}_t} - \hat{V}_{\theta^*}\|_D^2\right] + 2777\alpha^2\delta^2\tau G^2 \\
&\leq \left(1 - \alpha\omega(1-\gamma)\right)\mathbb{E}\left[\|\tilde{\theta}_t - \theta^*\|_2^2\right] + 2777\alpha^2\delta^2\tau G^2, \forall t \geq \tau,
\end{aligned}
\tag{65}
$$

where we used Lemma 3 in the last step. Unrolling the above recursion starting from $t = \tau$, we obtain

$$
\begin{aligned}
\mathbb{E}\left[\|\tilde{\theta}_T - \theta^*\|_2^2\right] &\leq \left(1 - \alpha\omega(1-\gamma)\right)^{T-\tau}\mathbb{E}\left[\|\tilde{\theta}_\tau - \theta^*\|_2^2\right] + 2777\alpha^2\delta^2\tau G^2\sum_{k=0}^{\infty}\left(1 - \alpha\omega(1-\gamma)\right)^k \\
&= \left(1 - \alpha\omega(1-\gamma)\right)^{T-\tau}\mathbb{E}\left[\|\tilde{\theta}_\tau - \theta^*\|_2^2\right] + 2777\frac{\alpha\tau\delta^2 G^2}{\omega(1-\gamma)}.
\end{aligned}
\tag{66}
$$

We now make use of the following equation twice.

$$\tilde{\theta}_t = \theta_t + \alpha e_{t-1} - e_{p,t}.$$

First, setting $t = \tau$ in the above equation, subtracting $\theta^*$ from both sides, and then simplifying, we observe that

$$\|\tilde{\theta}_\tau - \theta^*\| \le \|\theta_\tau - \theta^*\| + \alpha\|e_{\tau-1}\| + \|e_{p,\tau}\| = O\left(\alpha\delta G + G\right),$$

where we invoked Lemma 12. Thus,

$$\mathbb{E}\left[\|\tilde{\theta}_\tau - \theta^*\|_2^2\right] = O\left(\alpha^2\delta^2 G^2 + G^2\right).$$

Using similar arguments as above, one can also show that

$$\|\theta_T - \theta^*\|^2 \le 3\|\tilde{\theta}_T - \theta^*\|^2 + 3\alpha^2\|e_{t-1}\|^2 + 3\|e_{p,t}\|^2 \le 3\|\tilde{\theta}_T - \theta^*\|^2 + O(\alpha^2\delta^2 G^2).$$

Plugging the two bounds we derived above in equation 66 completes the proof. □

# D   Analysis of `EF-TD` without Projection: Proof of Theorem 1 and Theorem 2

In this section, we will prove Theorem 1. In particular, via a finer analysis relative to that in Appendix C, we will (i) show that the iterates generated by `EF-TD` remain bounded without the need for an explicit projection step to make this happen; and (ii) obtain a tighter bound w.r.t. the distortion parameter $\delta$. At this stage, we remind the reader about the dynamics we are interested in analyzing:

$$
\begin{aligned}
h_t &= \mathcal{Q}_\delta \left( e_{t-1} + g_t(\theta_t) \right), \\
\theta_{t+1} &= \theta_t + \alpha h_t, \\
e_t &= e_{t-1} + g_t(\theta_t) - h_t.
\end{aligned}
\tag{67}
$$

To proceed with the analysis of the above dynamics, let us define the perturbed iterate $\tilde{\theta}_t \triangleq \theta_t + \alpha e_{t-1}$. Using equation 67, we then obtain:

$$
\begin{aligned}
\tilde{\theta}_{t+1} &= \theta_{t+1} + \alpha e_t \\
&= \theta_t + \alpha h_t + \alpha \left( e_{t-1} + g_t(\theta_t) - h_t \right) \\
&= \tilde{\theta}_t + \alpha g_t(\theta_t).
\end{aligned}
\tag{68}
$$

The final recursion above looks almost like the `TD`(0) update, other than the fact that $g_t(\theta_t)$ is evaluated at $\theta_t$, and not $\tilde{\theta}_t$. To account for this "mismatch" introduced by the memory-variable $e_{t-1}$, we will analyze the following composite Lyapunov function:

$$
\psi_t \triangleq \mathbb{E}[\tilde{d}_t^2 + \alpha^2 \|e_{t-1}\|^2], \text{ where } \tilde{d}_t^2 = \|\tilde{\theta}_t - \theta^*\|^2.
\tag{69}
$$

Note that the above energy function captures the *joint dynamics* of the perturbed iterate and the memory variable. Our goal is to prove that this energy function decays exponentially over time (up to noise terms). To that end, we start by establishing a bound on $\tilde{d}_{t+1}^2$ in the following lemma.

**Lemma 16.** *(**Bound on Perturbed Iterate**) Suppose the step-size $\alpha$ satisfies $\alpha \leq 1/12$. For the `EF-TD` algorithm, the following bound then holds for $\forall t \geq 0$:*[3]

$$
\tilde{d}_{t+1}^2 \leq \left( 1 - \alpha\omega(1-\gamma) + 24\alpha^2 \right) \tilde{d}_t^2 + \frac{6\alpha^3}{\omega(1-\gamma)} \|e_{t-1}\|^2 + 2\alpha\langle\tilde{\theta}_t - \theta^*, g_t(\tilde{\theta}_t) - \bar{g}(\tilde{\theta}_t)\rangle + 32\alpha^2\sigma^2.
\tag{70}
$$

*Proof.* Subtracting $\theta^*$ from each side of equation 68 and then squaring both sides yields:

$$
\begin{aligned}
\tilde{d}_{t+1}^2 &= \tilde{d}_t^2 + 2\alpha\langle\tilde{\theta}_t - \theta^*, g_t(\theta_t)\rangle + \alpha^2\|g_t(\theta_t)\|^2 \\
&= \tilde{d}_t^2 + \underbrace{2\alpha\langle\tilde{\theta}_t - \theta^*, g_t(\tilde{\theta}_t)\rangle}_{(*)} + \underbrace{2\alpha\langle\tilde{\theta}_t - \theta^*, g_t(\theta_t) - g_t(\tilde{\theta}_t)\rangle}_{(**)} + \underbrace{\alpha^2\|g_t(\theta_t)\|^2}_{(***)}.
\end{aligned}
\tag{71}
$$

We now have:

$$
\begin{aligned}
(*) &= 2\alpha\langle\tilde{\theta}_t - \theta^*, \bar{g}(\tilde{\theta}_t)\rangle + 2\alpha\langle\tilde{\theta}_t - \theta^*, g_t(\tilde{\theta}_t) - \bar{g}(\tilde{\theta}_t)\rangle \\
&\leq -2\alpha\omega(1-\gamma)\tilde{d}_t^2 + 2\alpha\langle\tilde{\theta}_t - \theta^*, g_t(\tilde{\theta}_t) - \bar{g}(\tilde{\theta}_t)\rangle,
\end{aligned}
\tag{72}
$$

where in the second step, we invoked Lemmas 3 and 4. To bound $(**)$, we proceed as follows:

$$
\begin{aligned}
(**) &\overset{(a)}{\leq} 4\alpha\tilde{d}_t\|\theta_t - \tilde{\theta}_t\| \\
&\overset{(b)}{\leq} \frac{2\alpha}{\eta}\tilde{d}_t^2 + 2\alpha\eta\|\theta_t - \tilde{\theta}_t\|^2 \\
&\overset{(c)}{=} \frac{2\alpha}{\eta}\tilde{d}_t^2 + 2\alpha^3\eta\|e_{t-1}\|^2,
\end{aligned}
\tag{73}
$$

---

[3]The requirement that $\alpha \leq 1/12$ is not necessary to obtain the type of bound in equation 70. Instead, it only serves to simplify some of the leading constants in the bound.

where $\eta > 0$ is a constant to be decided shortly. In the above steps, (a) follows from the Cauchy-Schwarz inequality and the Lipschitz property of the TD(0) update direction in Lemma 8. For (b), we used the fact that for any two scalars $x, y \in \mathbb{R}$, the following holds for all $\eta > 0$,

$$xy \le \frac{1}{2\eta}x^2 + \frac{\eta}{2}y^2.$$

Finally, for (c), we simply used the fact that $\tilde{\theta}_t - \theta_t = \alpha e_{t-1}$. To bound $(***)$, observe that

$$
\begin{aligned}
\|g_t(\theta_t)\|^2 &\overset{(a)}{\le} 4(\|\theta_t\| + \sigma)^2 \\
&\le 8(\|\theta_t\|^2 + \sigma^2) \\
&\overset{(b)}{\le} 8\left(3\|\theta_t - \tilde{\theta}_t\|^2 + 3\|\tilde{\theta}_t - \theta^*\|^2 + 3\|\theta^*\|^2 + \sigma^2\right) \\
&\overset{(c)}{\le} 24\alpha^2\|e_{t-1}\|^2 + 24\tilde{d}_t^2 + 32\sigma^2,
\end{aligned}
\tag{74}
$$

where (a) follows from Lemma 6, (b) follows from equation 19, and (c) follows from noting that $\|\theta^*\| \le \sigma$. Plugging the bounds in equations 72, 73, and 74 in equation 71, we obtain:

$$
\begin{aligned}
\tilde{d}_{t+1}^2 &\le \left(1 - 2\alpha\omega(1-\gamma) + \frac{2\alpha}{\eta} + 24\alpha^2\right)\tilde{d}_t^2 + 2\alpha^3(\eta + 12\alpha)\|e_{t-1}\|^2 + 2\alpha\langle\tilde{\theta}_t - \theta^*, g_t(\tilde{\theta}_t) - \bar{g}(\tilde{\theta}_t)\rangle \\
&\quad + 32\alpha^2\sigma^2.
\end{aligned}
\tag{75}
$$

The result follows from setting $\eta = \frac{2}{\omega(1-\gamma)}$, and simplifying using $\alpha \le 1/12$. $\qquad\square$

Unlike the standard TD(0) analysis, we note from Lemma 16 that the distance to optimality of the iterates is intimately coupled with the magnitude of the memory variable $e_t$. As such, to proceed, we need to bound the growth of this memory variable. We do so in the following lemma.

**Lemma 17.** *(**Bound on Memory Variable**) For the* EF-TD *algorithm, the following bound holds for* $\forall t \ge 0$*:*

$$\|e_t\|^2 \le \left(1 - \frac{1}{2\delta} + 16\alpha^2\delta\right)\|e_{t-1}\|^2 + 64\delta\tilde{d}_t^2 + 96\delta\sigma^2. \tag{76}$$

*Proof.* We begin as follows:

$$
\begin{aligned}
\|e_t\|^2 &= \|e_{t-1} + g_t(\theta_t) - h_t\|^2 \\
&= \|e_{t-1} + g_t(\theta_t) - \mathcal{Q}_\delta\left(e_{t-1} + g_t(\theta_t)\right)\|^2 \\
&\overset{(a)}{\le} \left(1 - \frac{1}{\delta}\right)\|e_{t-1} + g_t(\theta_t)\|^2 \\
&\overset{(b)}{\le} \left(1 - \frac{1}{\delta}\right)\left(1 + \frac{1}{\eta}\right)\|e_{t-1}\|^2 + \left(1 - \frac{1}{\delta}\right)(1 + \eta)\|g_t(\theta_t)\|^2,
\end{aligned}
\tag{77}
$$

where (a) follows from the contraction property of $\mathcal{Q}_\delta(\cdot)$ in equation 4, (b) makes use of the relaxed triangle inequality in equation 18, and $\eta > 0$ is a constant to be chosen by us shortly. To ensure that $\|e_t\|$ contracts

over time, we set $\eta = \delta - 1$ to obtain:

$$
\begin{aligned}
\|e_t\|^2 &\leq \left(1 - \frac{1}{2\delta}\right) \|e_{t-1}\|^2 + 2\delta\|g_t(\theta_t)\|^2 \\
&\leq \left(1 - \frac{1}{2\delta}\right) \|e_{t-1}\|^2 + 2\delta\|g_t(\theta_t) - g_t(\tilde{\theta}_t) + g_t(\tilde{\theta}_t)\|^2 \\
&\leq \left(1 - \frac{1}{2\delta}\right) \|e_{t-1}\|^2 + 4\delta\|g_t(\theta_t) - g_t(\tilde{\theta}_t)\|^2 + 4\delta\|g_t(\tilde{\theta}_t)\|^2 \\
&\overset{(a)}{\leq} \left(1 - \frac{1}{2\delta}\right) \|e_{t-1}\|^2 + 16\delta\|\theta_t - \tilde{\theta}_t\|^2 + 4\delta\|g_t(\tilde{\theta}_t)\|^2 \\
&= \left(1 - \frac{1}{2\delta} + 16\alpha^2\delta\right) \|e_{t-1}\|^2 + 4\delta\|g_t(\tilde{\theta}_t)\|^2 \\
&\overset{(b)}{\leq} \left(1 - \frac{1}{2\delta} + 16\alpha^2\delta\right) \|e_{t-1}\|^2 + 32\delta\left(\|\tilde{\theta}_t\|^2 + \sigma^2\right) \\
&\leq \left(1 - \frac{1}{2\delta} + 16\alpha^2\delta\right) \|e_{t-1}\|^2 + 32\delta\left(2\tilde{d}_t^2 + 3\sigma^2\right).
\end{aligned}
\tag{78}
$$

In the above steps, for (a) we used the Lipschitz property of the noisy `TD`(0) update direction, and for (b), we appealed to Lemma 6. This completes the proof. $\square$

## D.1 Bounding the Drift and Bias Terms

Inspecting Lemma 16, it is apparent that we need to bound the "bias" term $\langle\tilde{\theta}_t - \theta^*, g_t(\tilde{\theta}_t) - \bar{g}(\tilde{\theta}_t)\rangle$. This requires some work for the following reasons.

- In the (compressed) optimization setting, one does not encounter this term since $g_t(\tilde{\theta}_t)$ is an unbiased version of $\bar{g}(\tilde{\theta}_t)$. Thus, taking expectations causes this term to vanish.

- In the standard analysis of `TD`(0), while one does encounter such a bias term (under Markovian sampling), such a term features the true iterate $\theta_t$, and not its perturbed version $\tilde{\theta}_t$. This is where we again need to carefully account for the error between $\theta_t$ and $\tilde{\theta}_t$.

- In Appendix C, we derived a bound on the bias term by leveraging the uniform bounds on the memory variable in Lemma 12. Such uniform bounds were made possible via the projection step. Since we no longer have such a projection step at our disposal, we need an alternate proof technique.

In order to bound $\langle\tilde{\theta}_t - \theta^*, g_t(\tilde{\theta}_t) - \bar{g}(\tilde{\theta}_t)\rangle$, we will require a mixing time argument where we condition sufficiently into the past. This, in turn, will create the need to bound the drift $\|\tilde{\theta}_t - \tilde{\theta}_{t-\tau}\|$ of the perturbed iterate $\tilde{\theta}_t$; here, recall that $\tau$ is the mixing time. In the analysis of vanilla `TD`(0), the authors in Srikant & Ying (2019) show how such a drift term can be related to the distance to optimality of the (true) iterate at time $t$. The presence of the memory variable $e_t$ (that accounts for past errors) makes it hard to establish such a result for our setting. As such, we will now establish a different bound on the drift $\|\tilde{\theta}_t - \tilde{\theta}_{t-\tau}\|$ as a function of the maximum amplitude of our constructed Lyapunov function (in equation 69) over the interval $[t - \tau, t]$. In this context, we have the following key result (Lemma 1 in the main body of the paper).

**Lemma 18.** *(**Relating Drift to Past Shocks**) For `EF-TD`, the following is true $\forall t \geq \tau$:*

$$
\mathbb{E}[\|\tilde{\theta}_t - \tilde{\theta}_{t-\tau}\|^2] \leq 12\alpha^2\tau^2 \max_{t-\tau \leq \ell \leq t-1} \psi_\ell + 48\alpha^2\tau^2\sigma^2.
\tag{79}
$$

*Proof.* Starting from equation 68, observe that

$$
\begin{aligned}
\|\tilde{\theta}_{t+1} - \tilde{\theta}_t\| &\le \alpha\|g_t(\theta_t)\| \\
&\le \alpha \left( \|g_t(\theta_t) - g_t(\tilde{\theta}_t)\| + \|g_t(\tilde{\theta}_t)\| \right) \\
&\stackrel{(a)}{\le} 2\alpha \left( \|\theta_t - \tilde{\theta}_t\| + \|\tilde{\theta}_t\| + \sigma \right) \\
&\le 2\alpha \left( \alpha\|e_{t-1}\| + \|\tilde{\theta}_t\| + \sigma \right) \\
&\le 2\alpha \left( \alpha\|e_{t-1}\| + \tilde{d}_t + 2\sigma \right),
\end{aligned}
\tag{80}
$$

where (a) follows from Lemmas 6 and 8. We thus have

$$
\begin{aligned}
\mathbb{E}\left[\|\tilde{\theta}_{t+1} - \tilde{\theta}_t\|^2\right] &\le 12\alpha^2 \mathbb{E}\left[\alpha^2\|e_{t-1}\|^2 + \tilde{d}_t^2 + 4\sigma^2\right] \\
&= 12\alpha^2\psi_t + 48\alpha^2\sigma^2,
\end{aligned}
\tag{81}
$$

where for the first step, we used equation 19, and for the second, the definition of $\psi_t$ in equation 69. Appealing to equation 19 again, observe that

$$
\begin{aligned}
\mathbb{E}\left[\|\tilde{\theta}_t - \tilde{\theta}_{t-\tau}\|^2\right] &\le \tau \sum_{\ell=t-\tau}^{t-1} \mathbb{E}\left[\|\tilde{\theta}_{\ell+1} - \tilde{\theta}_\ell\|^2\right] \\
&\stackrel{equation\ 81}{\le} 12\alpha^2\tau \sum_{\ell=t-\tau}^{t-1} \left(\psi_\ell + 4\sigma^2\right) \\
&\le 12\alpha^2\tau^2 \max_{t-\tau\le\ell\le t-1} \psi_\ell + 48\alpha^2\tau^2\sigma^2,
\end{aligned}
\tag{82}
$$

which is the desired claim. $\qquad\square$

Interpreting $\psi_\ell$ as a "shock" from time-step $\ell$, Lemma 18 tells us that the drift of the perturbed iterate over the interval $[t-\tau, t]$ can be bounded above by the maximum shock over this interval (up to noise terms). Fortunately, the effect of this shock is dampened by the presence of the $O(\alpha^2)$ term multiplying it. Equipped with Lemma 18, we now proceed to bound the bias term.

**Lemma 19.** *(**Bounding the Bias**) Suppose Assumption 1 holds. Let the step-size $\alpha$ be such that $\alpha\tau \le 1/6$. For* `EF-TD`*, the following is then true $\forall t \ge \tau$:*

$$
\mathbb{E}\left[\langle\tilde{\theta}_t - \theta^*, g_t(\tilde{\theta}_t) - \bar{g}(\tilde{\theta}_t)\rangle\right] \le 31\alpha\tau\mathbb{E}\left[\tilde{d}_t^2\right] + 103\alpha\tau V_t + 454\alpha\tau\sigma^2,
\tag{83}
$$

*where*

$$
V_t \triangleq \max_{t-\tau\le\ell\le t-1} \psi_\ell.
$$

*Proof.* We start by decomposing the bias term $T = \langle\tilde{\theta}_t - \theta^*, g_t(\tilde{\theta}_t) - \bar{g}(\tilde{\theta}_t)\rangle$ as follows:

$$
T = \underbrace{\langle\tilde{\theta}_t - \tilde{\theta}_{t-\tau}, g_t(\tilde{\theta}_t) - \bar{g}(\tilde{\theta}_t)\rangle}_{T_1} + \underbrace{\langle\tilde{\theta}_{t-\tau} - \theta^*, g_t(\tilde{\theta}_t) - \bar{g}(\tilde{\theta}_t)\rangle}_{T_2}.
$$

To bound $T_1$, we note that

$$
\begin{aligned}
T_1 &\le \|\tilde{\theta}_t - \tilde{\theta}_{t-\tau}\|\|g_t(\tilde{\theta}_t) - \bar{g}(\tilde{\theta}_t)\| \\
&\le \frac{1}{2\alpha\tau}\|\tilde{\theta}_t - \tilde{\theta}_{t-\tau}\|^2 + \frac{\alpha\tau}{2}\|g_t(\tilde{\theta}_t) - \bar{g}(\tilde{\theta}_t)\|^2 \\
&\le \frac{1}{2\alpha\tau}\|\tilde{\theta}_t - \tilde{\theta}_{t-\tau}\|^2 + \alpha\tau\left(\|g_t(\tilde{\theta}_t)\|^2 + \|\bar{g}(\tilde{\theta}_t)\|^2\right) \\
&\stackrel{(a)}{\le} \frac{1}{2\alpha\tau}\|\tilde{\theta}_t - \tilde{\theta}_{t-\tau}\|^2 + 10\alpha\tau(\|\tilde{\theta}_t\|^2 + \sigma^2) \\
&\le \frac{1}{2\alpha\tau}\|\tilde{\theta}_t - \tilde{\theta}_{t-\tau}\|^2 + 10\alpha\tau(2\tilde{d}_t^2 + 3\sigma^2),
\end{aligned}
\tag{84}
$$

where for (a), we used Lemma 6 and equation 16. Taking expectations on both sides of the above inequality, and using Lemma 18, we obtain:

$$\mathbb{E}\left[T_1\right] \leq 20\alpha\tau\mathbb{E}\left[\tilde{d}_t^2\right] + 6\alpha\tau V_t + 54\alpha\tau\sigma^2. \tag{85}$$

Next, to bound $T_2$, we decompose it as follows:

$$T_2 = \underbrace{\langle\tilde{\theta}_{t-\tau} - \theta^*, g_t(\tilde{\theta}_{t-\tau}) - \bar{g}(\tilde{\theta}_{t-\tau})\rangle}_{(*)} + \underbrace{\langle\tilde{\theta}_{t-\tau} - \theta^*, g_t(\tilde{\theta}_t) - g_t(\tilde{\theta}_{t-\tau})\rangle}_{(**)} + \underbrace{\langle\tilde{\theta}_{t-\tau} - \theta^*, \bar{g}(\tilde{\theta}_{t-\tau}) - \bar{g}(\tilde{\theta}_t)\rangle}_{(***)}.$$

We now proceed to bound each of the three terms above. Observe:

$$\begin{aligned}
(**) &\leq \tilde{d}_{t-\tau}\|g_t(\tilde{\theta}_t) - g_t(\tilde{\theta}_{t-\tau})\| \\
&\overset{(a)}{\leq} 2\tilde{d}_{t-\tau}\|\tilde{\theta}_t - \tilde{\theta}_{t-\tau}\| \\
&\leq 2(\tilde{d}_t + \|\tilde{\theta}_t - \tilde{\theta}_{t-\tau}\|)\|\tilde{\theta}_t - \tilde{\theta}_{t-\tau}\| \\
&\overset{(b)}{\leq} 2\left(\sqrt{\alpha\tau}\tilde{d}_t + \frac{\|\tilde{\theta}_t - \tilde{\theta}_{t-\tau}\|}{\sqrt{\alpha\tau}}\right)^2 \\
&\leq 4\left(\alpha\tau\tilde{d}_t^2 + \frac{\|\tilde{\theta}_t - \tilde{\theta}_{t-\tau}\|^2}{\alpha\tau}\right),
\end{aligned} \tag{86}$$

where for (a), we used the Lipschitz property in Lemma 8, and for (b), we used the fact that $\alpha\tau \leq 1$. Taking expectations on each side of the above inequality and appealing to Lemma 18, we obtain

$$\mathbb{E}\left[(**)\right] \leq 4\alpha\tau\mathbb{E}\left[\tilde{d}_t^2\right] + 48\alpha\tau V_t + 192\alpha\tau\sigma^2. \tag{87}$$

Using Lemma 7 and the same arguments as above, one can establish the exact same bound on $\mathbb{E}\left[(***)\right]$ as in equation 87. Before proceeding to bound $(*)$, we make the observation that $\tilde{\theta}_t$ inherits its randomness from all the Markov data tuples up to time $t-1$, i.e., from $\{X_k\}_{k=0}^{t-1}$. We now have:

$$\begin{aligned}
\mathbb{E}\left[(*)\right] &= \mathbb{E}\left[\langle\tilde{\theta}_{t-\tau} - \theta^*, g_t(\tilde{\theta}_{t-\tau}) - \bar{g}(\tilde{\theta}_{t-\tau})\rangle\right] \\
&= \mathbb{E}\left[\mathbb{E}\left[\langle\tilde{\theta}_{t-\tau} - \theta^*, g_t(\tilde{\theta}_{t-\tau}) - \bar{g}(\tilde{\theta}_{t-\tau})\rangle|\tilde{\theta}_{t-\tau}, X_{t-\tau}\right]\right] \\
&= \mathbb{E}\left[\langle\tilde{\theta}_{t-\tau} - \theta^*, \mathbb{E}\left[g_t(\tilde{\theta}_{t-\tau}) - \bar{g}(\tilde{\theta}_{t-\tau})|\tilde{\theta}_{t-\tau}, X_{t-\tau}\right]\rangle\right] \\
&\leq \mathbb{E}\left[\tilde{d}_{t-\tau}\|\mathbb{E}\left[g_t(\tilde{\theta}_{t-\tau}) - \bar{g}(\tilde{\theta}_{t-\tau})|\tilde{\theta}_{t-\tau}, X_{t-\tau}\right]\|\right] \\
&\overset{(a)}{\leq} \alpha\mathbb{E}\left[\tilde{d}_{t-\tau}\left(\|\tilde{\theta}_{t-\tau}\| + 1\right)\right] \\
&\leq \alpha\mathbb{E}\left[\tilde{d}_{t-\tau}\left(\|\tilde{\theta}_{t-\tau} - \tilde{\theta}_t\| + \tilde{d}_t + \|\theta^*\| + 1\right)\right] \\
&\leq \alpha\mathbb{E}\left[(\tilde{d}_t + \|\tilde{\theta}_t - \tilde{\theta}_{t-\tau}\|)\left(\|\tilde{\theta}_{t-\tau} - \tilde{\theta}_t\| + \tilde{d}_t + \|\theta^*\| + 1\right)\right] \\
&\leq \alpha\mathbb{E}\left[(\tilde{d}_t + \|\tilde{\theta}_t - \tilde{\theta}_{t-\tau}\|)\left(\|\tilde{\theta}_{t-\tau} - \tilde{\theta}_t\| + \tilde{d}_t + 2\sigma\right)\right] \\
&\overset{(b)}{\leq} \alpha\tau\mathbb{E}\left[(\tilde{d}_t + \|\tilde{\theta}_t - \tilde{\theta}_{t-\tau}\| + 2\sigma)^2\right] \\
&\leq 3\alpha\tau\mathbb{E}\left[\tilde{d}_t^2\right] + 3\alpha\tau\mathbb{E}\left[\|\tilde{\theta}_t - \tilde{\theta}_{t-\tau}\|^2\right] + 12\alpha\tau\sigma^2 \\
&\overset{(c)}{\leq} 3\alpha\tau\mathbb{E}\left[\tilde{d}_t^2\right] + \alpha\tau V_t + 16\alpha\tau\sigma^2.
\end{aligned} \tag{88}$$

In the above steps, (a) follows from the mixing property in Definition 1, (b) follows from the fact that $\tau \geq 1$, and (c) follows by invoking Lemma 18 and simplifying using $\alpha\tau \leq 1/6$. Combining the above bound with that in equation 87, we conclude that

$$E[T_2] \leq 11\alpha\tau\mathbb{E}\left[\tilde{d}_t^2\right] + 97\alpha\tau V_t + 400\alpha\tau\sigma^2.$$

Combining the above bound with that in equation 85 completes the proof. □

We now have all the pieces needed to prove Theorem 1.

*Proof.* (**Proof of Theorem 1**) We break up the proof into two parts. In the first step, we establish a recursion for our potential function $\psi_t$. In the second step, we analyze this recursion by making a connection to the analysis of the Incremental Aggregated Gradient (`IAG`) algorithm in Gurbuzbalaban et al. (2017).

**Step 1: Establishing a Recursion for $\psi_t$.** Combining the bound on the bias term in Lemma 19 with Lemma 16, and simplifying using $\tau \geq 1$, we obtain the following inequality $\forall t \geq \tau$:

$$\mathbb{E}\left[\tilde{d}_{t+1}^2\right] \leq \left(1 - \alpha\omega(1-\gamma) + 86\alpha^2\tau\right)\mathbb{E}\left[\tilde{d}_t^2\right] + 206\alpha^2\tau V_t + \frac{6\alpha^3}{\omega(1-\gamma)}\mathbb{E}\left[\|e_{t-1}\|^2\right] + 940\alpha^2\tau^2\sigma^2.$$

Combining the above display with the bound on the memory variable in Lemma 17, and using the definition of the potential function $\psi_t$, we then obtain $\forall t \geq \tau$:

$$\psi_{t+1} \leq \underbrace{\left(1 - \alpha\omega(1-\gamma) + 86\alpha^2\tau + 64\alpha^2\delta\right)}_{A_1}\mathbb{E}\left[\tilde{d}_t^2\right] + 206\alpha^2\tau V_t$$
$$+ \alpha^2\underbrace{\left(1 - \frac{1}{2\delta} + 16\alpha^2\delta + \frac{6\alpha}{\omega(1-\gamma)}\right)}_{A_2}\mathbb{E}\left[\|e_{t-1}\|^2\right] + \alpha^2(940\tau + 96\delta)\sigma^2. \tag{89}$$

Our immediate goal is to pick $\alpha$ such that $\max\{A_1, A_2\} < 1$. Accordingly, it is easy to check that if

$$\alpha \leq \frac{\omega(1-\gamma)}{344\tau} \quad \text{and} \quad \alpha \leq \frac{\omega(1-\gamma)}{256\delta}, \tag{90}$$

then

$$A_1 \leq 1 - \frac{\alpha\omega(1-\gamma)}{2}.$$

It is also easily verified that with the choice of step-size in equation 90, the following hold:

$$16\alpha^2\delta \leq \frac{1}{8\delta} \quad \text{and} \quad \frac{6\alpha}{\omega(1-\gamma)} \leq \frac{1}{8\delta},$$

implying that

$$A_2 \leq 1 - \frac{1}{4\delta}.$$

Finally, note that based on the choice of $\alpha$ in equation 90, we have

$$1 - \frac{1}{4\delta} \leq 1 - \frac{\alpha\omega(1-\gamma)}{2}.$$

Combining all the above observations, we obtain that for all $t \geq \tau$:

$$\boxed{\psi_{t+1} \leq \left(1 - \frac{\alpha\omega(1-\gamma)}{2}\right)\psi_t + 206\alpha^2\tau\left(\max_{t-\tau \leq \ell \leq t}\psi_\ell\right) + O(\alpha^2(\tau+\delta)\sigma^2).} \tag{91}$$

If the second term (i.e., the "max" term) in the above bound were absent, one could easily unroll the resulting recursion and argue linear convergence to a noise ball. In what follows, we will show that one can still establish such linear convergence guarantees for equation 91.

**Step 2: Analyzing equation 91 via a connection to the `IAG` algorithm.** To see how we can analyze equation 91, we take a quick detour and recap the basic idea behind the `IAG` method for finite-sum optimization. Say we want to minimize

$$f(x) = \frac{1}{M}\sum_{i\in[M]} f_i(x),$$

where each component function is smooth. The `IAG` method does so in a computationally-efficient manner by processing each of the component functions one at a time in a *deterministic* order, and crucially, by maintaining a *memory* of the most recent gradient values of each of the component functions. This memory introduces certain *delayed* gradient terms in the update rule. In Gurbuzbalaban et al. (2017), it was shown that the presence of these delayed terms leads to a recursion of the form in equation 91. This turns out to be the key observation needed to complete our analysis. In particular, we recall an important lemma from Feyzmahdavian et al. (2014) (used in Gurbuzbalaban et al. (2017)) that will help us reason about equation 91.

**Lemma 20.** *Let $\{G_t\}$ be a sequence of non-negative real numbers satisfying*

$$G_{t+1} \leq pG_t + q \max_{(t-\tau_t)_+ \leq \ell \leq t} G_\ell + r, \ \ t \in \mathbb{N},$$

*for some non-negative constants $p, q$, and $r$. Here, for any real scalar $x$, we use the notation $(x)_+ = \max\{x, 0\}$. If $p + q < 1$ and $0 \leq \tau_t \leq \tau_{max}, \forall t \geq 0$ for some positive constant $\tau_{max}$, then*

$$G_t \leq \rho^t G_0 + \varepsilon, \forall t \geq 0,$$

*where*

$$\rho = (p+q)^{\frac{1}{1+\tau_{max}}}, \ \ and \ \ \varepsilon = \frac{r}{(1-p-q)}.$$

Comparing equation 91 to Lemma 20, we note that for us:

$$p = 1 - \frac{\alpha\omega(1-\gamma)}{2}, q = 206\alpha^2\tau, r = O(\alpha^2(\tau+\delta)\sigma^2), \ \ and \ \ \tau_t = \tau,$$

where $\tau$ is the mixing time. Now suppose $\alpha$ is chosen such that

$$\alpha \leq \frac{\omega(1-\gamma)}{824\tau}. \tag{92}$$

Then, we immediately obtain that

$$p + q = 1 - \frac{\alpha\omega(1-\gamma)}{2} + 206\alpha^2\tau < 1 - \frac{\alpha\omega(1-\gamma)}{4}.$$

Setting $C_1 = \max_{0 \leq k \leq \tau} \psi_k$, and appealing to Lemma 20 then yields the following $\forall T \geq \tau$:

$$\psi_T \leq C_1 \left(1 - \frac{\alpha\omega(1-\gamma)}{8\tau}\right)^{T-\tau} + O\left(\frac{\alpha(\tau+\delta)\sigma^2}{\omega(1-\gamma)}\right),$$

where we used the facts that $(1-x)^a \leq 1 - ax$ for $x, a \in [0, 1]$, and $\tau \geq 1$, to simplify the final expression. Next, note that

$$\begin{aligned}
\mathbb{E}\left[\|\theta_T - \theta^*\|^2\right] &= \mathbb{E}\left[\|\theta_T - \tilde{\theta}_T + \tilde{\theta}_T - \theta^*\|^2\right] \\
&\leq 2\mathbb{E}\left[\|\tilde{\theta}_T - \theta^*\|^2\right] + 2\mathbb{E}\left[\|\theta_T - \tilde{\theta}_T\|^2\right] \\
&= 2\mathbb{E}\left[\tilde{d}_t^2\right] + 2\alpha^2\mathbb{E}\left[\|e_{T-1}\|^2\right] \\
&= 2\psi_T.
\end{aligned} \tag{93}$$

We conclude that $\forall T \geq \tau$,

$$\mathbb{E}\left[r_T^2\right] \leq 2C_1 \left(1 - \frac{\alpha\omega(1-\gamma)}{8\tau}\right)^{T-\tau} + O\left(\frac{\alpha(\tau+\delta)\sigma^2}{\omega(1-\gamma)}\right).$$

Furthermore, from the requirements on the step-size $\alpha$ in equations 90 and 92, we note that for the above inequality to hold, it suffices for $\alpha$ to satisfy:

$$\boxed{\alpha \leq \frac{\omega(1-\gamma)}{824\max\{\tau, \delta\}}.}$$

The only thing that remains to be shown is that $C_1 = \max_{0 \le k \le \tau} \psi_k = O(d_0^2 + \sigma^2)$ based on our choice of step-size above. This follows from straightforward calculations that we provide below for completeness.

From equation 68, we have

$$
\begin{aligned}
\tilde{d}_{t+1}^2 &= \tilde{d}_t^2 + 2\alpha\langle\tilde{\theta}_t - \theta^*, g_t(\theta_t)\rangle + \alpha^2\|g_t(\theta_t)\|^2 \\
&\le (1+\alpha)\tilde{d}_t^2 + 2\alpha\|g_t(\theta_t)\|^2 \\
&\le (1+49\alpha)\tilde{d}_t^2 + 48\alpha^3\|e_{t-1}\|^2 + 64\alpha\sigma^2,
\end{aligned}
\tag{94}
$$

where in the last step, we used equation 74. Combining the above bound with Lemma 17, we obtain the following inequality for all $t \ge 0$:

$$
\psi_{t+1} \le (1 + 49\alpha + 64\alpha^2\delta)\mathbb{E}\left[\tilde{d}_t^2\right] + \alpha^2\left(1 - \frac{1}{2\delta} + 16\alpha^2\delta + 48\alpha\right)\mathbb{E}\left[\|e_{t-1}\|^2\right] + (64\alpha + 96\alpha^2\delta)\sigma^2.
\tag{95}
$$

Using the fact that $\alpha\delta \le 1/824$, we can simplify the above display to obtain

$$
\psi_{t+1} \le (1 + 50\alpha)\psi_t + 65\alpha\sigma^2, \forall t \ge 0.
\tag{96}
$$

Unrolling the above inequality and using $e_{-1} = 0$, we have that for $0 \le k \le \tau$:

$$
\psi_k \le (1 + 50\alpha)^k d_0^2 + 65\alpha\sigma^2 \sum_{j=0}^{k-1}(1 + 50\alpha)^j.
\tag{97}
$$

Now since $(1 + x) \le e^x, \forall x \in \mathbb{R}$, and $\alpha \le 1/(200\tau)$, note that $(1 + 50\alpha)^k \le (1 + 50\alpha)^\tau \le e^{0.25} \le 2$. Thus, for $0 \le k \le \tau$, we have

$$
\psi_k \le 2d_0^2 + 130\alpha\tau\sigma^2 \le 2d_0^2 + \sigma^2,
$$

where in the last step, we used $130\alpha\tau \le 1$. Thus, $C_1 = \max_{0 \le k \le \tau} \psi_k = O(d_0^2 + \sigma^2)$. This concludes the proof. $\square$

We conclude this section with a note on the proof of Theorem 2.

**Proof of Theorem 2**: A careful inspection of the proof of Theorem 1 reveals that we never explicitly used the fact that the TD update direction is an affine function of the parameter $\theta$. This was done on purpose to provide a unified analysis framework to not only reason about linear stochastic approximation schemes with error-feedback, but also their nonlinear counterparts. As such, under Assumptions 2 and 3, the analysis for the nonlinear setting in Section 5 follows *exactly* the same steps as the proof of Theorem 1. All one needs to do is replace $\omega(1 - \gamma)$ in the proof of Theorem 1 with $\beta$, where $\beta$ is as in Assumption 3. Everything else essentially remains the same. We thus omit routine details here.

# E   Analysis of Multi-Agent `EF-TD`: Proof of Theorem 3

In this section, we will analyze the multi-agent version of `EF-TD` outlined in Algorithm 2. The main technical challenge relative to the single-agent analysis conducted in Appendices C and D is in establishing the *linear speedup property* w.r.t. the number of agents $M$ under the Markovian sampling assumption. As we mentioned earlier in the paper, this turns out to be highly non-trivial even in the absence of compression and error-feedback. The only work that establishes such a speedup (under Markovian sampling) is Khodadadian et al. (2022), where the authors use the framework of Generalized Moreau Envelopes to perform their analysis. Although the analysis in Khodadadian et al. (2022) is elegant, it is quite involved, and it is unclear whether their framework can accommodate the error-feedback mechanism. Moreover, the analysis in Khodadadian et al. (2022) leads to a sub-optimal $O(\tau^2)$ dependence on the mixing time $\tau$ in the main noise/variance term. In light of the above discussion, we will provide a different analysis in this section that:

- Departs from the Moreau Envelope approach in Khodadadian et al. (2022),

- Achieves the optimal $O(\tau)$ dependence on the mixing time $\tau$ in the dominant noise term,

- Establishes the desired linear speedup property, and

- Shows that the effect of the distortion parameter $\delta$ can be relegated to a higher-order term.

Crucial to achieving all of the above desiderata are a few novel ingredients in the proof that we now outline. First, we will require a more refined Lyapunov function than we used earlier. Before we introduce this Lyapunov function, let us define a couple of objects:

$$\bar{e}_t \triangleq \frac{1}{M} \sum_{i \in [M]} e_{i,t}, \text{ and } \bar{h}_t \triangleq \frac{1}{M} \sum_{i \in [M]} h_{i,t}.$$

Next, let us define a perturbed iterate for this setting as $\tilde{\theta}_t \triangleq \theta_t + \alpha \bar{e}_{t-1}$. The potential function we employ is as follows:

$$\Xi_t \triangleq \mathbb{E}\left[\|\tilde{\theta}_t - \theta^*\|^2\right] + C\alpha^3 E_{t-1}, \text{ where } E_t \triangleq \frac{1}{M}\mathbb{E}\left[\sum_{i=1}^{M}\|e_{i,t}\|^2\right], \tag{98}$$

and $C > 1$ is a constant that will be chosen by us later. Compared to the Lyapunov function $\psi_t$ in equation 69 that we used for the single-agent setting, $\Xi_t$ differs in two ways. First, it incorporates the memory dynamics of *all* the agents. A more subtle difference, however, stems from the fact that the second term of $\Xi_t$ is scaled by $\alpha^3$, and not $\alpha^2$ (as in equation 69). This higher-order dependence on $\alpha$ will serve a twofold purpose: (i) help to shift the effect of $\delta$ to a higher-order term, and (ii) *partially* help in achieving the linear-speedup property. Unfortunately, however, the new Lyapunov function on its own will not suffice in terms of achieving the linear speedup effect completely. For this, we need a careful way to bound the norm of the average TD direction defined below:

$$z_t(\theta) \triangleq \frac{1}{M} \sum_{i \in [M]} g_{i,t}(\theta), \forall \theta \in \mathbb{R}^K. \tag{99}$$

An immediate way to bound $z_t(\theta)$ is to appeal to the bound on the norm of the TD update direction in Lemma 6. Indeed, this is what we did while proving Theorem 1, and this is also the typical approach for bounding norms of TD update directions in the centralized setting (Srikant & Ying, 2019; Chen et al., 2019). The issue with adopting this approach in the multi-agent analysis is that it completely ignores the fact that the observations of the agents are *statistically independent*. As such, following this route will not lead to any "variance-reduction" effect (key to the linear speedup property). At the same time, it is important to realize here that while the Markov data tuples are independent across agents, for any fixed agent $i$, $\{X_{i,t}\}$ comes from a single Markov chain. This makes it trickier to analyze the variance of $z_t(\theta_t)$. Via a careful mixing time argument, our next result (Lemma 2 in the main body of the paper) shows how this can be done. Before stating this result, we remind the reader that we have used $\tau$ as a shorthand for $\tau_\epsilon$, with $\epsilon = \alpha^2$. While a precision of $\epsilon = \alpha$ sufficed for all our prior single-agent analyses, we will need $\epsilon = \alpha^2$ for the MARL case (to create higher-order noise terms in $\alpha$).

**Lemma 21.** *(**Controlling the norm of the Average TD Direction**) Suppose Assumption 1 holds. For Algorithm 2, the following are then true $\forall t \geq \tau$:*

$$\mathbb{E}\left[\|z_t(\theta_t)\|^2\right] \leq 8\mathbb{E}\left[d_t^2\right] + \left(\frac{32}{M} + 8\alpha^4\right)\sigma^2, \quad \text{where } d_t = \|\theta_t - \theta^*\|, \quad \text{and} \tag{100}$$

$$\mathbb{E}\left[\|z_t(\theta_t)\|^2\right] \leq 16\mathbb{E}\left[\tilde{d}_t^2\right] + 16\alpha^2 E_{t-1} + \left(\frac{32}{M} + 8\alpha^4\right)\sigma^2, \quad \text{where } \tilde{d}_t = \|\tilde{\theta}_t - \theta^*\|. \tag{101}$$

*Proof.* Let us start with the following set of observations:

$$
\begin{aligned}
\|z_t(\theta_t)\|^2 &= \frac{1}{M^2}\left\|\sum_{i=1}^{M} g_{i,t}(\theta_t)\right\|^2 \\
&= \frac{1}{M^2}\left\|\sum_{i=1}^{M}(g_{i,t}(\theta_t) - g_{i,t}(\theta^*)) + \sum_{i=1}^{M} g_{i,t}(\theta^*)\right\|^2 \\
&\leq \frac{2}{M^2}\left\|\sum_{i=1}^{M}(g_{i,t}(\theta_t) - g_{i,t}(\theta^*))\right\|^2 + \frac{2}{M^2}\left\|\sum_{i=1}^{M} g_{i,t}(\theta^*)\right\|^2 \\
&\leq \frac{2}{M}\sum_{i=1}^{M}\|g_{i,t}(\theta_t) - g_{i,t}(\theta^*)\|^2 + \frac{2}{M^2}\left\|\sum_{i=1}^{M} g_{i,t}(\theta^*)\right\|^2 \\
&\leq 8d_t^2 + \frac{2}{M^2}\left\|\sum_{i=1}^{M} g_{i,t}(\theta^*)\right\|^2,
\end{aligned}
\tag{102}
$$

where in the last step, we used the Lipschitz property in Lemma 8. Next, to bound the second term in the above display, we split it into two parts as follows.

$$\left\|\sum_{i=1}^{M} g_{i,t}(\theta^*)\right\|^2 = \underbrace{\sum_{i=1}^{M}\|g_{i,t}(\theta^*)\|^2}_{(*)} + \underbrace{\sum_{\substack{i,j=1 \\ i\neq j}}^{M}\langle g_{i,t}(\theta^*), g_{j,t}(\theta^*)\rangle}_{(**)}.$$

To bound $(*)$, we simply use Lemma 6 and the fact that $\|\theta^*\| \leq \sigma$ to conclude that

$$\sum_{i=1}^{M}\|g_{i,t}(\theta^*)\|^2 \leq 8M(\|\theta^*\|^2 + \sigma^2) \leq 16M\sigma^2.$$

Now to bound $(**)$, let us zoom in on a particular cross-term, and write it out in a way that highlights the sources of randomness. Accordingly, consider the term

$$\mathcal{T} = \langle g_{i,t}(\theta^*), g_{j,t}(\theta^*)\rangle = \langle g(X_{i,t}, \theta^*), g(X_{j,t}, \theta^*)\rangle.$$

Since $\theta^*$ is deterministic, we note that the randomness in $\mathcal{T}$ originates from the Markov data samples $X_{i,t}$ and $X_{j,t}$. Moreover, since $X_{i,t}$ and $X_{j,t}$ are independent for $i \neq j$, we have

$$\mathbb{E}[\mathcal{T}] = \langle \mathbb{E}[g(X_{i,t}, \theta^*)], \mathbb{E}[g(X_{j,t}, \theta^*)]\rangle.$$

Now if $X_{i,t}$ and $X_{j,t}$ were sampled i.i.d. from the stationary distribution $\pi$, each of the two expectations within the above inner-product would have amounted to $\bar{g}(\theta^*) = 0$. Thus, the cross-terms would have vanished. Since for each agent $i$, $X_{i,t}$ comes from a Markov chain, these expectations do not, unfortunately,

vanish any longer. Nonetheless, we now show that for $t \geq \tau$, one can still make the cross-terms suitably "small" by exploiting the mixing property in Definition 1. Observe:

$$
\begin{aligned}
\mathbb{E}\left[\mathcal{T}\right] &= \langle \mathbb{E}\left[g(X_{i,t}, \theta^*)\right], \mathbb{E}\left[g(X_{j,t}, \theta^*)\right]\rangle \\
&\overset{(a)}{=} \langle \mathbb{E}\left[\mathbb{E}\left[g(X_{i,t}, \theta^*)|X_{i,t-\tau}\right] - \bar{g}(\theta^*)\right], \mathbb{E}\left[\mathbb{E}\left[g(X_{j,t}, \theta^*)|X_{j,t-\tau}\right] - \bar{g}(\theta^*)\right]\rangle \\
&\overset{(b)}{\leq} \|\mathbb{E}\left[\mathbb{E}\left[g(X_{i,t}, \theta^*)|X_{i,t-\tau}\right] - \bar{g}(\theta^*)\right]\| \times \|\mathbb{E}\left[\mathbb{E}\left[g(X_{j,t}, \theta^*)|X_{j,t-\tau}\right] - \bar{g}(\theta^*)\right]\| \\
&\overset{(c)}{\leq} \mathbb{E}\left[\|\mathbb{E}\left[g(X_{i,t}, \theta^*)|X_{i,t-\tau}\right] - \bar{g}(\theta^*)\|\right] \times \mathbb{E}\left[\|\mathbb{E}\left[g(X_{j,t}, \theta^*)|X_{j,t-\tau}\right] - \bar{g}(\theta^*)\|\right] \\
&\overset{(d)}{\leq} \alpha^4(\|\theta^*\| + 1)^2 \\
&\leq 4\sigma^2\alpha^4.
\end{aligned}
\tag{103}
$$

In the above steps, (a) follows from the tower property of expectation in conjunction with $\bar{g}(\theta^*) = 0$. For (b), we used the Cauchy-Schwarz inequality, and (c) follows from Jensen's inequality. Finally, (d) is a consequence of the mixing property in Definition 1. We immediately conclude:

$$
\mathbb{E}\left[(**)\right] \leq 4M^2\sigma^2\alpha^4.
$$

Putting the above pieces together and simplifying leads to equation 100. To go from equation 100 to equation 101, we simply note that

$$
\mathbb{E}\left[d_t^2\right] \leq 2\mathbb{E}\left[\tilde{d}_t^2\right] + 2\alpha^2\mathbb{E}\left[\|\tilde{\theta}_t - \theta_t\|^2\right] = 2\mathbb{E}\left[\tilde{d}_t^2\right] + 2\alpha^2\mathbb{E}\left[\|\bar{e}_{t-1}\|^2\right] \leq 2\mathbb{E}\left[\tilde{d}_t^2\right] + 2\alpha^2 E_{t-1}.
\tag{104}
$$

This concludes the proof. □

Essentially, Lemma 21 shows how the variance of the average TD update direction can be scaled down by a factor of $M$, up to a $O(\alpha^4)$ term. As we shall soon see, this result will play a key role in our subsequent analysis of Algorithm 2. We now proceed to derive a multi-agent version of Lemma 16.

**Lemma 22.** *Suppose Assumption 1 holds and the step-size $\alpha$ satisfies $\alpha \leq 1/16$. For Algorithm 2, the following bound then holds for $\forall t \geq \tau$:*

$$
\mathbb{E}\left[\tilde{d}_{t+1}^2\right] \leq \left(1 - \alpha\omega(1-\gamma) + 16\alpha^2\right)\mathbb{E}\left[\tilde{d}_t^2\right] + \frac{5\alpha^3}{\omega(1-\gamma)}E_{t-1} + 2\alpha\mathbb{E}\left[A\right] + \alpha^2\left(\frac{32}{M} + 8\alpha^4\right)\sigma^2,
\tag{105}
$$

*where $A = \langle \tilde{\theta}_t - \theta^*, z_t(\tilde{\theta}_t) - \bar{g}(\tilde{\theta}_t)\rangle$.*

*Proof.* Simple calculations reveal that

$$
\tilde{\theta}_{t+1} = \tilde{\theta}_t + \alpha\left(\frac{1}{M}\sum_{i\in[M]} g_{i,t}(\theta_t)\right) = \tilde{\theta}_t + \alpha z_t(\theta_t).
\tag{106}
$$

Subtracting $\theta^*$ from each side of the above equation and then squaring both sides yields:

$$
\begin{aligned}
\tilde{d}_{t+1}^2 &= \tilde{d}_t^2 + 2\alpha\langle\tilde{\theta}_t - \theta^*, z_t(\theta_t)\rangle + \alpha^2\|z_t(\theta_t)\|^2 \\
&= \tilde{d}_t^2 + \underbrace{2\alpha\langle\tilde{\theta}_t - \theta^*, z_t(\tilde{\theta}_t)\rangle}_{(*)} + \underbrace{2\alpha\langle\tilde{\theta}_t - \theta^*, z_t(\theta_t) - z_t(\tilde{\theta}_t)\rangle}_{(**)} + \underbrace{\alpha^2\|z_t(\theta_t)\|^2}_{(***)}.
\end{aligned}
\tag{107}
$$

We now have:

$$
\begin{aligned}
(*) &= 2\alpha\langle\tilde{\theta}_t - \theta^*, \bar{g}(\tilde{\theta}_t)\rangle + 2\alpha\langle\tilde{\theta}_t - \theta^*, z_t(\tilde{\theta}_t) - \bar{g}(\tilde{\theta}_t)\rangle \\
&\leq -2\alpha\omega(1-\gamma)\tilde{d}_t^2 + 2\alpha\langle\tilde{\theta}_t - \theta^*, z_t(\tilde{\theta}_t) - \bar{g}(\tilde{\theta}_t)\rangle,
\end{aligned}
\tag{108}
$$

where in second step, we invoked Lemmas 3 and 4. To bound $(**)$, we proceed as follows:

$$
\begin{aligned}
(**) &\leq 4\alpha \tilde{d}_t \|\theta_t - \tilde{\theta}_t\| \\
&\leq \frac{2\alpha}{\eta} \tilde{d}_t^2 + 2\alpha\eta \|\theta_t - \tilde{\theta}_t\|^2 \\
&= \frac{2\alpha}{\eta} \tilde{d}_t^2 + 2\alpha^3 \eta \|\bar{e}_{t-1}\|^2 \\
&\leq \frac{2\alpha}{\eta} \tilde{d}_t^2 + 2\alpha^3 \eta E_{t-1},
\end{aligned}
\tag{109}
$$

where $\eta > 0$ is a constant to be decided shortly. In the first step above, we used the fact that since each $g_{i,t}(\theta)$ is 2-Lipschitz (see Lemma 8), the definition of $z_t(\theta)$ in equation 99 implies that $z_t(\theta)$ is also 2-Lipschitz. To bound $(***)$, we directly use Lemma 21. Combining the above bounds, and simplifying by setting $\eta = \frac{2}{\omega(1-\gamma)}$ and using $\alpha \leq 1/16$ leads to the desired claim. $\qquad\square$

In what follows, we will focus on bounding the bias term $A = \langle \tilde{\theta}_t - \theta^*, z_t(\tilde{\theta}_t) - \bar{g}(\tilde{\theta}_t) \rangle$ by following the same high-level steps as in the proof of Theorem 1. The key difference, however, will come from invoking Lemma 21 instead of Lemma 6. We start with a bound on the drift $\|\tilde{\theta}_t - \tilde{\theta}_{t-\tau}\|$.

**Lemma 23.** *Suppose Assumption 1 holds. Then for Algorithm 2, we have the following bound $\forall t \geq 2\tau$:*

$$
\mathbb{E}[\|\tilde{\theta}_t - \tilde{\theta}_{t-\tau}\|^2] \leq \alpha^2 \tau^2 \max_{t-\tau \leq \ell \leq t-1} G_\ell, \quad \text{where } G_\ell \triangleq 16\mathbb{E}\left[\tilde{d}_\ell^2\right] + 16\alpha^2 E_{\ell-1} + \left(\frac{32}{M} + 8\alpha^4\right)\sigma^2.
\tag{110}
$$

*Proof.* The proof is a direct application of Lemma 21. Indeed, notice that

$$
\begin{aligned}
\mathbb{E}\left[\|\tilde{\theta}_t - \tilde{\theta}_{t-\tau}\|^2\right] &\leq \tau \sum_{\ell=t-\tau}^{t-1} \mathbb{E}\left[\|\tilde{\theta}_{\ell+1} - \tilde{\theta}_\ell\|^2\right] \\
&\overset{\text{equation 106}}{\leq} \alpha^2 \tau \sum_{\ell=t-\tau}^{t-1} \mathbb{E}\left[\|z_\ell(\theta_\ell)\|^2\right] \\
&\overset{\text{equation 101}}{\leq} \alpha^2 \tau^2 \max_{t-\tau \leq \ell \leq t-1} G_\ell,
\end{aligned}
\tag{111}
$$

which is the desired claim. In the last step, we invoked Lemma 21 by noting that since $t \geq 2\tau$, we have that $\ell \geq \tau$ in the above steps - a requirement for using Lemma 21. $\qquad\square$

Equipped with Lemmas 21 and 23, we now proceed to bound the bias term following steps similar in spirit to the proof of Lemma 19.

**Lemma 24.** *Suppose Assumption 1 holds. Let the step-size $\alpha$ be such that $\alpha\tau \leq 1/3$. For Algorithm 2, the following is then true $\forall t \geq 2\tau$:*

$$
\mathbb{E}\left[\langle \tilde{\theta}_t - \theta^*, z_t(\tilde{\theta}_t) - \bar{g}(\tilde{\theta}_t) \rangle\right] \leq 13\alpha\tau \mathbb{E}\left[\tilde{d}_t^2\right] + 11\alpha\tau \max_{t-\tau \leq \ell \leq t} G_\ell + 12\alpha^2\sigma^2.
\tag{112}
$$

*Proof.* We start by decomposing the bias term $\langle \tilde{\theta}_t - \theta^*, z_t(\tilde{\theta}_t) - \bar{g}(\tilde{\theta}_t) \rangle$ as follows:

$$
A = \underbrace{\langle \tilde{\theta}_t - \tilde{\theta}_{t-\tau}, z_t(\tilde{\theta}_t) - \bar{g}(\tilde{\theta}_t) \rangle}_{A_1} + \underbrace{\langle \tilde{\theta}_{t-\tau} - \theta^*, z_t(\tilde{\theta}_t) - \bar{g}(\tilde{\theta}_t) \rangle}_{A_2}.
$$

To bound $A_1$, we note that

$$
\begin{aligned}
A_1 &\leq \|\tilde{\theta}_t - \tilde{\theta}_{t-\tau}\|\|z_t(\tilde{\theta}_t) - \bar{g}(\tilde{\theta}_t)\| \\
&\leq \frac{1}{2\alpha\tau}\|\tilde{\theta}_t - \tilde{\theta}_{t-\tau}\|^2 + \frac{\alpha\tau}{2}\|z_t(\tilde{\theta}_t) - \bar{g}(\tilde{\theta}_t)\|^2 \\
&\leq \frac{1}{2\alpha\tau}\|\tilde{\theta}_t - \tilde{\theta}_{t-\tau}\|^2 + \alpha\tau\left(\|z_t(\tilde{\theta}_t)\|^2 + \|\bar{g}(\tilde{\theta}_t)\|^2\right) \\
&\leq \frac{1}{2\alpha\tau}\|\tilde{\theta}_t - \tilde{\theta}_{t-\tau}\|^2 + \alpha\tau\left(\|z_t(\tilde{\theta}_t)\|^2 + 4\tilde{d}_t^2\right),
\end{aligned}
\tag{113}
$$

where the last step follows from Lemmas 3 and 5. Taking expectations on both sides of the above inequality, and using Lemmas 21 and 23 yields:

$$
\begin{aligned}
\mathbb{E}[A_1] &\leq 4\alpha\tau\mathbb{E}\left[\tilde{d}_t^2\right] + \frac{\alpha\tau}{2}\max_{t-\tau\leq\ell\leq t-1}G_\ell + \alpha\tau G_t \\
&\leq 4\alpha\tau\mathbb{E}\left[\tilde{d}_t^2\right] + 2\alpha\tau\max_{t-\tau\leq\ell\leq t}G_\ell.
\end{aligned}
\tag{114}
$$

Next, to bound $A_2$, we decompose it as follows:

$$
A_2 = \underbrace{\langle\tilde{\theta}_{t-\tau} - \theta^*, z_t(\tilde{\theta}_{t-\tau}) - \bar{g}(\tilde{\theta}_{t-\tau})\rangle}_{(*)} + \underbrace{\langle\tilde{\theta}_{t-\tau} - \theta^*, z_t(\tilde{\theta}_t) - z_t(\tilde{\theta}_{t-\tau})\rangle}_{(**)} + \underbrace{\langle\tilde{\theta}_{t-\tau} - \theta^*, \bar{g}(\tilde{\theta}_{t-\tau}) - \bar{g}(\tilde{\theta}_t)\rangle}_{(***)}.
$$

We now proceed to bound each of the three terms above. Let us start by observing that

$$
\begin{aligned}
(**) &\leq \tilde{d}_{t-\tau}\|z_t(\tilde{\theta}_t) - z_t(\tilde{\theta}_{t-\tau})\| \\
&\overset{(a)}{\leq} 2\tilde{d}_{t-\tau}\|\tilde{\theta}_t - \tilde{\theta}_{t-\tau}\| \\
&\leq 2(\tilde{d}_t + \|\tilde{\theta}_t - \tilde{\theta}_{t-\tau}\|)\|\tilde{\theta}_t - \tilde{\theta}_{t-\tau}\| \\
&\overset{(b)}{\leq} 2\left(\sqrt{\alpha\tau}\tilde{d}_t + \frac{\|\tilde{\theta}_t - \tilde{\theta}_{t-\tau}\|}{\sqrt{\alpha\tau}}\right)^2 \\
&\leq 4\left(\alpha\tau\tilde{d}_t^2 + \frac{\|\tilde{\theta}_t - \tilde{\theta}_{t-\tau}\|^2}{\alpha\tau}\right),
\end{aligned}
\tag{115}
$$

where for (a), we used the fact that $z_t(\theta)$ is 2-Lipschitz, and for (b), we used the fact that $\alpha\tau \leq 1$. Taking expectations on each side of the above inequality and invoking Lemma 23, we obtain

$$
\mathbb{E}[(**)] \leq 4\alpha\tau\mathbb{E}\left[\tilde{d}_t^2\right] + 4\alpha\tau\max_{t-\tau\leq\ell\leq t}G_\ell.
\tag{116}
$$

As before, the exact same bound as in the above display applies to $\mathbb{E}\left[(***)\right]$. We now turn to the main step in this proof.

$$
\begin{aligned}
\mathbb{E}\left[(*)\right] &= \mathbb{E}\left[\langle \tilde{\theta}_{t-\tau} - \theta^*, z_t(\tilde{\theta}_{t-\tau}) - \bar{g}(\tilde{\theta}_{t-\tau})\rangle\right] \\
&= \mathbb{E}\left[\langle \tilde{\theta}_{t-\tau} - \theta^*, \frac{1}{M}\sum_{i=1}^{M}\left(g_{i,t}(\tilde{\theta}_{t-\tau}) - \bar{g}(\tilde{\theta}_{t-\tau})\right)\rangle\right] \\
&= \mathbb{E}\left[\mathbb{E}\left[\langle \tilde{\theta}_{t-\tau} - \theta^*, \frac{1}{M}\sum_{i=1}^{M}\left(g_{i,t}(\tilde{\theta}_{t-\tau}) - \bar{g}(\tilde{\theta}_{t-\tau})\right)\rangle | \tilde{\theta}_{t-\tau}, \{X_{j,t-\tau}\}_{j\in[M]}\right]\right] \\
&= \mathbb{E}\left[\langle \tilde{\theta}_{t-\tau} - \theta^*, \frac{1}{M}\sum_{i=1}^{M}\left(\mathbb{E}\left[g_{i,t}(\tilde{\theta}_{t-\tau})|\tilde{\theta}_{t-\tau}, \{X_{j,t-\tau}\}_{j\in[M]}\right] - \bar{g}(\tilde{\theta}_{t-\tau})\right)\rangle\right] \\
&\overset{(a)}{=} \mathbb{E}\left[\langle \tilde{\theta}_{t-\tau} - \theta^*, \frac{1}{M}\sum_{i=1}^{M}\left(\mathbb{E}\left[g_{i,t}(\tilde{\theta}_{t-\tau})|\tilde{\theta}_{t-\tau}, X_{i,t-\tau}\right] - \bar{g}(\tilde{\theta}_{t-\tau})\right)\rangle\right] \\
&\overset{(b)}{\leq} \mathbb{E}\left[\tilde{d}_{t-\tau}\frac{1}{M}\sum_{i=1}^{M}\left\|\mathbb{E}\left[g_{i,t}(\tilde{\theta}_{t-\tau})|\tilde{\theta}_{t-\tau}, X_{i,t-\tau}\right] - \bar{g}(\tilde{\theta}_{t-\tau})\right\|\right] \\
&\overset{(c)}{\leq} \alpha^2 \mathbb{E}\left[\tilde{d}_{t-\tau}\left(\|\tilde{\theta}_{t-\tau}\| + 1\right)\right] \\
&\overset{(d)}{\leq} 3\alpha^2 \mathbb{E}\left[\tilde{d}_t^2\right] + 3\alpha^2 \mathbb{E}\left[\|\tilde{\theta}_t - \tilde{\theta}_{t-\tau}\|^2\right] + 12\alpha^2\sigma^2 \\
&\overset{(e)}{\leq} 3\alpha^2 \mathbb{E}\left[\tilde{d}_t^2\right] + 3\alpha^4\tau^2 \max_{t-\tau\leq\ell\leq t-1} G_\ell + 12\alpha^2\sigma^2 \\
&\overset{(f)}{\leq} \alpha\tau \mathbb{E}\left[\tilde{d}_t^2\right] + \alpha\tau \max_{t-\tau\leq\ell\leq t} G_\ell + 12\alpha^2\sigma^2.
\end{aligned}
\tag{117}
$$

In the above steps, (a) follows from the fact that the Markov data tuples are independent across agents; (b) follows from the Cauchy-Schwarz and the triangle inequality; (c) is a consequence of the mixing property in Definition 1; for (d), we used steps similar to those for arriving at equation 88; for (e), we used the bound on the drift from Lemma 23; and finally for (f), we simplified terms using $\alpha\tau \leq 1/3$ and $\tau \geq 1$. Combining the above bounds, we obtain

$$
\mathbb{E}\left[A_2\right] \leq 9\alpha\tau \mathbb{E}\left[\tilde{d}_t^2\right] + 9\alpha\tau \max_{t-\tau\leq\ell\leq t} G_\ell + 12\alpha^2\sigma^2.
$$

The above display, in tandem with equation 114, leads to the claim of the lemma. □

We are now ready to prove Theorem 3.

*Proof.* (**Proof of Theorem 3**) As in the proof of Theorem 1, our first step is to establish a recursion for the Lyapunov function $\Xi_t$ in equation 98. To that end, appealing to Lemmas 22 and 24, using the definition of $G_\ell$, and some algebra leads to the following bound for all $t \geq 2\tau$ :

$$
\begin{aligned}
\mathbb{E}\left[\tilde{d}_{t+1}^2\right] \leq &\left(1 - \alpha\omega(1-\gamma) + 42\alpha^2\tau\right)\mathbb{E}\left[\tilde{d}_t^2\right] + \frac{5\alpha^3}{\omega(1-\gamma)}E_{t-1} \\
&+ 352\alpha^2\tau \max_{t-\tau\leq\ell\leq t}\mathbb{E}\left[\tilde{d}_\ell^2\right] + 352\alpha^4\tau \max_{t-\tau\leq\ell\leq t}E_{\ell-1} + 24\alpha^2\left(\tau\left(\frac{32}{M} + 8\alpha^4\right) + \alpha\right)\sigma^2.
\end{aligned}
\tag{118}
$$

Now similar to the proof of Lemma 17, we have the following for each $i \in [M]$:

$$
\begin{aligned}
\mathbb{E}\left[\|e_{i,t}\|^2\right] &\leq \left(1 - \frac{1}{2\delta}\right)\mathbb{E}\left[\|e_{i,t-1}\|^2\right] + 2\delta\mathbb{E}\left[\|g_{i,t}(\theta_t)\|^2\right] \\
&\leq \left(1 - \frac{1}{2\delta}\right)\mathbb{E}\left[\|e_{i,t-1}\|^2\right] + 16\delta\mathbb{E}\left[\|\theta_t - \tilde{\theta}_t\|^2\right] + 4\delta\mathbb{E}\left[\|g_{i,t}(\tilde{\theta}_t)\|^2\right] \\
&\leq \left(1 - \frac{1}{2\delta}\right)\mathbb{E}\left[\|e_{i,t-1}\|^2\right] + 16\alpha^2\delta E_{t-1} + 64\delta\mathbb{E}\left[\tilde{d}_t^2\right] + 96\delta\sigma^2.
\end{aligned}
\tag{119}
$$

Averaging the above bound across all agents then yields:

$$
E_t \leq \left(1 - \frac{1}{2\delta} + 16\alpha^2\delta\right)E_{t-1} + 64\delta\mathbb{E}\left[\tilde{d}_t^2\right] + 96\delta\sigma^2.
\tag{120}
$$

Combining the above display with equation 118, and using the definition of $\Xi_t$, we obtain $\forall t \geq 2\tau$ :

$$
\begin{aligned}
\Xi_{t+1} \leq &\left(1 - \alpha\omega(1-\gamma) + 42\alpha^2\tau + 64C\delta\alpha^3\right)\mathbb{E}\left[\tilde{d}_t^2\right] + C\alpha^3\left(1 - \frac{1}{2\delta} + 16\alpha^2\delta + \frac{5}{C\omega(1-\gamma)}\right)E_{t-1} \\
&+ 352\alpha^2\tau \max_{t-\tau\leq\ell\leq t}\mathbb{E}\left[\tilde{d}_\ell^2\right] + 352\alpha^4\tau \max_{t-\tau\leq\ell\leq t}E_{\ell-1} + R \\
\leq &\left(1 - \alpha\omega(1-\gamma) + 42\alpha^2\tau + 64C\delta\alpha^3\right)\mathbb{E}\left[\tilde{d}_t^2\right] + C\alpha^3\left(1 - \frac{1}{2\delta} + 16\alpha^2\delta + \frac{5}{C\omega(1-\gamma)}\right)E_{t-1} \\
&+ 352\alpha^2\tau \max_{t-\tau\leq\ell\leq t}\Xi_\ell + \frac{352\alpha\tau}{C}\max_{t-\tau\leq\ell\leq t}C\alpha^3 E_{\ell-1} + R \\
\leq &\underbrace{\left(1 - \alpha\omega(1-\gamma) + 42\alpha^2\tau + 64C\delta\alpha^3\right)}_{B_1}\mathbb{E}\left[\tilde{d}_t^2\right] + C\alpha^3\underbrace{\left(1 - \frac{1}{2\delta} + 16\alpha^2\delta + \frac{5}{C\omega(1-\gamma)}\right)}_{B_2}E_{t-1} \\
&+ 352\alpha\tau\left(\alpha + \frac{1}{C}\right)\max_{t-\tau\leq\ell\leq t}\Xi_\ell + R,
\end{aligned}
\tag{121}
$$

where

$$
R = 24\alpha^2\left(\tau\left(\frac{32}{M} + 8\alpha^4\right) + \alpha\right)\sigma^2 + 96C\alpha^3\delta\sigma^2.
$$

Our goal is to now carefully pick $\alpha$ and $C$ so as to ensure that $\max\{B_1, B_2\} < 1$. Let us start with $B_2$. Suppose

$$
\alpha < \frac{1}{12\delta} \quad \text{and} \quad C = \frac{2816\max\{\delta, \tau\}}{\omega(1-\gamma)}.
\tag{122}
$$

It is easy to then verify that

$$
B_2 < 1 - \frac{1}{4\delta}.
$$

As for $B_1$, we note that if

$$
\alpha \leq \frac{\omega(1-\gamma)}{850\max\{\delta, \tau\}},
\tag{123}
$$

then

$$B_1 \leq 1 - \frac{\alpha\omega(1-\gamma)}{2}.$$

Also, under the requirement on the step-size $\alpha$ in equation 123, it is easy to check that

$$1 - \frac{1}{4\delta} < 1 - \frac{\alpha\omega(1-\gamma)}{2}.$$

In light of the above discussion, we conclude that $\forall t \geq 2\tau$:

$$\boxed{\Xi_{t+1} \leq \left(1 - \frac{\alpha\omega(1-\gamma)}{2}\right)\Xi_t + 352\alpha\tau\left(\alpha + \frac{1}{C}\right)\max_{t-\tau \leq \ell \leq t}\Xi_\ell + R.} \tag{124}$$

We are almost in a position to invoke Lemma 20. All that remains to be verified is whether

$$\underbrace{1 - \frac{\alpha\omega(1-\gamma)}{2}}_{p} + \underbrace{352\alpha^2\tau + 352\frac{\alpha\tau}{C}}_{q} < 1.$$

With the choice of $C$ in equation 122, if we set

$$\alpha \leq \frac{\omega(1-\gamma)}{2816\tau},$$

then one can verify that

$$p + q < 1 - \frac{\alpha\omega(1-\gamma)}{4}.$$

Combining all our prior requirements on $\alpha$, we conclude that if

$$\boxed{\alpha \leq \frac{\omega(1-\gamma)}{2816\max\{\delta, \tau\}}}, \tag{125}$$

then the following holds for all $T \geq 2\tau$:

$$\Xi_T \leq C_1\left(1 - \frac{\alpha\omega(1-\gamma)}{8\tau}\right)^{T-2\tau} + \frac{4R}{\alpha\omega(1-\gamma)}, \tag{126}$$

where $C_1 = \max_{0 \leq k \leq 2\tau}\Xi_k$. Simple calculations reveal that

$$\frac{4R}{\alpha\omega(1-\gamma)} = O\left(\frac{\alpha\tau}{\omega(1-\gamma)}\right)\frac{\sigma^2}{M} + O\left(\frac{\alpha^2\max\{\delta, \tau\}\delta}{\omega^2(1-\gamma)^2}\right)\sigma^2.$$

Using the above bound, and noting that $\mathbb{E}\left[\tilde{d}_t^2\right] \leq \Xi_T$, we have that $\forall T \geq 2\tau$:

$$\mathbb{E}\left[\tilde{d}_t^2\right] \leq C_1\left(1 - \frac{\alpha\omega(1-\gamma)}{8\tau}\right)^{T-2\tau} + O\left(\frac{\alpha\tau}{\omega(1-\gamma)}\right)\frac{\sigma^2}{M} + O\left(\frac{\alpha^2\max\{\delta, \tau\}\delta}{\omega^2(1-\gamma)^2}\right)\sigma^2. \tag{127}$$

The fact that $C_1 = O(d_0^2 + \sigma^2)$ follows from straightforward algebra similar to that in the proof of Theorem 1. This completes the proof. $\qquad\square$

# F   Analysis of Multi-Agent `EF-TD` under an I.I.D. Sampling Assumption

In this section, we provide a simpler (relative to that in Appendix E) analysis of the MARL setting under a common i.i.d. sampling assumption. Essentially, we consider a setting where for each agent $i$, at each time-step $t$, $s_{i,t}$ is sampled independently (from the past and across agents) from the stationary distribution $\pi$, and then $s_{i,t+1}$ is sampled from $P_\mu(\cdot|s_{i,t})$. As it turns out, this particular "i.i.d. model" has been widely studied in the RL literature (Lakshminarayanan & Szepesvári, 2017; Dalal et al., 2018; Bhandari et al., 2018; Doan et al., 2019; Liu & Olshevsky, 2021a); thus, we believe that providing an analysis for this setting would be useful to the reader. Our main result for this setting is as follows.

**Theorem 6.** *There exist universal constants $c, C \geq 1$, a step-size $\alpha \leq (1-\gamma)/(c\delta)$, and a set of convex weights $\{\bar{w}_t\}$, such that the iterates generated by Algorithm 2 satisfy the following after $T$ iterations:*

$$\mathbb{E}\left[\|\hat{V}_{\bar{\theta}_T} - \hat{V}_{\theta^*}\|_D^2\right] = O\left(\exp\left(\frac{-\omega(1-\gamma)^2 T}{C\delta}\right)\right) + \tilde{O}\left(\frac{\sigma^2}{\omega(1-\gamma)^2 MT}\right) + T_3, \tag{128}$$

*where $T_3 = \tilde{O}\left(\delta^2\sigma^2/(\omega^2(1-\gamma)^4 T^2)\right)$, $\bar{\theta}_T = \sum_{t=0}^T \bar{w}_t \theta_t$, and $\sigma^2 \triangleq \mathbb{E}\left[\|g_t(\theta^*)\|_2^2\right]$.[4]*

As in Theorem 3 where we analyzed the multi-agent setting under Markovian sampling, the above result also establishes a linear speedup in sample-complexity w.r.t. the number of agents $M$. The above result is somewhat cleaner in the sense that it provides a bound on the performance of the true iterate sequence $\{\theta_t\}$, as opposed to the perturbed iterate sequence $\{\tilde{\theta}_t\}$. We believe that it should be possible to provide a bound on the true iterates of the form in Theorem 6 under Markovian sampling as well; we leave this as future work.

To proceed with the analysis, we will require two auxiliary results. The first is taken from Bhandari et al. (2018).

**Lemma 25.** *Fix any $\theta \in \mathbb{R}^K$. The following holds under the i.i.d. sampling model:*

$$\mathbb{E}\left[\|g_t(\theta)\|^2\right] \leq 2\sigma^2 + 8\|\hat{V}_\theta - \hat{V}_{\theta^*}\|_D^2.$$

The next result is an "averaging lemma" that has been adapted from Koloskova et al. (2020) and Stich (2020).

**Lemma 26.** *Let $\{p_t\}_{t\geq 0}$ and $\{s_t\}_{t\geq 0}$ be sequences of positive numbers satisfying*

$$p_{t+1} \leq (1 - \alpha A)p_t - B\alpha s_t + \bar{C}\alpha^2 + D\alpha^3,$$

*for some positive constants $A, B \geq 0$, $C, D \geq 0$, and for constant step-sizes $0 < \alpha \leq \frac{1}{E}$, where $E > 0$. Then, there exists a constant step-size $\alpha \leq \frac{1}{E}$ such that*

$$\frac{B}{W_T}\sum_{t=0}^T w_t s_t \leq p_0(E + A)\exp\left(-\frac{A}{E}(T+1)\right) + \frac{2C \ln \bar{\tau}}{A(T+1)} + \frac{D \ln^2 \bar{\tau}}{A^2(T+1)^2}, \tag{129}$$

*for $w_t \triangleq (1 - \alpha A)^{-(t+1)}$, $W_T \triangleq \sum_{t=0}^T w_t$, and*

$$\bar{\tau} = \max\{\exp(1), \min\{A^2 p_0(T+1)^2/C, A^3 p_0(T+1)^3/D\}\}.$$

We start with the following result.

**Lemma 27.** *Suppose the step-size $\alpha$ is chosen such that $\alpha \leq (1-\gamma)/112$. Then, the iterates generated by Algorithm 2 satisfy the following under the i.i.d. observation model:*

$$\mathbb{E}\left[\|\tilde{\theta}_{t+1} - \theta^*\|^2\right] \leq \left(1 - \frac{\alpha\omega(1-\gamma)}{8}\right)\mathbb{E}\left[\|\tilde{\theta}_t - \theta^*\|^2\right] - \frac{\alpha(1-\gamma)}{4}\mathbb{E}\left[\|\hat{V}_{\theta_t} - \hat{V}_{\theta^*}\|_D^2\right] + \frac{5\alpha^3}{(1-\gamma)}E_{t-1} + \frac{8\alpha^2\sigma^2}{M}. \tag{130}$$

---

[4]We remind the reader here that $\omega$ is the smallest eigenvalue of the matrix $\Sigma = \Phi^\top D \Phi$.

*Proof.* Starting from equation 106, we have:

$$\mathbb{E}\left[\|\tilde{\theta}_{t+1} - \theta^*\|^2\right] = \mathbb{E}\left[\|\tilde{\theta}_t - \theta^*\|^2\right] + \underbrace{\frac{2\alpha}{M}\mathbb{E}\left[\langle\tilde{\theta}_t - \theta^*, \sum_{i\in[M]} g_{i,t}(\theta_t)\rangle\right]}_{T_1} + \underbrace{\alpha^2\mathbb{E}\left[\left\|\frac{1}{M}\sum_{i\in[M]} g_{i,t}(\theta_t)\right\|^2\right]}_{T_2}. \quad (131)$$

To bound $T_1$ and $T_2$, let us first define by $\mathcal{F}_{t-1}$ the sigma-algebra generated by all the agents' observations up to time-step $t-1$, i.e., the sigma-algebra generated by $\{X_{i,k}\}_{i\in[M],k=0,1,\ldots,t-1}$. From the dynamics of Algorithm 2, observe that $\theta_t, \tilde{\theta}_t$, and $\{e_{i,t}\}_{i\in[M]}$ are all $\mathcal{F}_{t-1}$-measurable. Using this fact, we bound $T_1$ as follows:

$$\begin{aligned}
T_1 &= \frac{2\alpha}{M}\mathbb{E}\left[\mathbb{E}\left[\langle\tilde{\theta}_t - \theta^*, \sum_{i\in[M]} g_{i,t}(\theta_t)\rangle|\mathcal{F}_{t-1}\right]\right] \\
&\overset{(a)}{=} 2\alpha\mathbb{E}\left[\langle\tilde{\theta}_t - \theta^*, \bar{g}(\theta_t)\rangle\right] \\
&= 2\alpha\mathbb{E}\left[\langle\theta_t - \theta^*, \bar{g}(\theta_t)\rangle\right] + 2\alpha\mathbb{E}\left[\langle\tilde{\theta}_t - \theta_t, \bar{g}(\theta_t)\rangle\right] \\
&\leq 2\alpha\mathbb{E}\left[\langle\theta_t - \theta^*, \bar{g}(\theta_t)\rangle\right] + \frac{\alpha(1-\gamma)}{4}\mathbb{E}\left[\|\bar{g}(\theta_t)\|^2\right] + \frac{4\alpha}{(1-\gamma)}\mathbb{E}\left[\|\theta_t - \tilde{\theta}_t\|^2\right] \\
&\leq 2\alpha\mathbb{E}\left[\langle\theta_t - \theta^*, \bar{g}(\theta_t)\rangle\right] + \frac{\alpha(1-\gamma)}{4}\mathbb{E}\left[\|\bar{g}(\theta_t)\|^2\right] + \frac{4\alpha^3}{(1-\gamma)}\mathbb{E}\left[\|\bar{e}_{t-1}\|^2\right] \\
&\overset{(b)}{\leq} 2\alpha\mathbb{E}\left[\langle\theta_t - \theta^*, \bar{g}(\theta_t)\rangle\right] + \frac{\alpha(1-\gamma)}{4}\mathbb{E}\left[\|\bar{g}(\theta_t)\|^2\right] + \frac{4\alpha^3}{(1-\gamma)}E_{t-1} \\
&\overset{(c)}{\leq} -\alpha(1-\gamma)\mathbb{E}\left[\|\hat{V}_{\theta_t} - \hat{V}_{\theta^*}\|_D^2\right] + \frac{4\alpha^3}{(1-\gamma)}E_{t-1}.
\end{aligned} \quad (132)$$

In the above steps, (a) follows from the fact that the agents' observations are assumed to have been drawn i.i.d. (over time and across agents) from the stationary distribution $\pi$; for (b), we applied equation 19; and (c) follows from Lemmas 4 and 5. To bound $T_2$, we first split it into two parts as follows:

$$\begin{aligned}
T_2 &= \frac{\alpha^2}{M^2}\mathbb{E}\left[\left\|\sum_{i\in[M]} (g_{i,t}(\theta_t) - \bar{g}(\theta_t)) + M\bar{g}(\theta_t)\right\|^2\right] \\
&\leq \frac{2\alpha^2}{M^2}\mathbb{E}\left[\left\|\sum_{i\in[M]} (g_{i,t}(\theta_t) - \bar{g}(\theta_t))\right\|^2\right] + 2\alpha^2\mathbb{E}\left[\|\bar{g}(\theta_t)\|^2\right].
\end{aligned} \quad (133)$$

To simplify the first term in the above inequality further, let us define $Y_{i,t} \triangleq g_{i,t}(\theta_t) - \bar{g}(\theta_t), \forall i \in [M]$. Conditioned on $\mathcal{F}_{t-1}$, observe that (i) $Y_{i,t}$ has zero mean for all $i \in [M]$; and (ii) $Y_{i,t}$ and $Y_{j,t}$ are independent for $i \neq j$ (this follows from the fact that $X_{i,t}$ and $X_{j,t}$ are independent by assumption). As a consequence of the two facts above, we immediately have

$$\mathbb{E}\left[\langle Y_{i,t}, Y_{j,t}\rangle|\mathcal{F}_{t-1}\right] = 0, \forall i, j \in [M] \, s.t. \, i \neq j.$$

We thus conclude that

$$
\mathbb{E}\left[\left\|\sum_{i\in[M]}(g_{i,t}(\theta_t)-\bar{g}(\theta_t))\right\|^2\right] = \mathbb{E}\left[\mathbb{E}\left[\|\sum_{i\in[M]}Y_{i,t}\|^2|\mathcal{F}_{t-1}\right]\right]
$$

$$
= \mathbb{E}\left[\sum_{i\in[M]}\mathbb{E}\left[\|Y_{i,t}\|^2|\mathcal{F}_{t-1}\right]\right] \tag{134}
$$

$$
\overset{(a)}{=} M\left(\mathbb{E}\left[\mathbb{E}\left[\|Y_{i,t}\|^2|\mathcal{F}_{t-1}\right]\right]\right)
$$

$$
= M\left(\mathbb{E}\left[\|Y_{i,t}\|^2\right]\right),
$$

where (a) follows from the fact that conditioned on $\mathcal{F}_{t-1}$, $Y_{i,t}$ and $Y_{j,t}$ are identically distributed for all $i,j\in[M]$ with $i\neq j$. Plugging the result in equation 134 back in equation 133, we obtain

$$
\begin{aligned}
T_2 &\leq \frac{2\alpha^2}{M}\mathbb{E}\left[\|(g_{i,t}(\theta_t)-\bar{g}(\theta_t))\|^2\right] + 2\alpha^2\mathbb{E}\left[\|\bar{g}(\theta_t)\|^2\right] \\
&\leq \frac{4\alpha^2}{M}\mathbb{E}\left[\|g_{i,t}(\theta_t)\|^2\right] + 2\alpha^2\left(1+\frac{2}{M}\right)\mathbb{E}\left[\|\bar{g}(\theta_t)\|^2\right] \\
&\leq 56\alpha^2\mathbb{E}\left[\|\hat{V}_{\theta_t}-\hat{V}_{\theta^*}\|_D^2\right] + \frac{8\alpha^2\sigma^2}{M},
\end{aligned} \tag{135}
$$

where in the last step, we used Lemmas 5 and 25. Now that we have bounds on each of the terms $T_1$ and $T_2$, we plug them back in equation 131 to obtain

$$
\begin{aligned}
\mathbb{E}\left[\|\tilde{\theta}_{t+1}-\theta^*\|^2\right] &\leq \mathbb{E}\left[\|\tilde{\theta}_t-\theta^*\|^2\right] - \alpha(1-\gamma)\left(1-\frac{56\alpha}{(1-\gamma)}\right)\mathbb{E}\left[\|\hat{V}_{\theta_t}-\hat{V}_{\theta^*}\|_D^2\right] + \frac{4\alpha^3}{(1-\gamma)}E_{t-1} \\
&\quad + \frac{8\alpha^2\sigma^2}{M} \\
&\leq \mathbb{E}\left[\|\tilde{\theta}_t-\theta^*\|^2\right] - \frac{\alpha(1-\gamma)}{2}\mathbb{E}\left[\|\hat{V}_{\theta_t}-\hat{V}_{\theta^*}\|_D^2\right] + \frac{4\alpha^3}{(1-\gamma)}E_{t-1} + \frac{8\alpha^2\sigma^2}{M},
\end{aligned} \tag{136}
$$

where in the last step, we used the fact that $\alpha\leq(1-\gamma)/112$. By splitting the second term in the above inequality into two equal parts, using Lemma 3, and the fact that

$$
-\mathbb{E}\left[\|\hat{V}_{\theta_t}-\hat{V}_{\theta^*}\|_D^2\right] \leq -\frac{1}{2}\mathbb{E}\left[\|\hat{V}_{\tilde{\theta}_t}-\hat{V}_{\theta^*}\|_D^2\right] + \alpha^2 E_{t-1},
$$

we further obtain that

$$
\mathbb{E}\left[\|\tilde{\theta}_{t+1}-\theta^*\|^2\right] \leq \left(1-\frac{\alpha\omega(1-\gamma)}{8}\right)\mathbb{E}\left[\|\tilde{\theta}_t-\theta^*\|^2\right] - \frac{\alpha(1-\gamma)}{4}\mathbb{E}\left[\|\hat{V}_{\theta_t}-\hat{V}_{\theta^*}\|_D^2\right] + \frac{5\alpha^3}{(1-\gamma)}E_{t-1} + \frac{8\alpha^2\sigma^2}{M},
$$

which is the desired conclusion. $\qquad\square$

We now complete the proof of Theorem 6 as follows.

*Proof.* (**Proof of Theorem 3**) Our goal is to establish a recursion of the form in Lemma 26. To that end, we need to first control the aggregate effect of the memory variables of all agents, as captured by the term $E_t$. This is easily done by first using the same analysis as in Lemma 17, and then appealing to Lemma 25, to conclude

$$
\mathbb{E}\left[\|e_{i,t}\|^2\right] \leq \left(1-\frac{1}{2\delta}\right)\mathbb{E}\left[\|e_{i,t-1}\|^2\right] + 16\delta\mathbb{E}\left[\|\hat{V}_{\theta_t}-\hat{V}_{\theta^*}\|_D^2\right] + 4\delta\sigma^2, \forall i\in[M].
$$

Averaging the above inequality over all agents, and using the definition of $E_t$ yields:

$$
E_t \leq \left(1-\frac{1}{2\delta}\right)E_{t-1} + 16\delta\mathbb{E}\left[\|\hat{V}_{\theta_t}-\hat{V}_{\theta^*}\|_D^2\right] + 4\delta\sigma^2.
$$

Using the above bound along with Lemma 27, we obtain:

$$
\begin{aligned}
\Xi_{t+1} &\leq \left(1 - \frac{\alpha\omega(1-\gamma)}{8}\right)\mathbb{E}\left[\|\tilde{\theta}_t - \theta^*\|^2\right] - \frac{\alpha(1-\gamma)}{4}\mathbb{E}\left[\|\hat{V}_{\theta_t} - \hat{V}_{\theta^*}\|_D^2\right] + \frac{5\alpha^3}{(1-\gamma)}E_{t-1} + \frac{8\alpha^2\sigma^2}{M} + C\alpha^3 E_t \\
&\leq \left(1 - \frac{\alpha\omega(1-\gamma)}{8}\right)\mathbb{E}\left[\|\tilde{\theta}_t - \theta^*\|^2\right] - \left(\frac{\alpha(1-\gamma)}{4} - 16C\alpha^3\delta\right)\mathbb{E}\left[\|\hat{V}_{\theta_t} - \hat{V}_{\theta^*}\|_D^2\right] \\
&\quad + C\alpha^3\left(1 - \frac{1}{2\delta} + \frac{5}{C(1-\gamma)}\right)E_{t-1} + \frac{8\alpha^2\sigma^2}{M} + 4C\alpha^3\delta\sigma^2.
\end{aligned}
$$
(137)

Based on the above inequality, our goal is to now choose $\alpha$ and $C$ in a way such that we can establish a contraction (up to higher order noise terms). Accordingly, let us pick these parameters $\alpha$ and $C$ as follows:

$$
C = \frac{20\delta}{(1-\gamma)}; \quad \alpha \leq \frac{(1-\gamma)}{55\delta}.
$$

With some simple algebra, it is then easy to verify that:

$$
\begin{aligned}
\Xi_{t+1} &\leq \left(1 - \frac{\alpha\omega(1-\gamma)}{8}\right)\mathbb{E}\left[\|\tilde{\theta}_t - \theta^*\|^2\right] - \frac{\alpha(1-\gamma)}{8}\mathbb{E}\left[\|\hat{V}_{\theta_t} - \hat{V}_{\theta^*}\|_D^2\right] + C\alpha^3\left(1 - \frac{1}{4\delta}\right)E_{t-1} \\
&\quad + \frac{8\alpha^2\sigma^2}{M} + \frac{80\alpha^3\delta^2\sigma^2}{(1-\gamma)} \\
&\leq \left(1 - \frac{\alpha\omega(1-\gamma)}{8}\right)\left(\mathbb{E}\left[\|\tilde{\theta}_t - \theta^*\|^2\right] + C\alpha^3 E_{t-1}\right) - \frac{\alpha(1-\gamma)}{8}\mathbb{E}\left[\|\hat{V}_{\theta_t} - \hat{V}_{\theta^*}\|_D^2\right] \\
&\quad + \frac{8\alpha^2\sigma^2}{M} + \frac{80\alpha^3\delta^2\sigma^2}{(1-\gamma)} \\
&= \left(1 - \frac{\alpha\omega(1-\gamma)}{8}\right)\Xi_t - \frac{\alpha(1-\gamma)}{8}\mathbb{E}\left[\|\hat{V}_{\theta_t} - \hat{V}_{\theta^*}\|_D^2\right] + \frac{8\alpha^2\sigma^2}{M} + \frac{80\alpha^3\delta^2\sigma^2}{(1-\gamma)}.
\end{aligned}
$$
(138)

We have thus succeeded in establishing a recursion of the form in Lemma 26. To spell things out explicitly in the language of Lemma 26, we have

$$
p_t = \Xi_t; s_t = \mathbb{E}\left[\|\hat{V}_{\theta_t} - \hat{V}_{\theta^*}\|_D^2\right]; A = \frac{\omega(1-\gamma)}{8}; B = \frac{(1-\gamma)}{8}; \bar{C} = \frac{8\sigma^2}{M}; D = \frac{80\delta^2\sigma^2}{(1-\gamma)},
$$

and $\alpha \leq (1-\gamma)/(112\delta)$ suffices for the recursion in equation 138 to hold. Thus, for us, $E = \frac{112\delta}{(1-\gamma)}$. Applying Lemma 26 along with some simplifications then yields:

$$
\begin{aligned}
\sum_{t=0}^{T}\bar{w}_t\mathbb{E}\left[\|\hat{V}_{\theta_t} - \hat{V}_{\theta^*}\|_D^2\right] &\leq O\left(\frac{\|\theta_0 - \theta^*\|^2\delta}{(1-\gamma)^2}\right)\exp\left(\frac{-\omega(1-\gamma)^2 T}{C'\delta}\right) + \tilde{O}\left(\frac{\sigma^2}{\omega(1-\gamma)^2 MT}\right) \\
&\quad + \tilde{O}\left(\frac{\sigma^2\delta^2}{\omega^2(1-\gamma)^4 T^2}\right),
\end{aligned}
$$
(139)

where $\bar{w}_t \triangleq w_t/W_T$, and $C'$ is a suitably large constant. The result follows by noting that

$$
\mathbb{E}\left[\|\hat{V}_{\bar{\theta}_T} - \hat{V}_{\theta^*}\|_D^2\right] \leq \sum_{t=0}^{T}\bar{w}_t\mathbb{E}\left[\|\hat{V}_{\theta_t} - \hat{V}_{\theta^*}\|_D^2\right],
$$

where $\bar{\theta}_T = \sum_{t=0}^{T}\bar{w}_t\theta_t$. $\qquad\square$

