# OpenReview forum: "Temporal Difference Learning with Compressed Updates: Error-Feedback meets Reinforcement Learning"
_TMLR — Accepted by TMLR_

### Review · Reviewer_fvTK · 2023-12-31

**Summary Of Contributions:**

This paper investigates the robustness of reinforcement learning (RL) algorithms to compressed updates. This attempts to extend to the RL setting the type of results that have been carried out for stochastic gradient descent. The authors first develop a general framework for compressed stochastic approximation, with a perturbed update direction. They also consider a collaborative multi-agent RL (MARL) setting and show that it is possible, even in the presence of compression, to have a scheme almost as efficient as in the non-compressed case.

**Audience:**

Yes

**Broader Impact Concerns:**

No specific concerns.

**Claims And Evidence:**

Yes

**Requested Changes:**

I do not have any specific changes to request.

**Strengths And Weaknesses:**

The main ideas are clearly presented, both for the results and the technical contributions.

---

> ### Author Response · Authors · 2024-02-07
> **Response to Reviewer fvTK**
>
> Dear Reviewer fvTK,
>
> Thank you for your review of our paper, and for your positive feedback.

---

### Review · Reviewer_GFLY · 2024-02-05

**Summary Of Contributions:**

The paper provides theoretical analysis of the impact of using compressed gradients on the convergence of policy evaluation.
Namely, the paper considers TD(0) algorithms for a quite general form of compression types (including using only the sign of the gradient, or the top-k dimensions) and analyses algorithms that use error feedback by locally accumulating the differences between actual and compressed update direction. In particular, the paper proves

* linear convergence for TD(0) with linear function approximation
* a similar theorem for the non-linear setting under the assumption that the update direction is Lipschitz continuous
* linear speedup with the number of agents in a multi-agent RL setting.

The theoretical results are validated on simple toy problems.

**Audience:**

Yes

**Broader Impact Concerns:**

I don't think that the paper requires a broader impact statement.

**Claims And Evidence:**

Yes

**Requested Changes:**

* Section 3 claims that the TD(0) update direction is not a stochastic gradient of any fixed objective. To me it is not immediately clear whether this is true or not, and hence, I would suggest to either provide evidence or soften the claim. (The claim is not critical to the contribution)

* The main theorems all state that there exist a $C \ge 1$ , but later uses $C_\tau$. Please make the notation consistent.

* In theorem, as far as I understand, should read "Suppose Assumption 2 holds **with L=2** [...]"

* I suggest to use logscale when plotting the MSE in Figure 2, to make it easier to spot differences (if any) between TD and EF-SignTD

**Strengths And Weaknesses:**

* The theoretical results seem to be novel and relevant. While similar analysis was performed for SGD (with similar guarantees), the paper seems to be the first to establish theoretical guarantees for MDPs. Furthermore, the paper makes the point, that this setting requires novel proof techniques.

* The presentation is excellent. The paper makes a very good job in motivating the problem, discussing the results and comparing to related work and pointing out the differences.

* I do not see any major weakness.

---

> ### Author Response · Authors · 2024-02-07
> **Response to Reviewer GFLY**
>
> Dear Reviewer GFLY,
>
> Thank you for your review of our paper and for your positive feedback. Below, we respond to each of the points you raised.
>
> - **Reviewer**: Section 3 claims that the TD(0) update direction is not a stochastic gradient of any fixed objective. To me it is not immediately clear whether this is true or not, and hence, I would suggest to either provide evidence or soften the claim.
>
> - **Reviewer**: We thank the Reviewer for bringing up this point. To see why the TD(0) update direction is not necessarily the gradient of any fixed objective function, let us consider the steady-state version of this direction $\bar{g}(\theta)$ as defined in Section 4 of our paper (a similar argument applies to the noisy direction). As discussed in the proof of Lemma 7 in the Appendix, the update direction $\bar{g}(\theta)$ takes the explicit affine form shown below:
> \begin{equation}
>     \bar{g}(\theta)= \bar{A}\theta-\bar{b}, \hspace{1mm} \textrm{where} \hspace{1mm} \bar{A} = \Phi^{\top} D \left(\gamma P_{\mu} - I \right) \Phi, \hspace{1mm} \textrm{and} \hspace{1mm} \bar{b} = - \Phi^{\top} D R_{\mu}.
>     \end{equation}
>
> Now if $\bar{g}(\theta)$ were indeed the gradient of some objective function, differentiating it (w.r.t. $\theta$) would lead to a *symmetric* Hessian matrix. However, taking the derivative of $\bar{g}(\theta)$ w.r.t. $\theta$  yields $\bar{A}$, which is not necessarily a symmetric matrix. The above explanation of why the TD(0) update direction is not the gradient of a fixed objective function has appeared in prior works, e.g., Bhandari et al., COLT 2018, and Liu and Olshevsky, ICML 2021.
>
> In our revised paper, we have now added footnote 2 on Page 7 to briefly explain the above point. We have also cited the above papers for further reference.
>
> - **Reviewer**: The main theorems all state that there exist a $C \geq 1$,  but later uses $C_\tau$. Please make the notation consistent.
>
> - **Response**: Perhaps there is some confusion here. There is no $C_\tau$ that shows up in any of our results. Instead, what does show up is $C \tau$, where $C \geq 1$ is some universal constant (independent of $\tau$), and $\tau$ is the mixing time. To be more explicit, our results feature a *product* between $C$ and $\tau$, as opposed to any new notation $C_{\tau}$. Please let us know if we understood your comment correctly and if the above response resolves it.
>
> - **Reviewer**: In theorem, as far as I understand, should read "Suppose Assumption 2 holds with L=2 [...]"
>
> - **Response**: Thank you for this comment. In the revised paper, we have now rephrased the statement of Theorem 2 according to your suggestion.
>
> - **Reviewer**: I suggest to use logscale when plotting the MSE in Figure 2, to make it easier to spot differences (if any) between TD and EF-SignTD.
>
> - **Response**: Thank you for this suggestion. We have now used a logscale plot for Figure 2.
>
> We hope that the above changes address all your concerns. We would be happy to answer any other questions you may have.
>
> Sincerely,
>
> Authors.

---

> > ### Comment · Reviewer_GFLY · 2024-02-08
> > **My comments have been addressed**
> >
> > Thank you for your reply. All my points have been adequately addressed and I think that the submission is in a very good shape.
> > I indeed misread $C \tau$ as $C_\tau$. I think the notation is sufficiently clear as is, but maybe changing it to $\tau C$ would not be a bad idea.

---

### Review · Reviewer_KbFY · 2024-02-08

**Summary Of Contributions:**

Summary of contribution This paper provides the first theoretical analysis for TD learning with compressed updates and error feedback, which achieves a similar convergence rate compared with the SGD counterpart without compression. This result reveals the robustness of RL algorithms to structured perturbations. The authors also conducted simulation to verify the theory.

**Audience:**

Yes

**Claims And Evidence:**

Yes

**Requested Changes:**

Summary of contribution This paper provides the first theoretical analysis for TD learning with compressed updates and error feedback, which achieves a similar convergence rate compared with the SGD counterpart without compression. This result reveals the robustness of RL algorithms to structured perturbations. The authors also conducted simulation to verify the theory.

Strengths and weaknesses Strengths

The paper makes solid technical contributions. The analysis is highly non-trivial considering the temporal correlations in Markovian sampling.
Going beyond the linear approximation case, the authors also extend their theory to cases of general nonlinear stochastic approximation and MARL, which greatly enriches and strengths the result. 3.The paper is well-written and easy to follow. Weaknesses & Questions 1.Question about the discussion on Thm. 1: As the authors stated, the effects of the mixing time  and the distortion measure  has an additive effect on inflating the variance. It seems that, to ensure the same level of
, one should use smaller  for larger . However, the paper states the opposite. Could the authors explain more on this? 2.Abuse of notation: The authors use  to denote both the reward and the convergence error (Thm. 1), which probably leads to confusion. 3.A M-fold speedup -> an M-fold speedup; the only other paper that does so is the very recent work (Khodadadian et al., 2022) -> the only other paper that does so is the very recent work Khodadadian et al., 2022 Requested changes Some clarifications and minor corrections requested in "Weaknesses & Questions"

**Strengths And Weaknesses:**

Strengths and weaknesses Strengths

The paper makes solid technical contributions. The analysis is highly non-trivial considering the temporal correlations in Markovian sampling.
Going beyond the linear approximation case, the authors also extend their theory to cases of general nonlinear stochastic approximation and MARL, which greatly enriches and strengths the result. 3.The paper is well-written and easy to follow. Weaknesses & Questions 1.Question about the discussion on Thm. 1: As the authors stated, the effects of the mixing time  and the distortion measure  has an additive effect on inflating the variance. It seems that, to ensure the same level of
, one should use smaller  for larger . However, the paper states the opposite. Could the authors explain more on this? 2.Abuse of notation: The authors use  to denote both the reward and the convergence error (Thm. 1), which probably leads to confusion. 3.A M-fold speedup -> an M-fold speedup; the only other paper that does so is the very recent work (Khodadadian et al., 2022) -> the only other paper that does so is the very recent work Khodadadian et al., 2022 Requested changes Some clarifications and minor corrections requested in "Weaknesses & Questions"

---

> ### Author Response · Authors · 2024-02-08
> **Response to Reviewer KbFY**
>
> Dear Reviewer KbFY,
>
> Thank you for your review of our paper and for your positive feedback. Below, we respond to each of the points you raised.
>
> - **Reviewer**: Question about the discussion on Thm. 1.
>
> - **Response**: Thank you for raising this interesting question. Our variance bound - as revealed by Theorem 1 - is $O(\alpha (\tau + \delta) \sigma^2)$, where $\alpha$ is the step-size, $\tau$ is the mixing-time of the underlying Markov chain, $\delta$ captures the effect of compression, and $\sigma^2$ is the variance of our noise model. Now the key thing to note is that even in the absence of compression,  the $O(\alpha \tau \sigma^2)$ term would inevitably persist - this linear dependence on the mixing time $\tau$ is known to be information-theoretically optimal (Nagaraj et al., 2020). We had mentioned this point in Section 6 of our paper.
>
> Thus, given that the effect of compression and the mixing-effect of the Markov chain show up *additively*, and the latter effect is *unavoidable*, as long as the compression factor $\delta$ is of the same order as the mixing time $\tau$, the resulting variance bound would be no worse than in the case without compression. Said differently, a Markov chain that mixes slowly, i.e., has a larger $\tau$, provides a greater leeway for aggressive compression. Finally, we note that once the policy to be evaluated is fixed, we have no control over the mixing-time $\tau$ of the Markov chain it induces. However, we do have control over the compression parameter $\delta$. Thus, it makes sense to tune $\delta$ to be of the same order as $\tau$, since this leads to no loss in performance relative to the uncompressed setting.
>
> We hope this clarifies your concern. We have briefly added parts of the above discussion to the revised version of our paper.
>
> -  **Reviewer**:  Abuse of notation: The authors use $r_t$ to denote both the reward and the convergence error (Thm. 1), which probably leads to confusion.
>
> - **Response**: Thank you for noticing this! We had missed this point in our original submission. Throughout the revised paper, we have now used $d_t$ to represent the MSE error at time-step $t$.
>
> In addition to the above changes, we have incorporated your pointers regarding grammatical issues in our revised paper.
>
> We hope that the above changes address all your concerns. We would be happy to answer any other questions you may have.
>
> Sincerely,
>
> Authors.

---

### Review · Reviewer_bgpG · 2024-02-14

**Summary Of Contributions:**

The paper introduces analysis for a variant of TD-learning where the updates are corrupted - the update vector is either computed to be a signed vector, or compressed to be just the top-k components. The paper proposes EF-TD, an error feedback mechanism to improve the convergence property of the algorithm. The paper analyzes such techniques, shows its convergence guarantee in both gradient descent and stochastic gradient descent case, and its extension to multi-agent case. The paper finishes with a simulation study on the practical behavior of such algorithms in tabular domains.

**Audience:**

Yes

**Broader Impact Concerns:**

No.

**Claims And Evidence:**

Yes

**Requested Changes:**

- What's the connection between EF-TD and momentum based methods for TD-learning? It feels that the error feedback mechanism is highly similar (at least in mathematical forms) of momentum based approaches for accelerating SGDs.

- Without the error feedback, is it possible to deliver convergence results to the corresponding TD algorithms?

- With the error correction feedback, I wonder if the update has been not as communication efficient as motivated. Concretely, each vector $h_t$ is sparse (for the top-k variant) and can be stored with fewer bytes (for the signed variant),  but the error correction term $e_t$ is not. In fact the complexity of the original update vector has been relayed to $e_t$ from $h_t$. So it feels like the motivation from efficient communication is not really addressed as expected.

- Fig 1 provides a geometric interpretation of $Q_\delta$, but I wonder whether it'd be more informative to provide geometric insights on how EF-TD works. Prior work (Bhandari et al 2018) provides a geometric insight into the convergence of TD-learning, that is the gradient moves in a direction that minimizes the distance between $\theta$ and $\theta^*$. If we do not carry out EF, does this become invalid? If we carry out EF, what kind of corrections do we gain in the geometric sense?

- I am not an expert in noisy robust SGD analysis, but I wonder how much novelty the proof technique is from a proof process that takes the analysis from a SGD literature and adapt to the TD case similar to the techniques used in Bhandari et al 2018.

- Simulation would be nice to see the scaling curve in terms of k for the top-k variant. How do the behavior changes as we increase k? We should expect to recover TD as k becomes large and worse performance as k becomes small.

- TD and EF-signed-TD appear to have exactly the same performance in Fig 2. This makes me wonder whether the EF process basically makes up for the difference between signed TD and TD, which defies the sparse or low-storage motivation for such updates in the first place.

**Strengths And Weaknesses:**

Overall the paper has made solid theoretical contributions to the field of TD-learning analysis. They consider a particular variant of the algorithm which is motivated by sparse or communication efficient update. The theoretical results seem novel, as they adopt new techniques for proving the convergence, and generalize prior results as special case. The theoretical results are also fairly interesting and showing the convergence of the EF-TD algorithm.

The weakness is mainly empirical: ideally the paper can provide more simulation results that showcase the practical efficiencies of the algorithm, and shed light on the theoretical insights. From the theoretical perspective, I would also hope that the paper provides more geometric insight of the proof technique and discuss how the EF-TD algorithm addresses the communication efficient motivation for the paper - in its current form, EF-TD does not seem to be communication efficient due to the error feedback vector that needs to be stored.

---

> ### Author Response · Authors · 2024-02-19
> **Response to Reviewer bgpG**
>
> Dear Reviewer bgpG,
>
> Thank you for your review of our paper, and for your valuable feedback. Below, we respond to each of your requested changes.
>
> - **Connection between EF-TD and momentum-based methods**: We thank the Reviewer for this interesting question. Indeed, in the context of optimization, the SignSGD method with momentum is known to exhibit convergence behavior very similar to adaptive optimization algorithms such as ADAM. In fact, this is one of the main reasons why SignSGD with momentum has been used for fast training of deep neural nets. Thus, it is natural to ask - as the Reviewer does - whether SignTD with momentum will also exhibit fast convergence? As it stands, a resolution of this question is well beyond the scope of this paper for the following reason: we are not aware of a finite-time analysis of any adaptive or momentum-based/accelerated TD method. Thus, unlike momentum-based SGD or ADAM, we don't have any "fast" benchmark to compare to when it comes to temporal difference learning. As such, we are unable to make any reasonable conjecture at this point. That said, we believe that this is indeed a very interesting question to pursue; we have now mentioned this in the Conclusion section of our revised paper.
>
> - **Without the error feedback, is it possible to deliver convergence results to the corresponding TD algorithms?**: It turns out that even without error-feedback, one can still provide convergence bounds for compressed TD algorithms. In the revised paper, we have added a new Appendix, namely Appendix B, to elaborate on this point. In Appendix B.1, we analyze a stead-state/mean-path version of compressed TD(0) *without error-feedback.* Our analysis reveals that one can guarantee linear convergence of this algorithm, provided the compression parameter $\delta$ satisfies the following criterion:
> $$ \delta < \frac{1}{1-\beta^2},$$
> where $\beta \in (0,1)$ is a parameter that depends on the underlying Markov chain induced by the policy to be evaluated, and also the feature vectors used for linear function approximation (details provided in Appendix B.1). Thus, without error-feedback, it appears that one cannot choose the compression parameter $\delta$ arbitrarily; rather, it is limited by the parameter $\beta$. This stands in sharp contrast to the setting with error-feedback, where no such restriction on $\delta$ is needed to guarantee convergence. The above discussion sheds further light on the benefit of error-feedback. We should note, however, that our analysis of compressed TD(0) without error-feedback only reveals a sufficient condition for convergence on the parameter $\delta$; whether such a condition is necessary remains open.
>
> - **With the error correction feedback, I wonder if the update has been not as communication efficient as motivated.** Thank you for raising this important point. In our description of EF-TD, perhaps we should have been clearer in terms of *what* exactly gets communicated. We emphasize here that our algorithm only involves transmission of the vector $h_t$ - the output of the compression operator $\mathcal{Q}_{\delta}(\cdot).$ Depending on the operator, $h_t$ can be encoded using just a few bits. As the Reviewer rightly observes, the error-vector $e_t$ will very likely be dense, but this dense vector *is never transmitted*. **As such, since our algorithms only involve transmitting the (sparse) vector $h_t$, they are indeed communication-efficient, as claimed**. We have made no claims about our algorithms being memory-efficient, so storing the potentially dense vector $e_t$ does not go against anything we have stated in this paper.
>
> In our revised paper, we have added a new Remark, namely Remark 1, to explain the above point. We also point the Reviewer to line 6 of the multi-agent version of EF-TD (Algorithm 2), where we clearly state that each agent only uploads its compressed TD direction to the server, not the error-feedback error vectors. These error vectors are just stored and updated locally. We hope this resolves your concern.

---

> > ### Author Response · Authors · 2024-02-19
> > **Response to Reviewer bgpG (Continued)**
> >
> > - **Comment on Geometric Insights**: This is again a great question. Indeed, Bhandari et al., provide insights regarding the behavior of EF-TD by establishing the following inequality
> > $$ \langle \theta - \theta^*, \bar{g}(\theta) \rangle \leq - \beta \Vert \theta - \theta^* \Vert^2, \forall \theta \in \mathbb{R}^K,$$
> > where $\bar{g}(\theta)$ is the mean-path/steady-state version of the TD update direction, and $\beta \in (0,1)$ is a parameter dictated by the underlying Markov chain and the feature vectors used for linear function approximation. It turns out that the convergence of TD learning hinges crucially on the above property, since it is precisely this property that leads to a contractive term. Inspired by the Reviewer's question, we have now added a new Appendix, namely Appendix B, where we analyze mean-path versions of compressed TD, with and without error-feedback. Let us start with the case without error-feedback. Here, we analyze the following algorithm:
> > $$ \theta_{t+1}  =  \theta_t + \alpha \mathcal{Q}(\bar{g}(\theta_t)) ,$$
> > where $\mathcal{Q}(\cdot)$ is the compression operator. Notice that no error-feedback mechanism is employed in the above scheme. In search of a geometric descent condition as in Bhandari et al., we manage to show the following:
> > $$ \langle \theta - \theta^*, \mathcal{Q}(\bar{g}(\theta)) \rangle \leq - \left(\beta - \sqrt{\left(1-\frac{1}{\delta}\right)} \right) \Vert \theta - \theta^* \Vert^2,$$
> > where $\delta$ is the compression factor. The above display tells us that the compressed TD direction $\mathcal{Q}_{\delta}(\bar{g}(\theta))$ can still enable us to make progress towards $\theta^*$, provided the compression parameter $\delta$ is not "too large", and satisfies:
> > $$ \delta < \frac{1}{(1-\beta^2)}.$$
> > One can interpret the above as a condition on $\delta$ for convergence of the iterates. This tells us that compressed TD without error-feedback can still guarantee convergence to $\theta^*$, provided the compression parameter $\delta$ does not exceed a limit. Whether the limit identified by our analysis is tight remains an open question.
> >
> > In sharp contrast, our analysis of mean-path TD with error-feedback (in Appendix B) reveals that one can guarantee convergence to $\theta^*$ without making any assumption at all on the compression parameter $\delta$. This highlights one of the key benefits of employing error-feedback. Since the steady-state/mean-path version of EF-TD is the simplest version of our setting we could think of (analogous to mean-path TD in Bhandari et al.), we hope this conveys sufficient insights into the functioning of our algorithm.
> >
> > - **On the Novelty in our Analysis**: In the original submitted version of our paper, we had dedicated a large part of Section 1.1. - where we explain our contributions - to highlighting the novel technical ingredients needed for proving each of our main results. In addition, we had devoted an entire section, namely Section 7, to further elaborating on the key technical challenges in our analysis, relative to the existing literature on both optimization and RL.
> >
> > We briefly touch upon the main points here, but would kindly urge the Reviewer to go over the above parts of our paper for more details; we have highlighted them for convenience.
> >
> > As we explain in detail in Section 1.1., the main technical challenge in analyzing EF-TD comes from the fact that there is an intricate relationship between the parameter sequence, the temporally correlated Markovian data samples, and the memory variable used for error-feedback. This leads to a stochastic dynamical system much more complex than the standard TD learning algorithm. We proceed to analyze this system by a constructing a novel Lyapunov function that captures the joint dynamics of the iterate-error and the error-feedback variable. However, the construction of this potential function is not enough. When we proceed to analyze the drift of this Lyapunov function - as is typically done in the analysis of stochastic approximation algorithms driven by Markovian noise - we run into non-standard delay terms due to the presence of the error-feedback variable. Such a difficulty never arises in the analysis of compressed optimization algorithms since one does not need to contend with Markovian data. Moreover, the delay terms we encounter above do not show up in the analysis of vanilla TD since there one does not use error-feedback variables from the past. As such, we need to develop novel proof techniques to overcome this hurdle. We do so by making an interesting connection to the analysis of the Incremental Aggregated Gradient (IAG) algorithm in optimization.

---

> > > ### Author Response · Authors · 2024-02-19
> > > **Response to Reviewer bgpG (Still Continued)**
> > >
> > > **(Novelty in Analysis)**: Below, we continue with the discussion of novelty in our technical analysis.
> > >
> > > It is important to emphasize that assuming a projection step - similar to what is done in Bhandari et al. - could have significantly simplified our analysis since we could have easily argued uniform boundedness of the iterates. However, we depart from Bhandari et al. by not assuming any such projection step; this further complicates the analysis.
> > >
> > > The above innovations are still not enough to prove the linear speedup result of Theorem 3 under Markovian sampling. This requires further work that we explain both in Section 7, and in Appendix E.
> > >
> > > In short, while our proofs do leverage existing ideas from the analyses of compressed optimization and TD learning, there are significant points of innovation and departure from each of these strands of literature.
> > >
> > > - **Additional simulations for top-k compression operator**: Thank you for your comment. In response, we have now added a new figure (Figure 3 in our revised paper) depicting the performance of EF-TD with a top-$k$ operator, where we vary $k$. The change in performance as we vary $k$ aligns both with intuition, and our theory.
> > >
> > > - **Reviewer bqpG**: TD and EF-signed-TD appear to have exactly the same performance in Fig 2. This makes me wonder whether the EF process basically makes up for the difference between signed TD and TD, which defies the sparse or low-storage motivation for such updates in the first place.
> > >
> > > **Response**: Perhaps there is some confusion. The purpose of EF-TD is **not to facilitate low-storage, but rather to facilitate low communication**. As we have explained in response to one of your earlier comments, our proposed algorithms are indeed communication-efficient as claimed, since they only require transmitting the compressed TD update directions which can be encoded using just a few bits. Notably, the potentially dense error vectors are stored and updated locally, but *do not need to be communicated.* So while error-feedback does indeed make up for the information that is lost due to compression (by retaining memory), this does not come at the expense of any additional communication. We also emphasize that the error-feedback mechanisms used for optimization operate in exactly the same way as in our paper: communicate sparse vectors, and locally store and update potentially dense error vectors.
> > >
> > > We hope that the above changes address all your concerns. We would be happy to answer any other questions you may have.
> > >
> > > Sincerely,
> > >
> > > Authors.

---

### Comment · Reviewer_KbFY · 2024-01-07
**Paper review**

Summary of contribution
This paper provides the first theoretical analysis for TD learning with compressed updates and error feedback, which achieves a similar convergence rate compared with the SGD counterpart without compression. This result reveals the robustness of RL algorithms to structured perturbations. The authors also conducted simulation to verify the theory.

Strengths and weaknesses
Strengths
1. The paper makes solid technical contributions. The analysis is highly non-trivial considering the temporal correlations in Markovian sampling.
2. Going beyond the linear approximation case, the authors also extend their theory to cases of general nonlinear stochastic approximation and MARL, which greatly enriches and strengths the result.
3.The paper is well-written and easy to follow.
Weaknesses & Questions
1.Question about the discussion on Thm. 1: As the authors stated,  the effects of the mixing time $\tau$ and the distortion measure $\delta$ has an additive effect on inflating the variance. It seems that, to ensure the same level of $E(r_T^2)$, one should use smaller $\delta$ for larger $\tau$. However, the paper states the opposite. Could the authors explain more on this?
2.Abuse of notation: The authors use $r_t$ to denote both the reward and the convergence error (Thm. 1), which probably leads to confusion.
3.A M-fold speedup -> an M-fold speedup; the only other paper that does so is the very recent work (Khodadadian et al., 2022) -> the only other paper that does so is the very recent work Khodadadian et al., 2022
Requested changes
Some clarifications and minor corrections requested in  "Weaknesses & Questions"

---

> ### Author Response · Authors · 2024-02-07
> **Response to Reviewer KbFY**
>
> Dear Reviewer KbFY,
>
> Thank you for your review of our paper and for your positive feedback. Below, we respond to each of the points you raised.
>
> - **Reviewer**: Question about the discussion on Thm. 1.
>
> - **Response**: Thank you for raising this interesting question. Our variance bound - as revealed by Theorem 1 - is $O(\alpha (\tau + \delta) \sigma^2)$, where $\alpha$ is the step-size, $\tau$ is the mixing-time of the underlying Markov chain, $\delta$ captures the effect of compression, and $\sigma^2$ is the variance of our noise model. Now the key thing to note is that even in the absence of compression,  the $O(\alpha \tau \sigma^2)$ term would inevitably persist - this linear dependence on the mixing time $\tau$ is known to be information-theoretically optimal (Nagaraj et al., 2020). We had mentioned this point in Section 6 of our paper.
>
> Thus, given that the effect of compression and the mixing-effect of the Markov chain show up *additively*, and the latter effect is *unavoidable*, as long as the compression factor $\delta$ is of the same order as the mixing time $\tau$, the resulting variance bound would be no worse than in the case without compression. Said differently, a Markov chain that mixes slowly, i.e., has a larger $\tau$, provides a greater leeway for aggressive compression. Finally, we note that once the policy to be evaluated is fixed, we have no control over the mixing-time $\tau$ of the Markov chain it induces. However, we do have control over the compression parameter $\delta$. Thus, it makes sense to tune $\delta$ to be of the same order as $\tau$, since this leads to no loss in performance relative to the uncompressed setting.
>
> We hope this clarifies your concern. We have briefly added parts of the above discussion to the revised version of our paper.
>
> -  **Reviewer**:  Abuse of notation: The authors use $r_t$ to denote both the reward and the convergence error (Thm. 1), which probably leads to confusion.
>
> - **Response**: Thank you for noticing this! We had missed this point in our original submission. Throughout the revised paper, we have now used $d_t$ to represent the MSE error at time-step $t$.
>
> In addition to the above changes, we have incorporated your pointers regarding grammatical issues in our revised paper.
>
> We hope that the above changes address all your concerns. We would be happy to answer any other questions you may have.
>
> Sincerely,
>
> Authors.

---

### Author Response · Authors · 2024-02-07
**We have responded to comments, and revised our paper**

Dear Action Editors,

Thank you once again for handling the submission of our paper. We were happy to see the positive feedback on our work from all the Reviewers. We have responded to each Reviewer individually and addressed all their comments/queries. We have also revised the paper according to the suggestions that were provided. All the main changes appear in green for ease of reference.

We would like to thank the Reviewers for their valuable feedback, and would be happy to engage in further discussion if needed.

Sincerely,

Authors.

---

> ### Comment · Reviewer_GFLY · 2024-02-08
> **Are all replies visible?**
>
> Dear authors,
>
> You mentioned that you replied to all reviews, however I can only see a response to reviewer KBfY. Can you check whether the permissions are set correctly?

---

> > ### Author Response · Authors · 2024-02-08
> > **Adjusted Permissions**
> >
> > Dear Reviewer GFLY,
> >
> > Thank you for pointing this out. We have now set the permissions correctly.
> >
> > Please let us know if we have addressed your concerns appropriately.
> >
> > Sincerely,
> >
> > Authors.

---

### Decision · Action_Editor_GXQD · 2024-03-28

**Recommendation:** Accept as is

**Comment:**

The paper is thoroughly written and introduces interesting proof techniques / approaches to the considered RL problems. As such it can be a worthwhile contribution to the relevant TMLR community. Also the reviewers recommend acceptance of the submisson and appreciated the authors' updates made during the discussion phase. Hence, following their recommendations, I am recommending acceptance of the paper.

**Audience:**

Yes, the paper is relevant to parts of the TMLR audience.

**Claims And Evidence:**

The made claims in form of mathematical statements are thoroughly proven. Additionally, simulation experiments augment those theoretical results.